# Proteogenomic analysis reveals adaptive strategies for alleviating the consequences of aneuploidy in cancer

Jan-Eric Bökenkamp [1,3], Kristina Keuper [1,2,3], Stefan Redel [1], Karen Barthel [1], Leah Johnson [1], Amelie Becker [1], Angela Wieland [1], Markus Räschle [1] & Zuzana Storchová [1✉]

## Abstract

**Aneuploidy is prevalent in cancer and associates with fitness advantage and poor patient prognosis. Yet, experimentally induced aneuploidy initially leads to adverse effects and impaired proliferation, suggesting that cancer cells must adapt to aneuploidy. We performed in vitro evolution of cells with extra chromosomes and obtained cell lines with improved proliferation and gene expression changes congruent with changes in aneuploid cancers. Integrated analysis of cancer multi-omics data and model cells revealed increased expression of DNA replicative and repair factors, reduced genomic instability, and reduced lysosomal degradation. We identified E2F4 and FOXM1 as transcription factors strongly associated with adaptation to aneuploidy in vitro and in cancers and validated this finding. The adaptation to aneuploidy also coincided with specific copy number aberrations that correlate with poor patient prognosis. Chromosomal engineering mimicking these aberrations improved aneuploid cell proliferation, while loss of previously present extra chromosomes impaired it. The identified common adaptation strategies suggest replication stress, genomic instability, and lysosomal stress as common liabilities of aneuploid cancers.**

**Keywords** Adaptation; Aneuploidy; Cancer; Chromosomal Engineering; FOXM1
**Subject Categories** Cancer; Cell Cycle; Proteomics

## Introduction

Aneuploidy, an imbalanced chromosome number that deviates from the multiples of a chromosome set, is a hallmark of cancer found in ~90% of solid tumors and ~70% of hematopoietic cancers (Vasudevan et al, 2021). This establishes aneuploidy as one of the most common types of genetic alterations in cancer. Yet, experiments conducted in budding yeast, plants, drosophila, and mammalian cell cultures demonstrate that gain or loss of a single chromosome often has detrimental consequences (Chunduri and Storchova, 2019; Stingele et al, 2012; Torres et al, 2007; Williams et al, 2008). In an acute response to aneuploidy immediately after chromosome missegregation in human cells, the cells show differential regulation of autophagy and lysosomal stress, metabolic changes, and genomic instability (Garribba et al, 2023; Ohashi et al, 2015; Santaguida et al, 2015; Simoes-Sousa et al, 2018). Human cells with a constitutive gain of an extra chromosome usually show reduced proliferation, conserved changes in gene expression patterns, altered protein homeostasis, and genomic instability (Nicholson and Cimini, 2015; Stingele et al, 2012; Vigano et al, 2018). The stresses arise due to a low overexpression of several hundreds of proteins encoded on the extra chromosome, which overloads the protein folding machinery, leading to the accumulation of cytosolic protein aggregates, deregulation of autophagy, and increased proteasomal activity (Donnelly et al, 2014; Oromendia et al, 2012; Santaguida et al, 2015). The genotoxic stress is characterized by reduced expression of key replicative factors, replication stress, the accumulation of DNA damage, and increased occurrence of de novo chromosomal rearrangements (Garribba et al, 2023; Passerini et al, 2016; Santaguida et al, 2017). The detrimental effects of aneuploidy also show in vivo tumor-suppressive effects, as trisomy in human cancer cells leads to reduced tumor formation in nude mice, and the few arising tumors mostly lose the extra chromosome (Sheltzer et al, 2017). These characteristic features of aneuploidy document that a gain of an extra chromosome puts a burden on the cellular machinery.

Although aneuploidy often results in reduced proliferation initially, chromosome copy number changes are frequent in cancer. While some aneuploidies might be random, there are recurrent patterns of copy number changes in cancer genomes suggesting that certain aneuploidies provide the cells with a selection advantage shaping the cancer genome by selection forces (Adell et al, 2023; Ben-David and Amon, 2020; Lakhani et al, 2023; Shih et al, 2023; Vasudevan et al, 2021). For example, the gains of chromosome arms 8q and 20q are prevalent across many different cancer types (Beroukhim et al, 2010), while loss of chromosome arm 3p is found in squamous cancer, loss of chromosome 10 in glioblastoma, and 13q gain in gastrointestinal tumors (Taylor et al,

[1]RPTU Kaiserslautern-Landau, Paul-Ehrlich Strasse 24, 67663 Kaiserslautern, Germany. [2]Danish Cancer Institute, Strandboulevarden 49, 2100 Copenhagen, Denmark. [3]These authors contributed equally: Jan-Eric Bökenkamp, Kristina Keuper. ✉E-mail: storchova@bio.uni-kl.de

2018), documenting that some aneuploidies provide specific advantages to cancer cells as well as facilitate evolution of drug resistance (Ippolito et al, 2021; Lukow et al, 2021; Shoshani et al, 2021; Trakala et al, 2021). Cancer-specific aneuploidy patterns are selected likely due to the presence of individual genes, which bring a context-dependent advantage. Indeed, the number of tumor suppressors and oncogenes on individual chromosomes correlates with the frequency of gains and losses of this chromosome in cancer (Davoli et al, 2013; Shih et al, 2023). Recent results demonstrate that expression changes due to gain of an individual chromosome might be sufficient to provide a proliferation advantage. For example, gain of chromosome arm 1q in some p53-proficient cancers improves their proliferation, possibly due to the increased expression of MDM4, which suppresses p53 signaling (Girish et al, 2023). Thus, specific cancer aneuploidies, similarly as oncogenic gene mutations, are selected according to their positive effect on cellular fitness (Davoli et al, 2013; Shih et al, 2023).

Despite the potential advantages associated with the gains of certain chromosomes, the concomitant burden arising from increased expression of hundreds of genes with no direct advantage remains. This is evidenced in engineered aneuploid human cell lines where the impairment of proliferation scales with the number of genes located on the extra chromosome (Hintzen et al, 2022; Kneissig et al, 2019). Diploid cancer cells tend to maintain more often chromosome arm aberrations, while cancer cells that underwent whole-genome doubling can tolerate whole chromosome aberrations (Prasad et al, 2022). In cells with elevated chromosomal instability (CIN +) induced by spindle assembly checkpoint inhibitors, survivors with complex karyotypes arise, but these cells become soon outcompeted by cells carrying smaller, more precise copy number alterations (Adell et al, 2023). Thus, even if a chromosome gain provides a selective advantage to the cells, they must concomitantly adapt to the adverse effects of excessive genetic material.

Previous research suggested potential modifications required to mitigate the adverse effects of aneuploidy. In budding yeasts, improved proliferation in different disomic strains was associated with gross chromosomal rearrangements and several point mutations, in particular, in genes related to ubiquitin–proteasome pathways, such as *UBP6*, *UBR1*, and others, suggesting that changes in protein turnover may help to improve the fitness of aneuploid yeasts (Torres et al, 2010). This is further corroborated by the fact that the deubiquitinase Ubp3 improves the proliferation of budding disomic yeast; its homolog in mammals, USP10, also improves the proliferation of human aneuploid cell lines arising from induced missegregation (Dodgson et al, 2016). In addition, increased RNA and protein degradation improves the proliferation of aneuploid cells by alleviating the consequences of proteotoxic stress (Ippolito et al, 2024). Recently, a systematic analysis of RPE1 cells with variable aneuploidy pointed out that reduced chromosomal instability and reduced inflammation are typical for adaptation to aneuploidy (Hintzen et al, 2024). Together, these results suggest that eukaryotic cells can improve their proliferation despite aneuploid karyotypes by processes largely affecting protein turnover and genome stability. Identification of the adaptive processes can provide new opportunities for therapies targeting aneuploid cancer cells.

The mechanisms by which cells adapt to tolerate or counteract the deleterious consequences of chromosome gain remain elusive.

Here, we investigated the mechanisms that enable human cells and cancers to adapt to aberrant karyotypes. To this end, we conducted in vitro evolution experiments, where constitutive polysomic cell lines were passaged over an extended period. The polysomic cell lines displayed improved proliferation after in vitro evolution while maintaining the extra chromosome. The improved proliferation of aneuploid cells was associated, among others, with reduced DNA damage and genomic instability, reduced replication stress, and an altered expression of factors involved in DNA replication and lysosomal degradation. The identified pathways and genes tightly correlate with pathways deregulated in aneuploid cancers. The gene expression changes in model cells aligned with gene expression changes observed in aneuploid cancers and uncovered the transcriptional factors E2F4 and FOXM1 to critically contribute to adaptation to chromosome gains, which we experimentally validated. Moreover, we demonstrated that specific chromosome 5 aberrations that alleviated the consequences of aneuploidy in vitro also correlated with poor patient prognosis. Our integrated approach elucidates the molecular landscape of adaptation to aneuploidy and identifies potential vulnerabilities in aneuploid cancers.

## Results

### Cells with additional chromosomes improve their proliferation over time

To study adaptive changes occurring during the evolution of cells harboring defined chromosome gains, we used newly prepared aneuploid cell lines into which chromosome 5 or chromosome 21 was introduced by microcell-mediated chromosome transfer (MMCT), as previously described (Kneissig et al, 2019, Stingele et al, 2012). Two cell types, HCT116 and RPE1, were used to generate polysomic cells with a trisomy of chromosome 5 (hereafter Htr5 and Rtr5), a tetrasomy of chromosome 5 (Hte5), or a trisomy of chromosome 21 (Rtr21). Gain of an extra chromosome leads to cellular stress manifested by reduced proliferation compared to the parental control (Stingele et al, 2012). To elucidate how human cells adapt to chromosome gain, we cultured the cell lines in a nutrient-rich medium for 50 passages in biological triplicates (p50, ~150 generations, Fig. 1A). The medium lacked the antibiotics required for the selection of the extra chromosome, allowing for the potential loss of any chromosome. After the in vitro evolution, almost all polysomic cells exhibited a significant increase in proliferation, quantified by the increased area under the growth curves (AUC), in comparison to the original, p0 population (Fig. 1B; Appendix Fig. S1A–C; Dataset EV1). The increase was observed in 10 out of 12 independent replicates, regardless of the specific cell line or the added chromosome, albeit to a varying degree (Appendix Fig. S1C). In contrast, the proliferation rate of the parental diploid cell line remained nearly unchanged (Fig. 1B; Appendix Fig. S1A,C). Despite the significant increase in proliferation, all evolved polysomic clones exhibited less efficient proliferation than the parental disomic cells (Fig. 1B; Appendix Fig. S1A). Thus, human cell lines adapt to chromosome gains after culturing over extended periods of time to counteract the adverse effects that initially caused their poor proliferation.

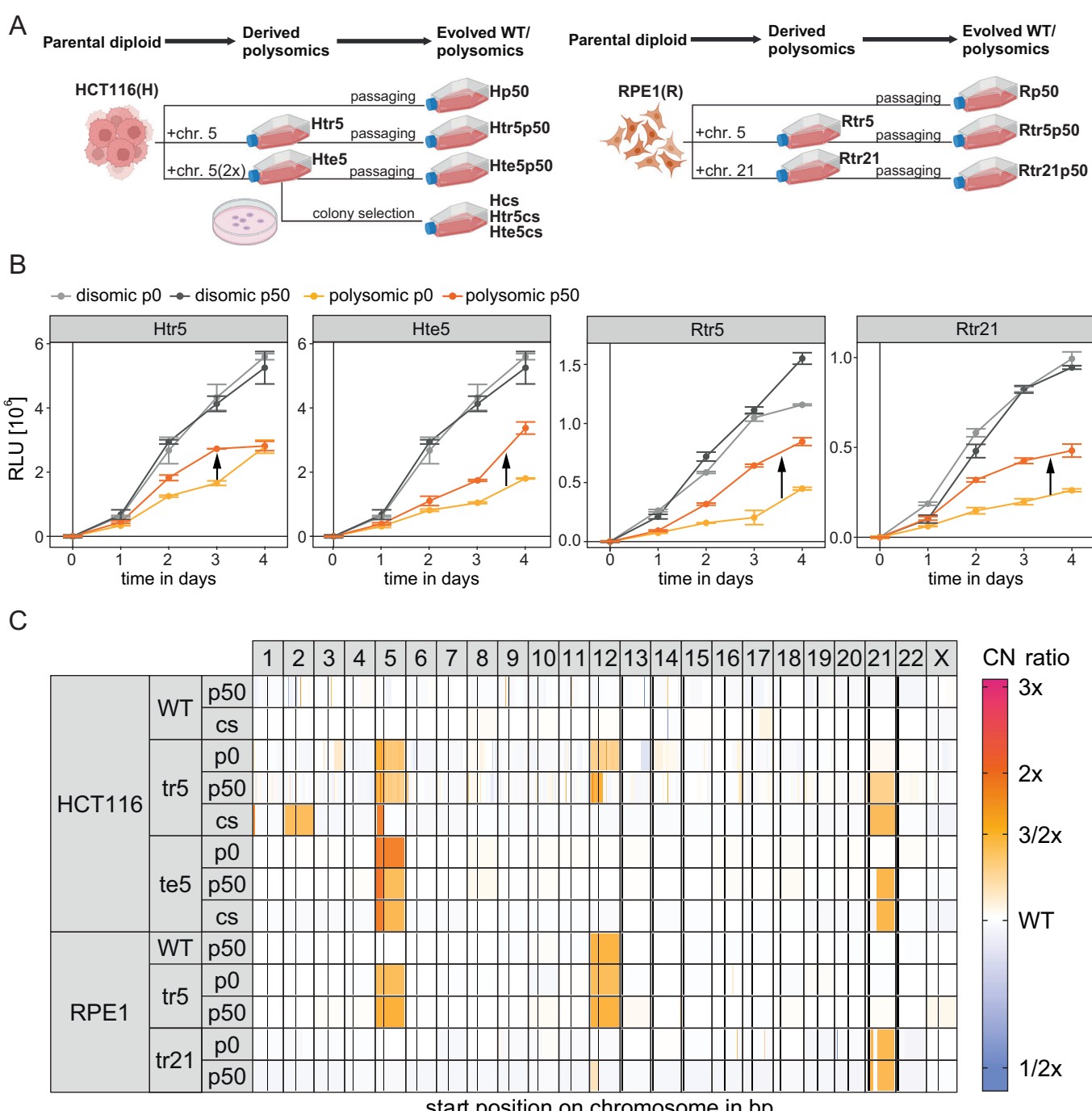

**Figure 1. Improved proliferation following in vitro evolution of polysomic human cells.**

(A) Schematic illustrating the experimental setup of in vitro evolution applied to polysomic model cell lines. (B) Population growth of the cell lines before and after in vitro evolution evaluated by MTT assay and normalized to the time point 0. Points represent mean relative light units (RLU) of the seeded triplicates. Error bars represent SEM. (C) Chromosome copy number ratios relative to parental, unevolved WT for each cell line (row) and chromosome (column). Vertical bars within chromosomes depict the location of centromeres. Source data are available online for this figure.

## Loss of the additional chromosome is not a pre-requisite for improved proliferation in polysomic cells

To further characterize the adaptive processes that help aneuploid cells to overcome the initial growth inhibition, we subjected the evolved clones to an in-depth karyotype analysis. From each cell line, we selected the clonal lines that showed the highest increase in proliferation compared to the unevolved p0 cell line (Fig. 1B and Appendix Fig. S1A) and subjected them to either low-coverage whole-genome sequencing (WGS) or array-based comparative

genomic hybridization (aCGH) (Dataset EV2). The in vitro evolved clones maintained the extra chromosomes, and additional copy number changes affecting chromosome arms, as well as smaller regions, were observed (Fig. 1C; Dataset EV2). For instance, we identified subclonal losses of the q-arm of chromosome 5 in both Htr5 and Hte5 cell (Fig. 1C). In addition, full or partial gains of chromosome 21, and less frequently 2, were observed in polysomic cell lines after in vitro evolution (Fig. 1C). The trisomic cell lines derived from noncancerous RPE1 cells maintained the extra chromosomes as well and displayed no clonal changes in their karyotypes, in line with the previous observation that polysomy leads to only mild genomic instability in RPE1 cells (Passerini et al, 2016). The parental RPE1 gained an additional chromosome 12 and showed some characteristics of trisomic cells (Fig. 1C). The gain of chromosome 12 is in line with previous observation that RPE1 cells tend to gain this chromosome (Potapova et al, 2016). Based on these data, we conclude that the improved proliferation did not occur simply due to the loss of the extra DNA. Indeed, we found that the total amount of altered DNA in the adapted polysomic cells relative to WT only weakly correlates with the observed proliferation improvement (Appendix Fig. S1D).

In an alternative approach, we plated the HCT116-derived cells at a low density and selectively collected the largest, best-proliferating colonies originating from a single cell (referred to as colony selection or "cs," Fig. 1A). These cells showed similar chromosomal copy number changes (Fig. 1C) suggesting that the cells with these specific aberrations were present in the original population and the prolonged passaging allowed them to expand.

Cell proliferation can be enhanced through reduced duration of the phases of the cell cycle, or decreased rates of senescence or cell death. Cells after chromosome gain exhibit prolonged G1 phase duration and subtle changes in the percentage of senescent and dead cells compared to the parental cell lines. We found that the fraction of cells in the G1 phase decreased, while the fraction of cells in the S phase increased following in vitro evolution in polysomic clones (Appendix Fig. S2A,B). There were no uniform changes in the fraction of dead and senescent cells within the population which would fully explain the proliferation improvement (Appendix Fig. S2C,D). Moreover, the p53 abundance and signaling were not changed in the evolved clones (Appendix Fig. S2E). We conclude that the in vitro evolved polysomic cell lines improved their proliferation despite maintaining the extra chromosome and functional p53 signaling. Thus, the obtained clones provide a valuable model for identifying the physiological changes required for adaptation to chromosome gain.

## Analysis of gene expression changes reveals differentially regulated pathways in evolved polysomic cells

To elucidate the molecular mechanisms that enable the enhanced proliferation of the adapted polysomic cells, we analyzed the global proteome before and after in vitro evolution by quantitative mass spectrometry using a tandem mass tag (TMT)-based quantification strategy (Dataset EV3). These results were then compared to aneuploidy-specific expression signatures extracted from the TCGA and CPTAC pan-cancer data repositories by independent bioinformatic analysis (Fig. 2A) (Cancer Genome Atlas Research,

Weinstein et al, 2013; Hoadley et al, 2018, Data ref: Hoadley et al, 2018; Li et al, 2023; Data ref: Li et al, 2023).

Normalizing the protein abundance values to the disomic parental cell lines to obtain log2 fold changes (FC) and visualizing them grouped by chromosome identity (Appendix Fig. S3A; Dataset EV3) confirmed the increased expression from the extra chromosomes in evolved cell lines. The abundance of proteins encoded on the supernumerary chromosome should be increased by 50% for trisomic cells and doubled in cells with two additional chromosome copies. However, previous work revealed gene dosage compensation, which reduces the abundance of some proteins towards levels measured in diploids (Schukken and Sheltzer, 2022; Stingele et al, 2012). Efficient dosage compensation might reduce the burden on polysomic cells and thus improve their proliferation without loss of the additional chromosome. By calculating the difference between average log2 DNA dosage and protein abundance on aneuploid chromosome arms, we found on average a slight, but non-significant increase of dosage compensation after in vitro evolution, as the degree of dosage compensation was increased for six chromosome arms after in vitro evolution experiments, while there was a reduction or no change in two cases (Appendix Figs. S3A and S4A,B). Next, we investigated the potential of individual proteins to drive the proliferation improvement through increased dosage compensation. Analysis of the differential abundance of individual tumor suppressor genes (TSG) and oncogenes (OG) (Sondka et al, 2018, Data ref: Sondka et al, 2018) located specifically on the extra chromosomes showed a few cases of significant down- and upregulation of protein abundance after in vitro evolution, such as the downregulation of TSG *SDHA* in Hte5 and OG *U2AF1* in Rtr21, but no general pattern was observed (Appendix Fig. S4C,D). Next, we asked whether the protein abundance after in vitro evolution resembles the expression patterns observed in cancers with the same chromosome arm gains more closely than before evolution. To this end, we analyzed CPTAC tumors with and without gains of the respective chromosome and identified the proteins whose abundance scaled with the gain (category "scaling") or not ("compensated"). We then tested whether the proteins compensated in cancers became less abundant after in vitro evolution or vice versa for scaling proteins (Appendix Fig. S4E,F). No general similarity in protein abundance changes in cancers and in in vitro evolved cells was observed, although we identified certain proteins, such as SKP2 and HMGCS1, whose abundance scaled with chromosome arm 5p gain in tumors and which were overexpressed in Htr5 and Hte5 after in vitro evolution. Only Hte5 showed a significant decrease in average protein abundance of proteins compensated in tumors with chromosome arm 5p gain (mean log2 difference of $-0.11$, $P < 0.05$). We conclude that the improved proliferation in adapted polysomic cells cannot be explained entirely by an enhanced dosage compensation of genes located on the supernumerary chromosomes.

We next applied standard differential expression analysis to identify genes whose expression changes show similarity among polysomic cell lines. Upon chromosome gain, at p0, we identified 55 commonly downregulated and 40 upregulated proteins (FDR < 0.05, Fig. EV1A), confirming previous observations that chromosome gain elicits a common gene expression dysregulation independently of the gained chromosome or the cell line (Sheltzer et al, 2012; Stingele et al, 2012). In contrast, the paths to adaptation to polysomy are less

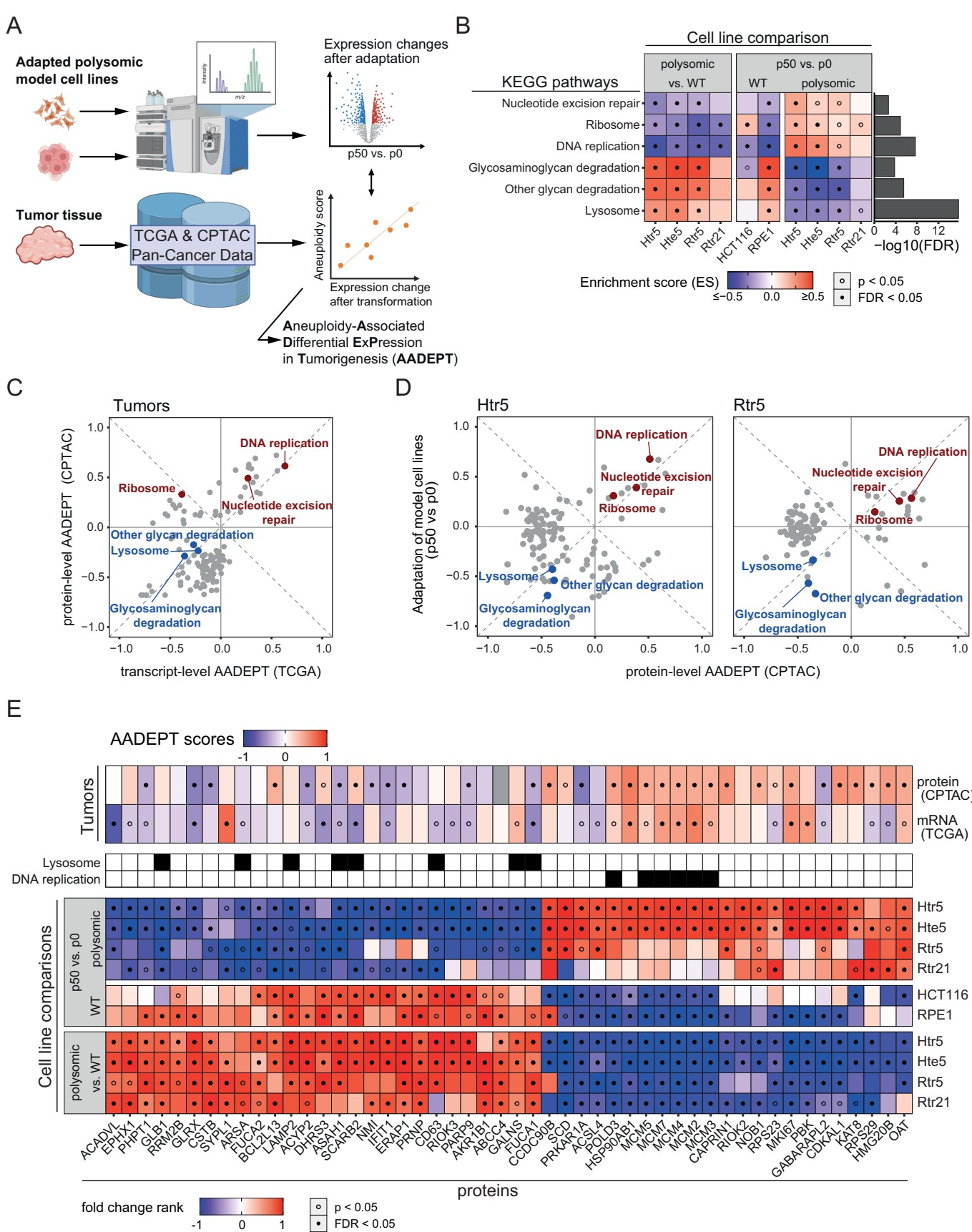

**Figure 2. Expression changes in genes and pathways after in vitro evolution of polysomic human cell correlate with changes in human aneuploid cancers.**

(A) Schematic depiction of the data analysis pipeline including the calculation of correlations between aneuploidy scores and gene expression changes after transformation (primary tumor vs. normal tissue) in TCGA and CPTAC tumor samples—AADEPT (Aneuploidy-Associated Differential ExPression in Tumorigenesis). (B) Enrichment scores of pathways significantly altered in at least three polysomic cell lines after in vitro evolution. The statistical significance of fold change enrichment in individual comparisons is indicated by the points within each tile. FDR bars (cut-off at 0.05) are based on multivariate ANOVA for overall enrichment between p50 vs. p0 comparisons. (C) 2D pathway enrichment of AADEPT scores on transcript- (TCGA) and protein-level (CPTAC), showing enrichment scores for pathways with FDR < 0.05 and highlighting those with shared enrichment in adapted polysomic cell lines (blue—negative, red—positive). (D) 2D pathway enrichment comparing protein-level AADEPT scores with fold changes in polysomic Htr5 and Rtr5 after adaptation, showing enrichment scores for pathways with FDR < 0.05. (E) Top 50 proteins according to their relevance score, which shows the correlation of their AADEPT scores (above) and their relative abundance changes in adapted polysomic model cell lines (below). Fold changes are ranked and scaled from −1 (lowest) to 1 (highest). Statistical significance for individual proteins was determined using empirical Bayes moderated Student's *t* tests for cell line comparisons, and using Spearman rank testing for tumor AADEPT scores (see "Methods").

convergent, as only two upregulated and three downregulated factors were shared among all evolved polysomies, but not in the parental cell lines (Fig. EV1A). These were GLB1 (encoding lysosomal hydrolase galactosidase beta 1), CTSA (encoding lysosomal peptidase cathepsin Z) and BCL2L13 (inhibitor of apoptosis from the BCL2 family), whose abundance was commonly decreased with adaptation, while PRPF19 (coding for ubiquitin-protein ligase involved in splicing and DNA repair) and OAT (encoding mitochondrial ornithine aminotransferase), were more abundant after evolution (Fig. EV1B).

With this low turnout of individual proteins, we sought a better insight into the functional changes by using the fold changes of relative protein abundances between p50 and p0 to test pathways for significant enrichment (KEGG) (Kanehisa, 2019; Data ref: Liberzon et al, 2023). This analysis revealed a set of six pathways including DNA replication, nucleotide excision repair (NER) and ribosome, which became downregulated upon chromosome gain, but showed increased expression following in vitro evolution in both, HCT116 and RPE1-derived polysomic cell lines (Fig. 2B; Dataset EV4, see "Methods"). In addition, lysosomal pathways essential for the degradation and recycling of biomolecules, were upregulated after the addition of a chromosome, but showed a marked downregulation after 50 passages (Fig. 2B). Thus, we identified pathways changes associated with cellular response to aneuploidy that were reversed after in vitro evolution, independently of the parental cell line and added chromosomes.

## Adaptations to aneuploidy in vitro correspond with the adaptations in aneuploid cancer

We next aimed to leverage our model of adapted polysomic cell lines together with publicly available multi-omics data of patient-derived aneuploid tumors to identify molecular mechanisms that facilitate the adaptations to aneuploidy in an integrative bioinformatic analysis. To this end, we used the transcriptome data from the TCGA PanCanAtlas and the proteome data from CPTAC for both cancerous and adjacent noncancerous tissues, as well as the corresponding genomic copy number data (Data ref: Hoadley et al, 2018; Data ref: Li et al, 2023). We then calculated the difference in gene expression between primary tumors and the respective patient's normal tissue and correlated the expression changes with the tumor's aneuploidy score (AS), which was derived as a sum of altered chromosome arms (Taylor et al, 2018) (Dataset EV4). With this approach, we obtained a comparative measure called AADEPT (Aneuploidy-Associated Differential ExPression during Tumorigenesis), which quantifies the cellular adaptations in cancer transcriptome and proteome of malignant tumors linked to varying

degrees of aneuploidy (Appendix Fig. S5A,B; Dataset EV5). Comparing the calculated AADEPT scores from transcript- (TCGA) and protein-level (CPTAC) showed no similarities in the top scoring proteins (Fig. EV1C), but a striking correlation of pathway enrichments, with DNA replication and nucleotide excision repair being upregulated and lysosomal pathways being downregulated (Fig. 2C). Most strikingly, all pathways that were universally deregulated in the evolved polysomic model cells correlated with the AADEPT in transformed aneuploid tissues except for ribosomes on transcript-level. The quantified AADEPT scores also correlated strongly with the changes in evolved polysomic cell lines, including an overlap in overabundance of RNA polymerase, spliceosome, and DNA repair proteins, and reduced abundance in the lysosome and lysosomal degradation pathways (Figs. 2D and EV2A). Finally, we explored a possible overlap with the hallmark gene sets (Liberzon et al, 2015; Data ref: Liberzon et al, 2023) and its enrichment with AADEPT scores. Again, we observed a striking level of congruence with the protein abundance changes in adapted model cells, with a stronger similarity observed in the polysomic cell lines derived from cancerous HCT116 (Fig. EV2B). Specifically, E2F target proteins, MYC targets, and G2/M checkpoint proteins were more abundant, whereas innate immune response proteins reduced upon adaptation to aneuploidy in both model cell lines and malignant tumors. Thus, the significantly differentially regulated pathways after in vitro evolution of polysomic cells strongly correlate with the differentially regulated pathways in aneuploid tumors.

Since the pathways analysis revealed similarities in expression changes, we reevaluated the protein-level changes shared among evolved polysomic cell lines and cancer cells using a less conservative approach. To this end, we merged the proteome fold change values to calculate for each protein a "relevance score", which represents the expression changes between p50 vs. p0, and between polysomic vs. parental cell lines at p0. Moreover, this score penalizes the changes in protein abundance occurring during the evolution of wild-type cells to emphasize the aneuploidy-specific adaptations (Appendix Fig. S5C; Dataset EV3). By this approach, we determined the top 50 proteins representative of the reversion of aneuploidy-induced expression changes after in vitro evolution observed on pathway-level. Strikingly, many of these changes correspond with the AADEPT score on both transcript- and protein-level in TCGA and CPTAC patient samples (Fig. 2E). For example, we identified upregulation of several replication proteins (five out of the six subunits of the MCM2-7, or DNA polymerase delta 3), upregulation of heat shock protein HSP90, and downregulation of factors related to lysosome function (LAMP2, CSTB)

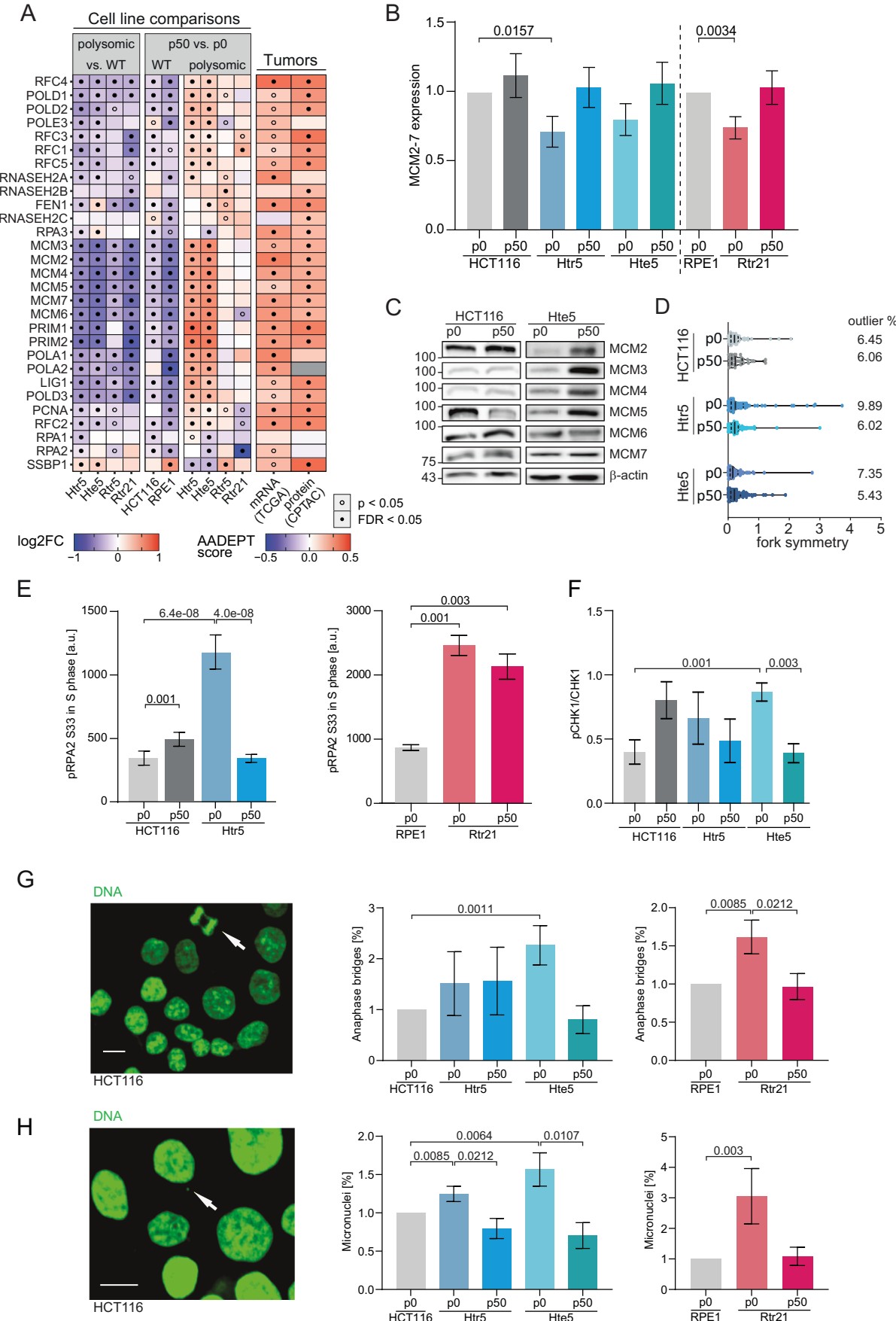

**Figure 3. Increased expression of replicative factors and reduced replication stress, genomic instability, and DNA damage after in vitro evolution.**

(A) Expression changes of factors involved in replication after chromosome gain (polysomic vs. WT) and after in vitro evolution (p50 vs. p0) and their comparison with the AADEPT scores (as in Fig. 2E). (B, C) Immunoblotting of the MCM2-7 subunits (Appendix Fig. 6A shows an extended version of the immunoblotting) and the quantification in the model cell lines. The values for all subunits were normalized to the respective parental cell line and pooled together (n: 16–18, each MCM subunit at least 2×). (D) Fork symmetry in the model cell lines before and after evolution and the 0.5 percentile of outliers (identified with ROUT). At least three biological replicates were analyzed, n ≥ 33 in each experiment. (E) Median of relative phosphorylation of the RPA2 on the S33 during S phase evaluated by flow cytometry. Three biological replicates analyzed with two technical replicates each are shown. (F) Quantification of relative phosphorylation of CHK1 on S345 evaluated by immunoblotting. Three biological replicates were analyzed. (G) Representative image and quantification of the occurrence of anaphase bridges in model cell lines. At least three biological replicates were analyzed (n: 3–12), at least 30 anaphase cells from each were evaluated. (H) Representative image and quantification of the occurrence of micronuclei in model cell lines. Three biological replicates were analyzed with two technical replicates each, at least 2000 nuclei were evaluated in each experiment. All cell lines were normalized to the respective wild type. Data information: Bar plots show mean with SEM. P values were calculated using unpaired Student's t test. Scale bars in microscopy images = 10 μm. Source data are available online for this figure.

as well as factors related to interferon response (e.g., IFIT1). Of note, the degree of changes in the noncancerous RPE1 models is generally lower in both the multivariate pathway enrichment (Fig. 2B) as well as in the protein relevance analysis (Fig. 2E). Based on these data, there are common pathways and proteins that emerge as decisive factors influencing proliferation in aneuploid cells. Consequently, these pathways may constitute the central anti-proliferative mechanisms associated with chromosome gains.

## Gene mutations in evolved polysomic cells

To identify gene mutations that potentially contribute to improved fitness, we performed deep whole-genome sequencing at 30× of evolved and unevolved Rtr21 and Htr5, and their respective parental cell lines. Identified variants were filtered to exclude silent mutations and mutations that occurred also in parental cell lines. We found 135 and 14 heterozygous, nonsynonymous mutations after the evolution, which mapped to 81 and 12 unique genes exclusively mutated in evolved Htr5 and Rtr21, respectively (Fig. EV3A; Dataset EV7). The high number of mutations in HCT116-derived cell lines is likely due to their mismatch repair deficiency. By aligning the mutated genes in Htr5p50 with the respective proteome data, we obtained 25 mutated genes with known protein abundance and confirmed that most of the variants were associated with a significant reduction after evolution. Interestingly, 20 out of 25 showed >1% prevalence in colon adenocarcinoma and pan-cancer tumors (Fig. EV3B). For example, we identified mutations in *IRAK4* encoding a kinase that activates NF-ƙB and is essential for most innate immune responses, in E3 SUMO-protein ligase *RANBP2*, which facilitates SUMO1 and SUMO2 conjugation and contributes to mRNA export, or in *NPBL1*, a gene encoding for a protein required for the cohesin loading onto DNA. Mutations in these genes can be found in 3.5%, 9.1%, and 12.1% of colorectal adenomas, respectively (Fig. EV3B). Similar analysis in Rtr21p50 yielded mutations in five genes, all related to DNA repair and cell cycle regulation: *TP53BP1*, a key protein required for DNA damage response and p53 accumulation; *TAOK1* kinase that activates p38/MAPK14 cascade upon G2 DNA damage; *MYCBP1* whose product regulates the cAMP and mTOR signaling pathways, and autophagy; *ERCC6* encoding DNA-binding protein important in transcription-coupled excision repair; and finally *SETDB1*, histone methyltransferase which regulates transcriptional repression (Fig. EV3C). However, none of the identified mutations was linked specifically to aneuploidy in cancer, as we found by analyzing the association between the identified gene

mutations and aneuploidy score in respective tumors; in fact, they were rather anticorrelating with aneuploidy (Fig. EV3D). Moreover, there was no overlap between the genes mutated after the in vitro evolution of HCT116 and RPE1-derived polysomic cells, nor did we detect any enrichment for specific pathways, tumor suppressor genes or oncogenes. Nevertheless, we decided to follow up on *TP53BP1*, which carried a deleterious heterozygous mutation associated with a significant reduction in protein abundance in Rtr21p50. Indeed, the depletion of *TP53BP1* partially improved proliferation in non-evolved Rtr21 cells (Fig. EV3E,F). Together, we demonstrate that while there is no general association between the identified gene mutations and aneuploidy in cancer, some of them may confer a selective advantage in certain polysomic cell lines. In the following experiments, we focused on a broader analysis of the pathways affected by in vitro evolution that were commonly shared among the analyzed cell lines and aneuploid cancers.

## Restoration of DNA replication and repair processes reduce genomic instability in polysomic clones after in vitro evolution

Among the most prominent changes observed upon in vitro evolution was the restored abundance of proteins that are assigned to "DNA Replication" (KEGG), which are generally downregulated upon chromosome gain (Fig. 3A) (Ohashi et al, 2015; Passerini et al, 2016). Many of the identified proteins were also identified by their high AADEPT scores in TCGA and CPTAC (Fig. 3A). Among the most affected proteins were the subunits of the MCM2-7, a heterohexameric complex required for replication licensing, which together with CDC45 and GINS complex acts as a replicative helicase during DNA replication (so-called CMG), which we confirmed by immunoblotting of the MCM2-7 subunits in non-evolved and evolved polysomic cells (Figs. 3B,C; Appendix Fig. S6A). No changes in MCM2-7 expression were observed in the HCT116 (RPE1p50 was not used due to the gain of chromosome 12, Fig. 1B). We asked whether the changes in abundance of replication factors have an impact on the replication dynamics. To this end, we used DNA combing and visualized full track lengths of labeled replicating DNA (Appendix Fig. S6B). Quantification of the replication rate revealed a reduction upon in vitro evolution in Rtr21 and Hte5 (Appendix Fig. S6C). Next, we measured the inter-origin distance, where we observed a significant increase in the inter-origin distance of Hte5 cell line, carrying the highest amount of additional DNA. This is likely due to the reduced MCM abundance in polysomic cells, as the inter-origin distance

was previously shown to increase in cells where the MCM2-7 proteins were depleted (Ge et al, 2007). The inter-origin distance was then reduced upon evolution (Appendix Fig. S6D). Since replication stress often results in disturbances of the fork symmetry, we measured the neighboring fork lengths. Strikingly, the percentage of asymmetric forks was increased in cells with an additional chromosome, but this percentage decreased after evolution (Fig. 3D), suggesting reduced replication defects after in vitro evolution.

Evaluation of the replication protein RPA2, which becomes phosphorylated at RPA2 S33 by the ATR kinase upon replication stress, showed that the phosphorylation in the S-phase increases upon chromosome gain, in line with our previous findings (Passerini et al, 2016). Importantly, the phosphorylation of RPA2 was reduced in evolved polysomic cell lines (Fig. 3E). Similar reduction was observed in the phosphorylation of CHK1 S345, another substrate of the ATR kinase (Fig. 3F). Accordingly, the occurrence of anaphase bridges, which results from defective DNA repair or replication and increases upon chromosome gain, was reduced in evolved Hte5p50 and Rtr21p50 cell lines compared to the respective p0 (Fig. 3G). Similarly, accumulation of micronuclei, another marker of genomic instability, and the expression of double-strand break marker, γH2Ax, were also reduced in evolved polysomic cells, indicating decreased DNA damage after in vitro evolution (Fig. 3H; Appendix Fig. S6E). We conclude that one of the strongest changes upon in vitro evolution of polysomic cells is an increased expression of DNA repair and replication factors and reduced genomic instability, which is reflected by reduced checkpoint signaling and decreased DNA damage.

Our analysis also revealed a reduction of lysosomal gene expression in the evolved aneuploid cells, such as lysosomal cathepsins CTSA and CTSB, LAMP1, and 2 (lysosomal-associated membrane protein 1 and 2) and others (Appendix Fig. S7A). It should be noted, however, that the AADEPT scores of these proteins were more heterogenous, suggesting a weaker overlap with gene expression changes in aneuploid cancers. Immunofluorescence imaging of LAMP1 revealed fewer foci in the evolved cells, while lysotracker and LAMP2 analysis indicates rather mild and heterogenous effects (Appendix Fig. S7B,C). Thus, abundance of lysosomes and lysosomal proteins likely decreases upon evolution of aneuploid cells, but further studies will be required to characterize the observed changes.

## FOXM1 and E2F4 transcription factors contribute to improved proliferation in evolved polysomic cells

To get more insight into the molecular pathways responsible for the improved proliferation, we asked which transcription factors orchestrate the gene expression changes observed in polysomic cells after in vitro evolution. We focused on HCT116-derived polysomic cell lines, as there was a strong significant correlation between the evolution of polysomic cancer cell lines derived from HCT116 and AADEPT scores calculated from the TCGA and CPTAC data (Fig. 2D,E). We determined the most differentially regulated proteins after in vitro evolution in Htr5 and Hte5 (FC > 3/2 or FC < 2/3, FDR < 0.05) and specifically excluded proteins with evidence for corresponding, significant abundance changes in the evolution of the WT cell lines. The 43 identified proteins exhibited the pattern characterized by the reversion of

expression changes with the adaptation of our polysomic HCT116 cells (Fig. 4A). Ten of these proteins were upregulated after chromosome gain and downregulated following in vitro evolution. Among them, we again found factors involved in interferon response and integrin signaling, such as IRF9 (transcription factor that mediates innate immune response), ITGA3 (integrin subunit alpha 3 is a receptor involved in cell migration), and RBCK1 (E3 ubiquitin ligase, which regulates the inflammatory response) and other factors, strongly overlapping with the global meta-analysis (Fig. 4A). This suggests, together with the comparison with hallmark gene signature enrichment (Fig. EV2B), that down-regulation of the inflammatory response also contributes to improved proliferation of aneuploid cells.

Among the top proteins that were downregulated after chromosome gain, but upregulated upon adaptation, we found many cell cycle proteins, such as CDK1, CEP55, MKI67, CDC20, KIF20A, PRIM1, and UBE2T (Fig. 4A). These proteins are all essential for various aspects of cell cycle regulation and cell division and are frequently dysregulated in cancer (Fischer et al, 2016). Next, we identified an overlap between the top upregulated proteins after in vitro evolution and the factors with high AADEPT scores in the TCGA and CPTAC to determine the significant overrepresentation of consensus transcription factor target sets from ENCODE and ChEA (ENCODE Project Consortium, 2012; Lachmann et al, 2010; Xie et al, 2021, Data ref: Xie et al, 2021). This analysis revealed a significant enrichment of E2F4 and FOXM1 targets in evolved cancer cell lines and in aneuploid tumors (Fig. 4B–D). In addition, the expression of FOXM1 itself significantly positively correlates with aneuploidy in cancer (Fig. 4E). In fact, FOXM1 is among factors with the highest AADEPT score (Fig. 4F). Expression of the E2F4 targets and E2F4 also show high AADEPT score, although to a lesser degree than FOXM1 (Figs. 4F and EV4A–C). In contrast, the expression of the transcription factor MYC, another pro-proliferative TF, does not correlate with aneuploidy. We conclude that the targets of the transcription factors FOXM1 and E2F4 are pivotal in mediating the adaptation to aneuploidy.

We were particularly interested in the transcription factor FOXM1, which has been associated with aneuploidy previously (Carter et al, 2006; Macedo et al, 2018; Pan et al, 2023). Immunoblotting revealed that the FOXM1 expression did not increase, but its accumulation in the nucleus, where it is active as a transcription factor, was elevated, suggesting that the regulation of FOXM1 was altered in evolved clones (Appendix Fig. S8A,B). Indeed, the initially decreased levels of Aurora B kinase, a target of FOXM1, were increased in the trisomic cells after in vitro evolution (Appendix Fig. S8C). To directly test the involvement of FOXM1 in proliferation of aneuploid cells, we overexpressed the constitutively active truncated form of FOXM1 (FOXM1-dNdK) in the parental HCT116 and in Htr5 (p0; before evolution). Immunoblotting revealed a strong expression of FOXM1 and increased expression of its targets, such as Aurora B or CDK1 (Figs. 5A and EV4D–F). High abundance of FOXM1-dNdK resulted in an improved proliferation in Htr5 cells, but not in the parental HCT116 (Fig. 5B,C; Appendix Fig. S8D,E). Next, we performed a competition assay to test whether cells overexpressing FOXM1 will gain an advantage in a mixed population. Here, we mixed HCT116 or Htr5 cells expressing BFP and FOXM1 with cells expressing RFP and a control plasmid, and the fluorescence ratio was measured over 10 days of passaging (Fig. 5D). The unevolved polysomic cells

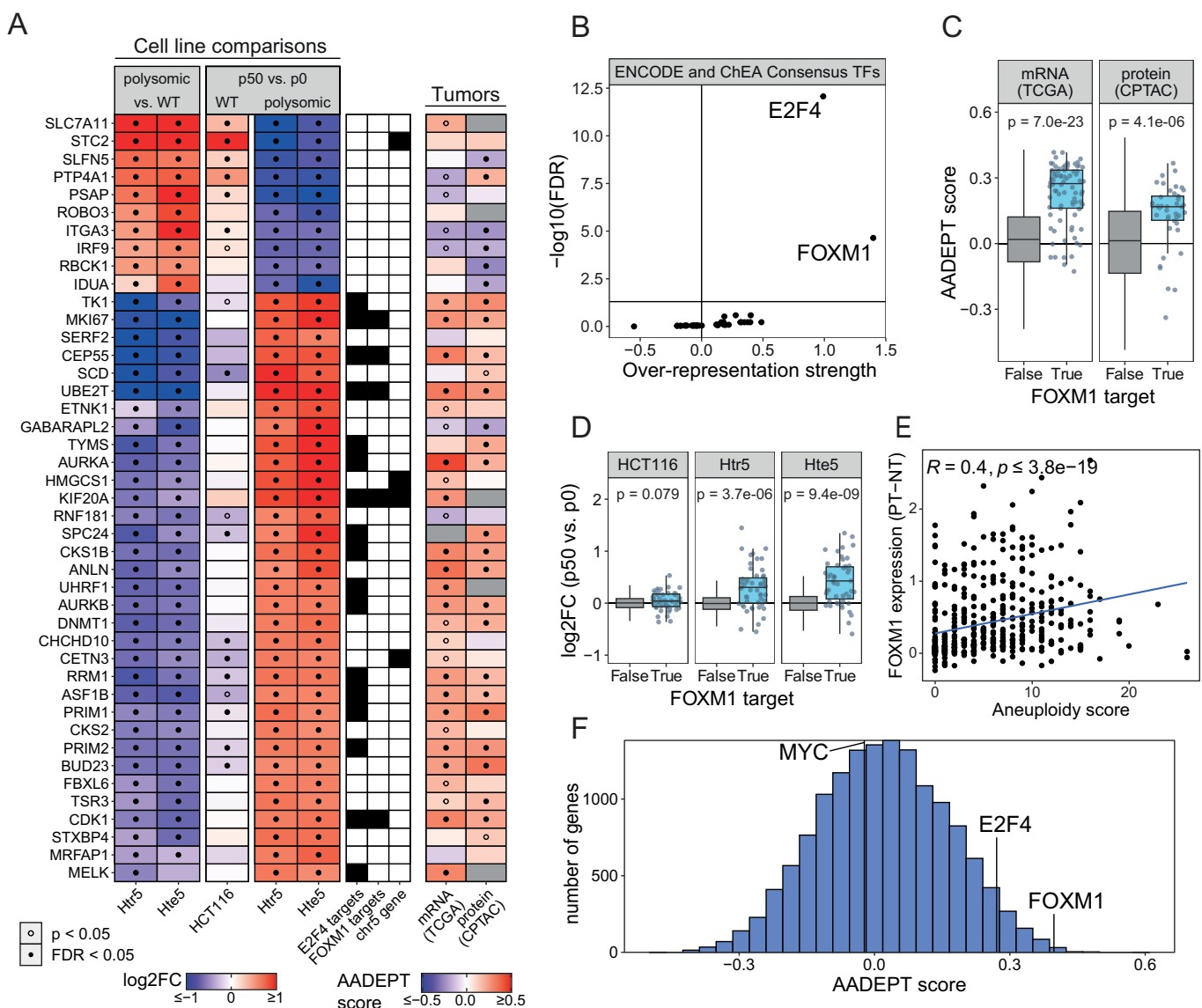

**Figure 4. FOXM1 and E2F4 targets are enriched among the proteins overexpressed in adapted cells with extra chromosomes and in aneuploid cancers.**

(A) Top 42 proteins with fold changes higher than 3/2 or lower than 2/3 before and after in vitro evolution of HCT116-derived polysomic model cell lines, and the corresponding AADEPT scores. Statistical significance for individual proteins was determined using empirical Bayes moderated Student's t tests for cell line comparisons, and using Spearman rank testing for tumor AADEPT scores (see "Methods"). (B) Overrepresentation of ENCODE and ChEA Consensus transcription factor (TF) targets among the differentially regulated genes in the evolved polysomic cell lines and aneuploid tumors. (C) Transcript- (TCGA) and protein-level (CPTAC) AADEPT scores of FOXM1 target genes ($n = 82$ and $n = 42$, respectively) were tested against all other proteins using Welch's t tests. (D) Protein abundance fold changes of all FOXM1 targets ($n = 56$) in evolved model cell lines, tested against all other proteins using Welch's t tests. (E) Spearman correlation coefficient between TCGA patient tumor aneuploidy score and FOXM1 gene expression relative to normal tissue (N: 467). (F) Histogram of transcript-level AADEPT scores derived from TCGA with highlighted scores of MYC, E2F4, and FOXM1.

overexpressing FOXM1 outgrew polysomic control cells, while FOXM1 overexpression rather reduced the proliferation of parental HCT116 (Fig. 5E). Similar results were obtained when the BFP and RFP staining was reverted (Fig. EV4G). To identify the mechanisms by which FOXM1 overexpression contributes to improved proliferation of the trisomic cells, we analyzed DNA damage response by γH2AX expression, as well as relative phosphorylation of CHK1 S345 and CHK2 T68, and micronuclei as a marker for genomic instability. Interestingly, FOXM1 overexpression resulted

in slightly elevated DNA damage response in diploid and trisomic cell lines, and slight reduction in micronuclei formation (Appendix Fig. S8F–H). In contrast, cell cycle analysis revealed that the FOXM1 overexpression reduced the fraction of cells in G1/G0 phase of the cell cycle (Figs. 5F and EV4H,I; Appendix Fig. S8H). The G1 phase and transition to the S phase is generally extended in cells with trisomic cells and thus overexpression of FOXM1 likely accelerates the G1/S-transition specifically in cells with extra chromosomes. In addition, we knocked out FOXM1 in diploid

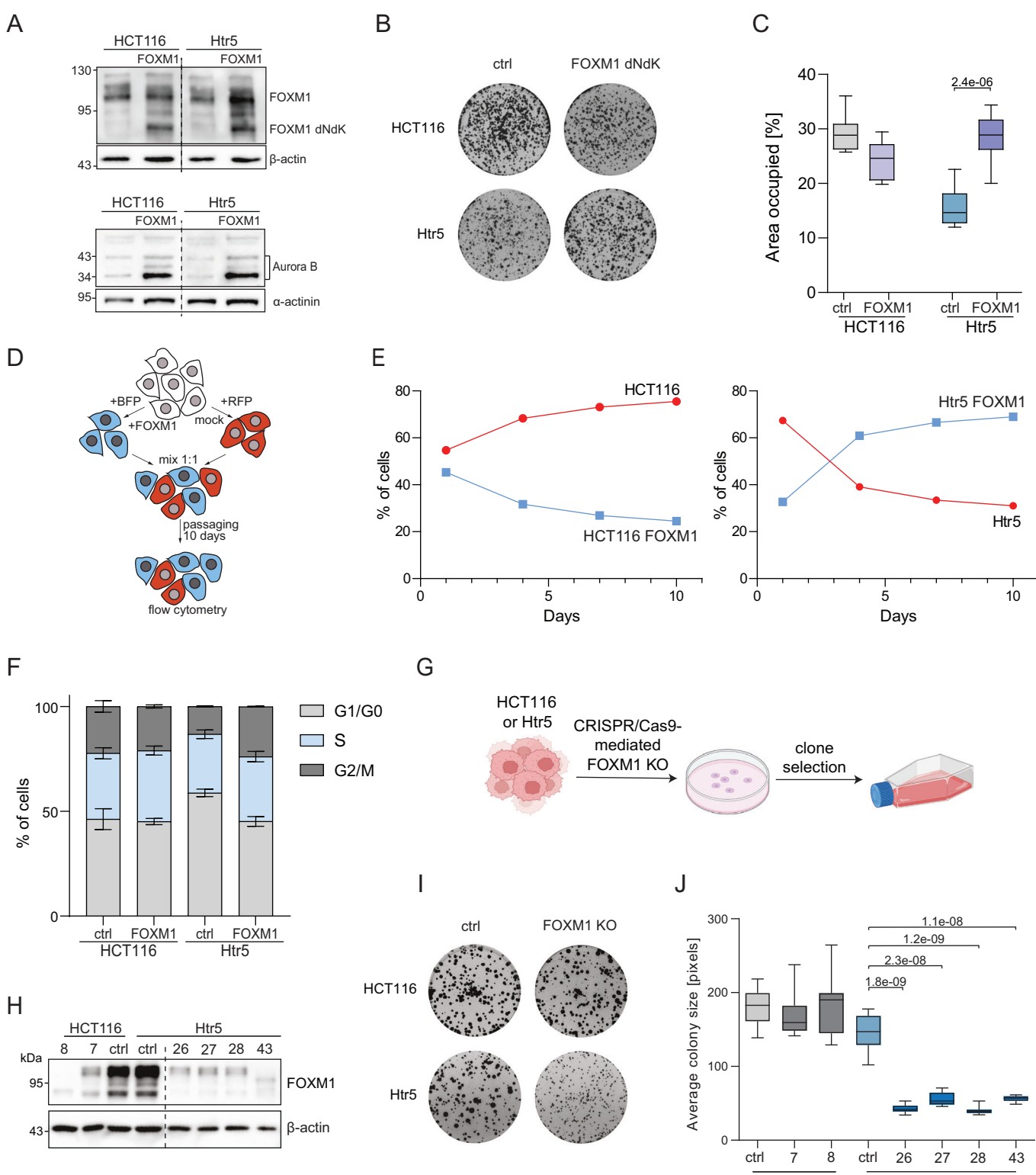

and trisomic cells and isolated individual clones following a scheme resembling the evolution by clone selection (Figs. 1A and 5G,H). Four analyzed Htr5 clones lacking FOXM1 showed strongly reduced proliferation after ~30 population doublings; no effect was observed in the parental cells lacking FOXM1 over the same

period (Fig. 5I,J). We propose that chromosome gain triggers a reduced expression of pro-proliferative FOXM1 targets and that increased expression of these proteins due to increased FOXM1 facilitates proliferation in trisomic cells, but not in cells with normal ploidy.

**Figure 5. FOXM1 expression changes alter the proliferation of cells with extra chromosomes.**

(A) Representative immunoblot of overexpression of FOXM1 and FOXM1 targets in the parental HCT116 and unevolved Htr5. (B) Representative images of clonogenic assay of HCT116 and Htr5 with and without overexpression of FOXM1. (C) Quantification of the percentage of area covered by cells in the clonogenic assay. Three biological replicates with three technical replicates each were analyzed. (D) Schematic depiction of the competition assay. (E) Quantification of the RFP- and BFP-positive cell fraction in competition assay during the incubation period. Data from representative experiment are shown. (F) Cell cycle analysis of HCT116 and Htr5 mock transfected (ctrl) or upon FOXM1 overexpression determined with flow cytometry using EdU incorporation and DAPI staining. Mean of three biological replicates with 100,000 cells tested for each. Error bars represent SEM. (G) Schematic depiction of the CRISPR/Cas9-mediated FOXM1 loss. (H) Representative immunoblotting of the FOXM1 expression in clones obtained after CRISPR/Cas9-mediated FOXM1 loss. (I) Representative images of clonogenic assay of HCT116 and Htr5 lacking FOXM1 compared to their respective control. (J) Quantification of the average colony size in the clonogenic assay. Three biological replicates with three technical replicates each were analyzed. Data information: Boxplots represent the 25th and 75th percentile with the median, whiskers show minimum and maximum values. P values were calculated using unpaired Student's t test. Source data are available online for this figure.

## Loss of chromosome arm 5q positively affects the proliferation of cell lines with trisomy 5

One of the striking observations following the evolution of cells with extra chromosome 5 in HCT116 cell line was the frequent subclonal loss of the 5q, while maintaining the 5p arm after in vitro evolution (Fig. 1C; Appendix Fig. S9A). Chromosome 5 is a frequent target of large copy number alterations in several malignancies, such as ovarian, gastric, and esophageal cancer, and malignant myeloid diseases (Lakhani et al, 2023). Moreover, analysis of chromosome arm-level events in the TCGA dataset clearly shows that loss of chromosome arm 5q and gain of 5p are among the most frequent events (Fig. 6A). We asked whether these specific changes in copy numbers of chromosome 5—loss of 5q with simultaneous retention of 5p—could affect the proliferation of the evolved cells. To this end, we used the recently developed technique ReDACT-TR (Restoring Disomy in Aneuploid cells using CRISPR Targeting with Telomere Replacement (Girish et al, 2023)), and transfected cells of a separate clone of Htr5p0 (before evolution) with a gRNA that cuts near the centromere of chromosome 5, simultaneously with a cassette encoding ~100 repeats of the human telomere seed sequence (Fig. 6B). Targeting the q-arm generated two independent cell lines with 5p trisomy, which were confirmed by FISH with probes specific for 5p and 5q arms and by shallow WGS (Fig. 6C,D). No cell lines with trisomy of the q-arm were generated, but we obtained two diploid cell lines, most likely due to the CRISPR/Cas9-induced loss of chromosome 5 (Papathanasiou et al, 2021; Tsuchida et al, 2023). Clonogenic assay showed that the trisomy of 5p accompanied by deletion of the 5q arm resulted in an improved proliferation in both clones Htr5p_1 and Htr5p_2 compared to the full trisomy of chromosome 5 (Fig. 6E). The ReDACT-TR also generated two clones with restored disomy of chromosome 5, Hdi5_1 and Hdi5_2 (Fig. 6C,D). The elimination of the supernumerary chromosome should theoretically lead to improved cellular proliferation, given that the original defect - chromosome gain - has been corrected. Intriguingly, the proliferation of both Hdi5 clones was significantly impaired, even when compared to the parental trisomy (Fig. 6E). Interestingly, Hdi_2 showed a partial loss of chromosome 8, where MYC is located; reduced MYC expression might therefore contribute to the growth defect. However, immunoblotting did not reveal any changes in MYC abundance in the analyzed clones (Appendix Fig. S9B).

To gain further insights into the molecular changes associated with the deletion of the arms of chromosome 5, we quantified the proteome of the ReDACT clones and compared them with the parental Htr5 (Fig. EV5A,B; Dataset EV8). We identified 204 proteins that were overabundant exclusively in both Htr5p clones; these genes significantly over-represent biological processes (Gene Ontology Consortium et al, 2023; Data ref: Liberzon et al, 2023) related to chromosome segregation and DNA replication, thus partly resembling the pathways that were activated after in vitro evolution (of note, the MCM2-7 helicase was not affected Fig. EV5C; Appendix Fig. S9C). Only two of these proteins, TRIP13 involved in homologous recombination, and NIPBL, required for cohesion loading, are located on the chromosome arm 5p, suggesting that also genome-wide changes contribute to the phenotype (Fig. EV5C,D). In contrast, no such overrepresentation was observed in the Hdi5 cell lines; instead, the overabundant proteins exclusive to Hdi5 were enriched for mitochondrial proteins (Fig. EV5B; Dataset EV8). The 158 and 150 proteins with decreased abundance specific for Htr5p and Hdi5 clones, respectively, showed in both cases an overrepresentation of processes related to cellular respiration and cytoplasmic translation. Here, we noticed that PAIP1, which stimulates translation, is located on chromosome 5p; its reduced expression levels can contribute to impaired proliferation. Based on this data, we propose that the effect of 5p gain and 5q loss is caused simultaneously by (1) 5p-encoded proteins that stimulate pro-proliferative processes when overexpressed upon 5p gain, and (2) proteins that cause negative effects when their previously increased abundance is reduced upon loss of the extra chromosome arm.

Next, we asked whether the proliferative advantages of chromosome 5 q-arm deletion are also reflected by reduced DNA damage or DNA damage signaling. We observed no reduction in DNA damage, as evaluated by γH2AX signal and RPA2 S33 phosphorylation (Appendix Fig. S9C–F). Phosphorylation of CHK2, an ATM-dependent marker for DNA damage, was also not altered (Appendix Fig. S9E). In contrast, the cell lines lacking an extra copy of chromosome 5q arm showed slightly reduced CHK1 phosphorylation by ATR at S345, as well as a milder increase in CHK1 phosphorylation upon aphidicolin treatment (Fig. 6F; Appendix Fig. S9C,G). This suggests that the loss of the 5q arm leads to a reduced CHK1-dependent signaling, which can provide a proliferative advantage in polysomic cells suffering from replication stress. How exactly the loss of chromosome 5q contributes to reduced CHK1-dependent signal, and whether it is sufficient to improve the proliferation is a subject for further investigation.

Strikingly, the examination of replication factors and CHK1-dependent signaling in both reverted disomic clones Hdi5_1 and Hdi5_2 uncovered an increase in CHK1 phosphorylation along with a low abundance of MCM7 (Fig. 6F; Appendix Fig. S9H).

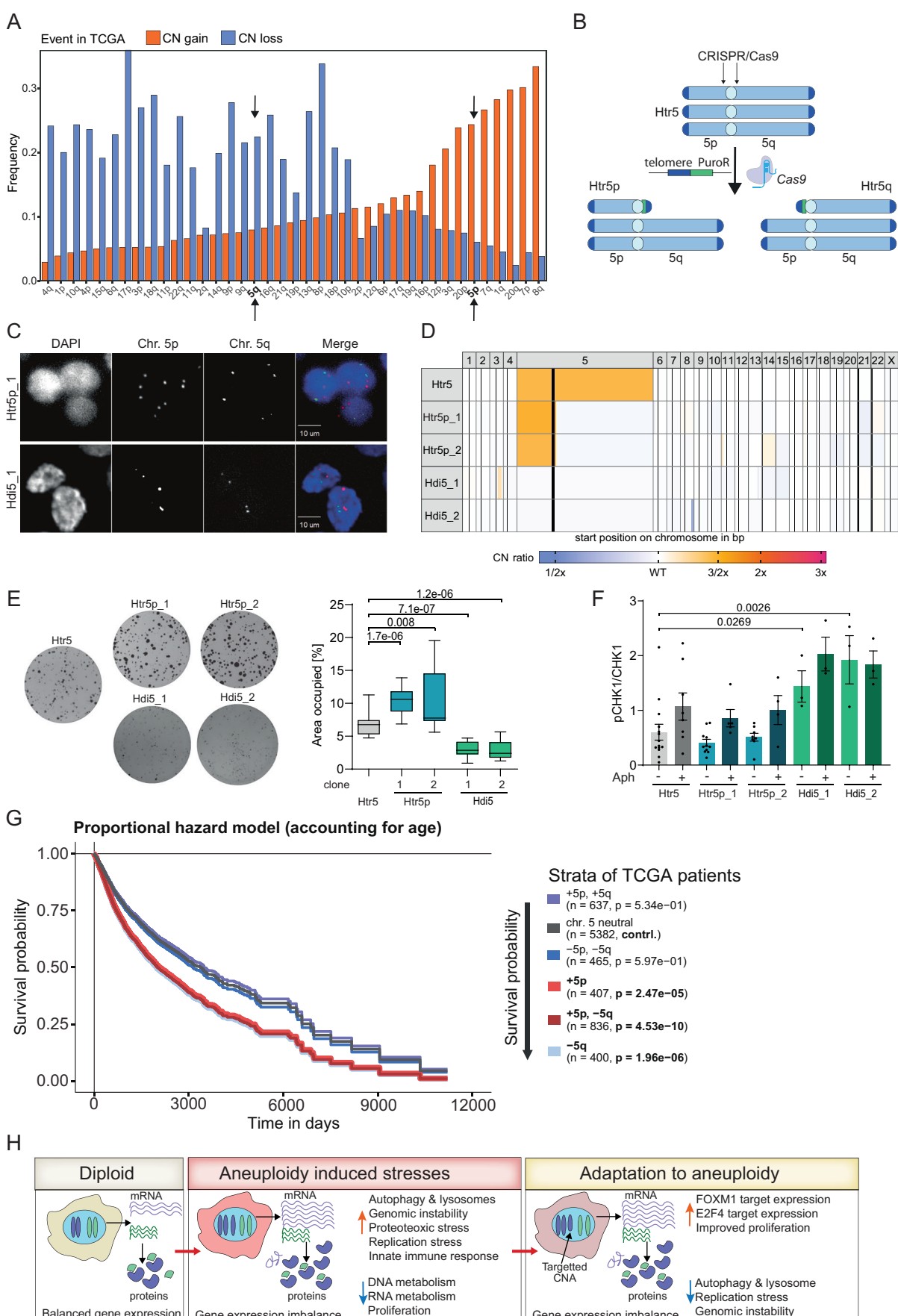

**Figure 6. Chromosome 5p arm shows a strong effect on cellular proliferation.**

(A) Frequency of chromosome arm gains and losses in cancer calculated from TCGA data. Black arrows indicate the frequent loss of 5q and frequent gain of 5p. (B) Schematic depiction of the chromosome engineering using ReDACT-TR. (C) Representative images of the fluorescence in situ hybridization of probes against 5p (red) and 5q (green) arms. (D) Chromosome-assigned shallow WGS of the parental Htr5 with Htr5p and Hdi5 cell lines. (E) Representative images of the clonogenic assay with quantification of the area occupied by colonies. For Htr5 and Htr5p cell lines, 16 data points were obtained from three biological replicates at 4–6 technical replicates each, for Hdi5 cell lines, 12 data points were obtained from two biological replicates at 4–6 technical experiments each. The box represents the 25th and 75th percentile with the median, whiskers show minimum and maximum values. (F) Mean pCHK1(S345)/CHK1 ratio determined via western Blot (at least three biological replicates; 1 to 3 technical replicates each), aphidicolin (Aph) treatment was performed for 24 h, 200 nM. Error bars represent SEM. (G) Survival curves of TCGA tumor patients stratified by the cooccurrence of arm-level chromosome 5 copy number gains and losses as fitted by a Cox proportional hazard model, showing the predicted survival curves of a patient of average age (58.7 years). *P* values are derived from Wald tests and represent the statistical significance of the hazard ratios relative to patients with copy number neutral chromosome 5. (H) Model of adaptation to aneuploidy in human cell lines and tumors. Data information: If not specified otherwise, *P* values were calculated using unpaired Student's *t* test. Source data are available online for this figure.

While it is possible that additional disadvantageous mutations may have accumulated in the two independently engineered disomic strains, we favor the hypothesis that restoring disomy imposes an additional burden on the previously aneuploid cells. Thus, the removal of a supernumerary chromosome from established trisomies does not necessarily translate to enhanced cellular proliferation, in line with our observation that none of the in vitro evolution resulted in a loss of the entire extra chromosome.

Our data show that arm-level chromosome 5 copy number aberrations have a strong effect on cell proliferation. Strikingly, similar effects can be observed when analyzing the patient survival of TCGA patients. Here, a full gain or loss of chromosome 5 does not increase patient hazard, while a selective gain of the p-arm and a loss of the q-arm exhibit significantly increased hazard compared to copy number neutral tumors (Fig. 6G), corroborating the different contribution of 5q and 5p chromosome arms to malignancy. Together, our analysis provides an example of specific chromosomal aneuploidy that corelates with proliferation and patient prognosis and provides first insights into possible molecular mechanisms.

## Discussion

The gain of even a single additional chromosome disrupts cellular homeostasis, exerting adverse effects on most cell types analyzed so far. Whether resulting from spontaneous or induced chromosome missegregation, or through chromosome transfer, the emergence of chromosomal gain in human somatic cells leads to diminished proliferation, pronounced alterations in gene expression, defects in protein homeostasis, compromised DNA replication, accumulation of DNA damage, and the initiation of sterile inflammatory responses, collectively constituting so-called aneuploidy-associated stresses (Chunduri and Storchova, 2019; Li and Zhu, 2022; Torres, 2023). These features are mirrored in aneuploid cancers, where replication stress and DNA damage, protein folding defects, and inflammation are frequently observed (Chen et al, 2022; Donnelly and Storchova, 2015; Macheret and Halazonetis, 2015). Intriguingly, cancer cells proliferate efficiently despite these challenges, suggesting the possibility that adaptive changes accumulate during tumorigenesis to counteract the negative consequences of aneuploidy.

To identify pathways contributing to the adaptation to the deleterious effects of chromosomal gains, we used isogenic human cell lines with an additional chromosome and subjected them to in vitro evolution experiments. We observed an improvement in proliferation within these cell lines, and remarkably, this enhancement did not depend on the loss of the extra chromosome (Fig. 1A–C). Instead, the cell lines retained either a part or the entirety of the additional chromosome while accumulating further chromosomal abnormalities. Similar results were recently observed in RPE1-derived complex aneuploid cells, suggesting that karyotype correction to diploidy is not a common mechanism of adaptation (Hintzen et al, 2024). The persistent presence of the supernumerary chromosome provided a unique opportunity for elucidating the physiological changes necessary for adapting to chromosomal gains.

To detect the factors underlying the enhanced proliferation, we compared the protein abundance in the evolved cell lines with the corresponding unevolved cells as well as the parental diploid wild type. This showed alterations in the expression of genes required for DNA replication, repair, and lysosomal functions. Strikingly, the comparison with data provided by TCGA and CPTAC databases showed that the adaptive changes observed in the evolved clones correlate well with the newly established AADEPT score, which quantifies the correlation of gene expression changes observed in primary tumors with the AS (Fig. 2B–D). This correlation was evident at both the gene level, as well as when assessing pathway changes (Fig. 2D,E). Our findings underscore the utility of our model systems in identifying factors relevant to the adaptive responses to aneuploidy in malignant aneuploid tumors.

Increased DNA damage and genomic instability are a major phenotype of cells with extra chromosomes (e.g., (Garribba et al, 2023; Passerini et al, 2016; Zerbib et al, 2024)). We found that the abundance of DNA repair and replication proteins increases after in vitro evolution, and the replication stress and genomic instability become reduced (Figs. 2 and 3E–H). The same trend is observed in tumor tissues, where a high abundance of proteins required for replication often correlates with poor prognosis (Fig. 2E) (Baak et al, 2009; Smith and Sheltzer, 2022). While the enrichment of cell cycle proteins could be a consequence of the improved growth after in vitro evolution rather than a cause, it may nevertheless allow insights into the underlying mechanisms. Strikingly, the consensus targets of transcription factors E2F4 and FOXM1 were strongly enriched in our in vitro evolution model, as well as in aneuploid cancers (Fig. 4). This suggests that to overcome the burdens of aneuploidy, increased activity of specific transcription factors promoting cell cycle progression is required.

While our comparison with cancer data shows a remarkable similarity among pathways identified in in vitro evolution of aneuploid cells and pathways, whose changes during tumorigenesis

correlate with aneuploidy (AADEPT), the situation in vivo will be certainly more complex. For example, aneuploidy is associated with the presence of cytosolic DNA which induces inflammatory pathways. We observed reduced expression of several innate immune response genes after in vitro evolution (Fig. 4A), supporting the notion that reduced immunogenicity is associated with aneuploidy and important for propagation of aneuploid cancers (Davoli et al, 2017).

We were particularly intrigued by the clear association of FOXM1 targets with the adaptation to aneuploidy. FOXM1 is upregulated during early cancer development and exerts multiple tumor-promoting activities through stimulation of cell cycle progression and suppression of senescence (Koo et al, 2012; Sadasivam and DeCaprio, 2013). Moreover, FOXM1 has been found to support proliferation of chromosomally unstable and aging aneuploid cells in both mouse and human (Macedo et al, 2018; Pan et al, 2023). We found that overexpression of FOXM1 in non-evolved polysomic cells improved their proliferation, while the knockdown of FOXM1 was detrimental for their growth. No such effects were observed in the parental diploid cells (Fig. 5B–J). FOXM1 is a cell cycle transcription regulator with a general positive effect on proliferation. However, other transcription factors that enhance cell cycle progression (e.g., E2F1 or YAP/TAZ coactivators) were not identified as contributing to the improved proliferation of aneuploid cells. Similarly, the expression of FOXM1 and its targets correlates with cancer aneuploidy in time (AADEPT score, Fig. 4E), while other pro-proliferative transcription factors do not show this association (e.g., MYC, Fig. 4E). We conclude that the association of FOXM1 activity with improved proliferation in polysomic cells is not only due to its general role in the cell cycle, but rather underscores the critical role of FOXM1 in adaptive responses to aneuploidy.

While most polysomic cell lines maintained the extra chromosome, trisomy and tetrasomy of chromosome 5 showed frequent loss of the 5q arm after in vitro evolution. Gain of 5p arm and loss of 5q arm are also enriched in tumors (Fig. 6A). By using the recently developed ReDACT-TR approach (Girish et al, 2023) for targeted chromosome arm engineering, we created unevolved trisomic cell lines lacking 5q and 5p arm, respectively. We showed the adverse effect of 5q trisomy, while the cells that maintained only an extra copy of 5p proliferated significantly better than the parental trisomy (Fig. 6E); this pattern is highly congruent with the survival of TCGA cancer patients that have tumors with respective karyotypes (Fig. 6G). We also obtained two diploid clones, most likely due to induced chromosome loss via CRISPR/Cas9-mediated targeting (Papathanasiou et al, 2021; Tsuchida et al, 2023). Despite the regained disomy, these cell lines showed a strong proliferation defect (Fig. 6E), in line with the fact that loss of the extra chromosome during in vitro evolution is not selected for (this work and (Hintzen et al, 2024)). Recently, a concept of aneuploidy addiction has been demonstrated in cancer cell lines where loss of additional chromosomes compromised cancer-like growth (Girish et al, 2023). This is likely due to the contribution of tumor suppressors and oncogenes to cancer aneuploidies, such as the *TERT* oncogene, encoded on chromosome 5p, and tumor suppressors like *MAP3K1*, encoded on chromosome 5q (Davoli et al, 2013; Shih et al, 2023). Our observation that the loss of an entire extra chromosome 5 from trisomic cell lines reduced proliferation in comparison to the full trisomy, while loss of 5q only and maintaining 5p led to generally improved proliferation, further accentuates the nuanced effects of aneuploidy on cancer cell proliferation.

In summary, we propose that aneuploidy initially impairs cellular fitness due to increased proteotoxic stress and expression of lysosomal proteins, as well as increased replication stress and low expression of replication factors, however, cells can adapt to these adverse effects through increased expression of FOXM1 and E2F4 targets or loss of anti-proliferative chromosome segments (Fig. 6H). One limitation of this study is the low amount of analyzed polysomic cell lines. However, the cross-comparison of changes identified in vitro with pathway regulation in cancer patients showed remarkable convergence. Our approach provides novel insights into molecular mechanisms of cellular adaptations to aneuploidy and may open a new route to potential cancer therapies.

# Methods

## Reagents and tools table

| Reagent/resource | Reference or source | Identifier or catalog number |
|---|---|---|
| **Experimental models** | | |
| HCT116 (p0) | Stingele et al, 2012 | |
| HCT116p50 | This study | |
| Hcs | This study | |
| Htr5(p0) | Stingele et al, 2012 | |
| Hte5p50 | This study | |
| Htr5cs | This study | |
| Hte5(p0) | Stingele et al, 2012 | |
| Hte5p50 | This study | |
| Hte5cs | This study | |
| RPE1 (p0) | Prof. Stephen Taylor (University of Manchester, UK) | |
| Rp50 | This study | |
| Rtr5 (p0) | Stingele et al, 2012 | |
| Rtr5p50 | This study | |
| Rtr21 (p0) | Stingele et al, 2012 | |
| Rtr21p50 | This study | |
| Htr5p_1 | This study | |
| Htr5p_2 | This study | |
| Hdi5_1 | This study | |
| Hdi5_2 | This study | |
| HEK 293T | Prof. Stefan Kins (RPTU Kaiserslautern-Landau) | |
| HCT116 FOXM1 KO cl7 | This study | |
| HCT116 FOXM1 KO cl8 | This study | |
| Htr5 FOXM1 KO cl26 | This study | |
| Htr5 FOXM1 KO cl27 | This study | |
| Htr5 FOXM1 KO cl28 | This study | |
| Htr5 FOXM1 KO cl43 | This study | |
| HCT116 FOXM1 OE | This study | |
| Htr5 FOXM1 OE | This study | |
| **Recombinant DNA** | | |
| pMMLV-EBFP-P2A-Puro | Addgene | #160227 |
| pMMLV-mRFP1-P2A-Puro | Addgene | #160228 |
| pHDM-Hgpm2 | Addgene | #164441 |

| Reagent/resource | Reference or source | Identifier or catalog number |
|---|---|---|
| pHDM-Tat1b | Addgene | #164442 |
| pHDM-VSV-G | Addgene | #164440 |
| pRC-CMV-Rev1b | Addgene | #164443 |
| pLVX–Tight-Puro–FoxM1-dNdK | Dr. Elsa Logarinho | |
| pX330-U6-Chimeric_BB-CBh-hSpCas9 | Addgene | #42230 |
| EF1a_Puro_Telo_v1 | Addgene | #195138 |
| **Antibodies** | | |
| Rabbit anti-LAMP1 IF;1:1000 | Abcam | ab278043 |
| Mouse anti-LAMP2 IF; 1:1000 | Abcam | ab25631 |
| Mouse anti-β-actin WB; 1:1000 | Sigma-Aldrich | A5441 |
| Mouse anti-α-actinin WB; 1:1000 | Santa Cruz | sc-17829 |
| Rabbit anti-MCM2 WB; 1:1000 | Abcam | ab4461 |
| Rabbit anti-MCM3 WB; 1:1000 | Cell Signaling | 4012 |
| Rabbit anti-MCM4 WB; 1:1000 | Cell Signaling | 12973 |
| Rabbit anti-MCM5 WB; 1:1000 | Biorbyt | orb128349 |
| Rabbit anti-MCM6 WB; 1:1000 | Biorbyt | orb48461 |
| Rabbit anti-MCM7 WB; 1:1000 | Cell Signaling | 3735 |
| Rabbit anti-Chk1 WB; 1:1000 | Abcam | ab32531 |
| Rabbit anti-pChk1 (S345) WB; 1:1000 Flow cytometry; 1:1000 | Cell Signaling | 2341 |
| Mouse anti-Chk2 WB; 1:1000 | Santa Cruz | sc-9064 |
| Rabbit anti-pChk2 (T68) WB; 1:1000 | Cell Signaling | 2197 |
| Mouse anti-RPA2 / RPA2 WB; 1:1000 | Abcam | ab2175 |
| Rabbit anti-pRPA2 (S33) WB; 1:1000 Flow cytometry; 1:500 | Bethyl | A300-246A |
| Rabbit anti-yH2A.X (S139) WB; 1:1000 IF; 1:1000 | Abcam | ab11174 |
| Rabbit anti-p21 WB; 1:1000 | Cell Signaling | 2947 |
| Mouse anti-p53 WB; 1:1000 | Santa Cruz | sc-124 |
| Rabbit anti-FOXM1 WB; 1:1000 IF; 1:100 | Cell Signaling | 5436 |
| Rabbit anti-Aurora B WB; 1:1000 | Abcam | ab2254 |
| Mouse anti-Cdk1 WB; 1:1000 | Santa Cruz | sc-54 |
| Mouse IgG HRP-conjugate WB; 1:5000 | Bio-techne | HAF007 |
| Rabbit IgG HRP-conjugate WB; 1:5000 | Bio-techne | HAF008 |

| Reagent/resource | Reference or source | Identifier or catalog number |
|---|---|---|
| Alexa Fluor 594-Goat anti-rabbit Flow cytometry; 1:500 | Jackson ImmunoResearch | 111-585-003 |
| Alexa Fuor 647-Donkey anti-rabbit Flow cytometry; 1:300 | Jackson ImmunoResearch | 711-605-152 |
| Alexa Fluor 594-Goat anti-mouse Flow cytometry; 1:500 | Jackson ImmunoResearch | 115-585-003 |
| Mouse anti-BrdU (IdU) DNA combing; 1:10 | BD Pharmigen | 347580 |
| Rat anti-BrdU (CldU) DNA combing; 1:10 | Abcam | ab6326 |
| Rabbit anti-ssDNA DNA combing; 1:5 | IBL | 18731 |
| Goat anti-mouse Cy3 / 647 DNA combing; 1:25 | Jackson Immunoresearch | 115-165-003 |
| Goat anti-rat Cy5 / 594 DNA combing; 1:25 | Jackson Immunoresearch | 112-175-143 |
| Goat anti-rabbit BV480 DNA combing; 1:25 | Jackson Immunoresearch | 111-685-144 |
| Mouse anti-53BP1 (E-10) WB 1:500 | Santa Cruz | sc-515841 |
| Mouse anti-pMPM-2 Flow cytometry; 1:500 | Sigma-Aldrich | 05-368 |
| Mouse anti-C-Myc (9E10) WB 1:1000 | Santa Cruz | sc-40 |
| **Oligonucleotides and other sequence-based reagents** | | |
| Chr. 5p fw (HCN1) | This study | CACCGTGTGCTTCAAGGTGGACGG |
| Chr. 5p rev (HCN1) | This study | AAACCCGTCCACCTTGAAGCACAC |
| Chr. 5q fw (PELO) | This study | CACCGGGTGATTCCAGTTCTGAAG |
| Chr. 5q rev (PELO) | This study | AAACCTTCAGAACTGGAATCACCC |
| Control siRNA-A | Santa Cruz | sc-37007 |
| 53BP1 siRNA (h) | Santa Cruz | sc-37455 |
| CRISPR739962_SGM | Thermo Fisher Scientific | A35533 |
| CRISPR739967_SGM | Thermo Fisher Scientific | A35533 |
| **Chemicals, enzymes, and other reagents** | | |
| LysoTrackerTM Red DND-99 | Invitrogen | L7528 |
| eBioscience™ Calcein Violett 450 | Invitrogen | 65-0854-39 |
| DDAO-Galactoside | Invitrogen | D6488 |
| Colchicine | Sigma-Aldrich | C9754 |
| Mounting Medium with DAPI | Vectashield | VECH-1200 |
| EdU | Sigma-Aldrich | 900584 |
| Fix/Perm solution | Thermo Fisher Scientific | 00-5123-43 |
| Perm wash | Thermo Fisher Scientific | 00-8333-56 |
| Eterneon-Red | Baseclick GmbH | BCFA-201-1 |
| DAPI | Carl Roth | 6335.1 |
| RNase | Thermo Fisher Scientific | 10334860 |
| Doxorubicin-hydrochloride | Sigma-Aldrich | D1515 |
| Protease-Inhibitor-Cocktail | Roche Diagnostics GmbH | 11836170001 |

| Reagent/resource | Reference or source | Identifier or catalog number |
|---|---|---|
| PhosSTOP | Roche Diagnostics GmbH | 4906845001 |
| Precision Plus Protein All Blue Standard | BioRad | #1610373 |
| Color Prestained Protein Standard, Broad Range | New England Biolabs | P7719L |
| Aphidicolin | Sigma-Aldrich | 178273 |
| Lactacystin | Sigma-Aldrich | 426100 |
| SytoxGreen | Thermo Fisher Scientific | S7020 |
| 5-chloro-2'-deoxyuridine (CldU) | Sigma-Aldrich | C6891 |
| 5-iodo-2'deoxyuridine (IdU) | Sigma-Aldrich | 57830 |
| Lipofectamine™ 2000 | Invitrogen | 11668027 |
| Polybrene | Sigma-Aldrich | TR-1003 |
| Puromycin-Dihydrochloride | Gibco | A1113803 |
| BbsI-HF | New England Biolabs | #R0539 |
| KpnI | New England Biolabs | #R3142S |
| BstZ17I | New England Biolabs | #R3594S |
| XL Cri-Du-Chat Deletion Probe | Metasystems | D-5417-050-OG |
| Proteomics-grade trypsin | Sigma-Aldrich | T6567 |
| PEI-transfection reagent | Polysciences | 23966 |
| Lipofectamine™ CRISPRMAX™ Cas9 Transfection Reagent | Thermo Fisher Scientific | CMAX00008 |
| TrueCut™ Cas9 Protein v2 | Thermo Fisher Scientific | A36497 |
| **Software** | | |
| Fiji | Schindelin et al, 2012 | |
| FlowJo™ V10.8 Software | BD Life Sciences | |
| ImageJ (v. 1.52i) | https://github.com/imagej/ImageJ/releases/tag/v1.52i | |
| E-CRISP gRNA design tool | Heigwer et al, 2014 | |
| Slidebook6 | Intelligent Imaging Innovations | |
| Cell Profiler V4.2.1 | Stirling et al, 2021 | |
| R software (4.2.1) | R Core Team, 2022; https://www.R-project.org/ | |
| ggplot2 (3.4.4) | Wickham, 2016; https://CRAN.R-project.org/package=ggplot2 | |
| gghx4 (0.2.6) | https://github.com/teunbrand/ggh4x | |
| ggpubr (0.6.0) | https://CRAN.R-project.org/package=ggpubr | |
| cowplot (1.1.1) | https://CRAN.R-project.org/package=cowplot | |
| lspline (1.0) | https://CRAN.R-project.org/package=lspline | |
| car (3.0) | Fox and Weisberg, 2019; https://CRAN.R-project.org/package=car | |
| HMMcopy | https://bioconductor.org/packages/HMMcopy | https://doi.org/10.18129/B9.bioc.HMMcopy |
| DNAcopy (1.70) | https://bioconductor.org/packages/DNAcopy | https://doi.org/10.18129/B9.bioc.DNAcopy |
| biomaRt (2.52.0) | Durinck et al, 2009 | |

| Reagent/resource | Reference or source | Identifier or catalog number |
|---|---|---|
| limma (3.52.4) | Ritchie et al, 2015; https://www.bioconductor.org/packages/release/bioc/html/limma.html | https://doi.org/10.18129/B9.bioc.limma |
| vsn (3.64.0) | Huber et al, 2002; https://www.bioconductor.org/packages/release/bioc/html/vsn.html | https://doi.org/10.18129/B9.bioc.vsn |
| survival (3.3-1) | Therneau and Grambsch, 2000; https://CRAN.R-project.org/package=survival | |
| survminer (0.4.9) | https://CRAN.R-project.org/package=survminer | |
| vcf2maf (1.6.22) | Kandoth et al, 2018; https://github.com/mskcc/vcf2maf/releases/tag/v1.6.22 | https://doi.org/10.5281/zenodo.593251 |
| bcftools (1.20) | Danecek et al, 2021 | |
| samtools (1.20) | Danecek et al, 2021 | |
| htslib (1.20) | Danecek et al, 2021 | |
| DRAGEN Bio-IT Platform | Illumina | |
| BWA-MEM algorithm | Li and Durbin, 2009 | |
| Bowtie2 | Langmead and Salzberg, 2012 | |
| BamUtil | Jun et al, 2015 | |
| HMM Copy Utils repository | https://github.com/shahcompbio/hmmcopy_utils | |
| MaxQuant software (V1.6.3.3) | https://www.maxquant.org/ | |
| **Other** | | |
| Gibco™ DMEM, high glucose, GlutaMAX™ Supplement, pyruvate | Thermo Fisher Scientific | 31966021 |
| Gibco™ Opti-MEM™ I Serum-reduced medium | Thermo Fisher Scientific | 31985070 |
| Gibco™ Trypsin-EDTA | Thermo Fisher Scientific | 25200056 |
| CellTiter-Glo® | Promega | G7570 |
| Proteasome-Glo™ Chymotrypsin-Like Cell-Based Assay | Promega | G8660 |
| Bradford protein assay | Biorad | 5000006 |
| Clarity™ Western ECL substrate solution | Biorad | #1705061 |
| DNA easy Blood and Tissue Kit | Qiagen | 69506 |
| Pierce™ BCA Protein Assay Kit | Thermo Fisher Scientific | #23225 |
| TMT 6-plex Isobaric Mass Tagging Kit | Thermo Fisher Scientific | #90061 |
| TMT 10-plex Isobaric Mass Tagging Kit | Thermo Fisher Scientific | #90406 |
| Pierce High pH Reversed-Phase Peptide Fractionation Kit | Thermo Fisher Scientific | #84868 |
| GloMax Explorer | Promega | |
| SH800S Cell Sorter | Sony Biotechnology | |
| Attune™ NxT Flow Cytometer | Invitrogen | |
| Azure c300 system | Azure Biosystems | |

| Reagent/resource | Reference or source | Identifier or catalog number |
|---|---|---|
| Trans-Blot® Turbo™ | BioRad | |
| Centrifuge 5415 R | Eppendorf | |
| Rotina 420 R | Hettich | |
| AxioObserver Z1 ASI MS-2000 stage CSU-X1 spinning disk confocal head Laser Stack Cool-Snap HQ camera | Zeiss Applied Scientific Instrumentation Yokogawa Intelligent Imaging Innovations Roper Scientific | |
| HiSeq4000 | Illumina | |
| MoFlo Astrios cell sorter | Beckman Coulter | |
| NextSeq 2000 | Illumina | |
| EASY nano-LC 1200TM system Q Exactive HF Nanospray Flex Ion Source | Thermo Fisher Scientific Thermo Fisher Scientific Thermo Fisher Scientific | |
| Q Exactive HF | Thermo Fisher Scientific | |

## Cell culture and treatment

RPE1 hTERT (referred to as RPE1) and RPE1 hTERT H2B-GFP were a kind gift of Stephen Taylor (University of Manchester, UK). HCT116 H2B-GFP was generated by lipofection (FugeneHD, Roche) of HCT116 (ATCC No. CCL-247) with pBOS-H2B-GFP (BD Pharmingen) according to the protocols from the manufacturer. Trisomic and tetrasomic cell lines were generated by microcell-mediated chromosome transfer as described in (Stingele et al, 2012). All cell lines were cultured at 37 °C with 5% $CO_2$ atmosphere in Dulbecco's Modified Eagle Medium (DMEM) (Gibco™) containing 10% fetal bovine serum (FBS) (Gibco™) and 1% penicillin/streptomycin (PenStrep) (Gibco™).

## Microcell-mediated chromosome transfer

Mouse A9 donor cells containing a single human chromosome and tagged with a drug-resistant gene were treated with colchicine for over 48 h to induce micronucleation. The microcells with micronuclei containing individual chromosomes were isolated through centrifugation with cytochalasin B. The filtered micronuclei were then treated with phytohemagglutinin-P (PHA-P) to allow adherence to the recipient cells. Polyethylene glycol (PEG) was used to fuse the micronuclei to the recipient cells. Cells with additional chromosomes were selected by antibiotic selection respective to the drug-resistant gene. For further details, see (Kneissig et al, 2019; Stingele et al, 2012).

## Proliferation assay

RPE1 and HCT116-derived cells were seeded in triplicates into a white 96-well plate (RPE1: $1.5 \times 10^3$ cells/well; HCT116: $6 \times 10^3$ cells/well). The CellTiter-Glo® Luminescent Cell Viability Assay by Promega was used to assess the proliferation. Twenty-four hours intervals were chosen for measurements. CellTiter-Glo reagent was added to each well according to the manufacturer's instructions. The luminescence was measured using the Promega GloMax

Explorer plate reader. Values for each cell line were standardized to day 0 of the respective sample.

## Clonogenic assay

In total, 500 cells/well were seeded to six-well plates and incubated for 10 days at 37 °C in 5% $CO_2$ incubator. Cells were then washed with PBS, fixed, and stained using 20% methanol and 0.5% crystal violet solution for 20 min at room temperature. To remove the staining, cells were washed once using PBS and twice using water. After drying, the plates were imaged, and colonies were analyzed using Fiji, with the threshold values adjusted to the respective test (Schindelin et al, 2012).

## Senescence and viability assay

The proportion of dead and senescent cells was determined by flow cytometry of CV450-stained cells and the fluorescent SA-β-Gal substrate DDAO-Galactoside, respectively. The FlowJo™ V10.8 Software (BD Life Sciences) was utilized to analyze the obtained data.

## Metaphase spreads

At a confluency of 70–80%, cells were treated with 400 ng/ml colchicine for 6 h. Cells were collected by trypsinization and centrifuged at 1500 rpm for 10 min. Cell pellets were resuspended in 75 mM KCl and incubated for 10 min at 37 °C. Fixation was performed with 3:1 methanol/acetic acid. Fixed samples were dropped onto glass slides. Vectashield Mounting Medium with DAPI was used to visualize the DNA.

## Chromosome painting

Samples were prepared for metaphase spreads as described above and further processed according to the manufacturer's protocol. Briefly, the glass slides were immersed in 2×SSC for 2 min at room temperature and dehydrated in a series of ethanol (70%, 85%, 100%) for 2 min each. Probes were spotted on each sample (7.5 μl per probe) and sealed with a coverslip. Denaturation was performed on a hot plate at 75 °C for 2 min, and subsequently placed in a humid chamber at 37 °C overnight to allow hybridization. Coverslips were removed and the samples were incubated with 0.4× SSC for 2 min at 72 °C followed by 2× SSC, 0.05% Tween20 for 30 s at room temperature. Vectashield Mounting Medium with DAPI was used to visualize the DNA.

## Fluorescence in situ hybridization

At a confluency of 70–80%, cells were collected by trypsinization and centrifuged at 1500 rpm for 3 min. Cell pellets were resuspended in 75 mM KCl and incubated for 10 min at 37 °C. Fixation was performed with 3:1 methanol/acetic acid. The samples were prepared according to the manufacturer's protocol. Briefly, the cells were spotted on glass slides and immersed in 2×SSC for 2 min at room temperature. The slides were dehydrated in a series of ethanol (70%, 85%, 100%) for two min each. Probes were spotted on each sample (10 μl) and sealed with a coverslip. Denaturation was performed on a hot plate at 75 °C for 2 min, and subsequently

placed in a humid chamber at 37 °C overnight to allow hybridization. Coverslips were removed and the samples were incubated with 0.4× SSC for 2 min at 72 °C followed by 2× SSC, 0.05% Tween20 for 30 s at room temperature. Vectashield Mounting Medium with DAPI was used to visualize the DNA.

## Analysis of DNA replication by EdU incorporation

EdU (10 μM) was added to the medium for 30 min at 37 °C. After the incorporation, the medium containing EdU was discarded, and the cells were harvested via trypsinization (5 min) and collected into 15 ml falcon tubes. After centrifugation for 3 min at 1300 rpm (Rotina 420 R), the cell pellet was washed with PBS and centrifuged again (3 min, 1300 rpm, Rotina 420 R). The supernatant was discarded, and the cells were permeabilized for 15 min with Fix/Perm solution (Thermo Fisher Scientific) according to the manufacturer's instructions. After permeabilization, 1 ml PBS was added, and the samples were then centrifuged for 3 min at 1300 rpm (Rotina 420 R). The supernatant was discarded, and the cell pellets were resuspended in 100 μl Perm wash (Thermo Fisher Scientific). Subsequently, 500 μl Click-iT Reaction Mix (1 μM Eterneon-Red (baseclick GmbH), 6.6% (v/v) 1.5 M Tris (pH 8.8), 500 μM $CuSO_4$, 100 mM Ascorbic Acid in PBS) was added to the samples for 20 min at room temperature in the dark. After the incubation, the samples were washed three times with 500 μl Perm wash (centrifugation for 2 min at 1600 rpm (Eppendorf centrifuge 5415 R)). The samples for the respective tests were additionally incubated in 100 μl 1:500 pRPA2 (S33) or MPM-2 antibody and subsequently in 100 μl 1:500 Alexa Fluor 594-Goat anti-rabbit or Alexa Fluor 594-Goat anti-mouse, respectively, for 30 min in the dark, each. Finally, the DNA was stained by incubating the cells in 300–500 μl (volume depending on the pellet size) PBS containing 1 μg/ml DAPI and 10 μg/ml RNase. The cells were analyzed by flow cytometry.

## Flow cytometry analysis of EdU-incorporated cells

The flow cytometry was performed with an Attune™ NxT Flow Cytometer (Invitrogen) following the manufacturer's instructions. DAPI and EdU-bound Eterneon-Red were excited by a 405 nm and a 638 nm laser, respectively. The emission from the excited DAPI was collected with a 440/50 filter. The light emitted by the excited Eterneon-Red was collected with a 670/14 filter. The FlowJo™ v10.8 Software (BD Life Sciences) was utilized to analyze the obtained data. First, the population of cells was gated by excluding particles based on the pulse area of the Side and Forward Scatter. Subsequently, the population of single cells was gated from the population of cells by pulse height versus pulse area of Forward Scatter. Finally, the pulse area of the Eterneon-Red and the DAPI emissions were plotted against one another to assign the single cell sub-populations based on their DAPI and Eterneon-Red emission displaying the DNA content and the EdU incorporation, respectively.

## Protein isolation

Cells were either seeded into 6-cm dishes ($1 \times 10^6$ cells per dish) or into 10-cm dishes ($2.2 \times 10^6$ cells per dish). As a positive control, cells were treated with 1 μM doxorubicin or other drugs (as specified in the figures) and incubated overnight to trigger DNA damage response. For harvesting, the cells were detached from the dish via trypsinization (5 min) and collected into 15 ml falcon tubes. After centrifugation for 3 min at 1300 rpm (Rotina 420 R), the cell pellet was washed with PBS and centrifuged again (3 min, 1300 rpm, Rotina 420 R). The supernatant was discarded, and the cell pellet was resuspended in 50–200 μl RIPA buffer (volume depending on the pellet size) supplemented with protease (Roche Diagnostics GmbH) and phosphatase inhibitors (Roche Diagnostics GmbH). The samples were sonicated on ice for 12 to 15 min and then centrifuged for 10 min at 13,200 rpm and 4 °C (Eppendorf centrifuge 5415 R) to remove the cell debris. The protein concentrations of the supernatants (whole-cell protein lysates) were determined in a Bradford protein assay. The protein concentrations were then adjusted to either 1 μg/μl or 10 μg/μl in 1× Laemmli buffer. After the samples were boiled for 5–10 min at 95 °C, they were stored at −20 °C until further usage.

## SDS-PAGE and immunoblotting

Depending on the well size, 10–15 μg of total protein was loaded on the gel per well. The Precision Plus Protein All Blue Standard (BioRad, Hercules, USA) or the Color Prestained Protein Standard, Broad Range (NEB) was used as a molecular weight marker. For stacking, gels were run at a constant voltage of 85–100 V for ~30 min. After that, the voltage was increased to 120–150 V for separation. Subsequently, either a wet transfer or a semidry transfer was used to transfer the proteins to a nitrocellulose blotting membrane. For the semidry transfer, Trans-Blot® Turbo™ (BioRad Laboratories, Hercules, USA) and Bjerrum–Schäfer–Nielsen buffer were used. The wet transfer was either done overnight at a constant current of 0.16 A or for 1 h at a constant voltage of 100 V (both at 4 °C). The transfer was verified by staining the membranes with Ponceau solution for 5 min. Nonspecific binding sites were blocked by incubating the membranes in 5% milk in TBS-T for 1 h at room temperature. After blocking, the incubation with the primary antibodies diluted in 5% milk or 5% BSA in TBS-T was done overnight at 4 °C. One day later, the membranes were washed three times with TBS-T for 5 min before the corresponding secondary antibodies (conjugated with HRP) diluted in 5% milk in TBS-T were added. This was incubated for at least 1 h at room temperature. Unbound secondary antibodies were removed by washing again three times for 5 min with TBS-T. Finally, the protein signals were detected by using the Clarity™ Western ECL substrate solution (BioRad) and the Azure c300 system (Azure Biosystems, Dublin, USA). For quantification, the signal intensities were measured using the software ImageJ (v. 1.52i). All used antibodies are listed in the reagents and tools table.

## Flow cytometry of CHK1 phosphorylation

Cells were seeded to 6 well plates and incubated (37 °C, 5% $CO_2$) for 24 h. Then, 200 nM aphidicolin was added to the cells and incubated again for 24 h. Cells were harvested using trypsin and washed using PBS. Fixation was performed using 70% EtOH for at least 30 min at 4 °C, followed by permeabilization for 15 min (0.25% Triton X-100 in PBS). Subsequently, the cells were incubated in PBS + 1% BSA for 30 min at RT. After blocking, the cells were incubated with primary antibody solution (1% BSA in PBS; pCHK1 primary AB 1:1000) at 4 °C overnight. Upon removal of the primary antibody solution, the cells were incubated with

secondary antibody solution (1% BSA in PBS; AlexaFluor647 anti-rabbit 1:300) for 30 min at room temperature. Finally, staining solution containing DAPI (0.1 μg/ml) and RNase A (0.01 mg/mL) in PBS was used to stain the DNA for 5 min. Fluorescence intensities were measured using Attune™ NxT Flow Cytometer (Invitrogen) and analysis was performed using FlowJo™ v10.8 Software (BD Life Sciences).

## Competition assay

HCT116 or Htr5 cells overexpressing FOXM1, and control cells were transduced with the pMMLV-EBFP-P2A-Puro or pMMLV-mRFP1-P2A-Puro, respectively, or vice versa. The cells were mixed in a 1:1 ratio and the proportion of cell fractions expressing RFP or BFP was quantified by flow cytometry after 1 day, 4 days, 7 days, and 10 days. For each measurement, 10,000 cells were harvested per mix and the cells were resuspended in PBS with 2 mM EDTA and the fluorescence ratio was measured with the SH800S Cell Sorter (Sony Biotechnology, Weybridge, UK). The rest of the cell mix was passaged with each measurement. The ratio between RFP- and BFP-positive cells was analyzed using FlowJo™ v10.8 Software (BD Life Sciences).

## Immunofluorescence staining

Cells were seeded in black, glass-bottom 96-well plates (Thermo Fisher Scientific™) and cultured in DMEM (with FBS and PenStrep) to a confluence of 70–80%. Fixation was performed for 12 min at RT with freshly prepared 3:1 methanol:acetic acid or with 3% formaldehyde. As permeabilization buffer, 0.5% Triton X-100 (Carl Roth) in PBS was used for a 5-min incubation at RT. Subsequently, the cells were blocked in 0.1% BSA for 30 min at RT. Primary antibody incubation was performed overnight at 4 °C. Subsequent washing was done 3× with PBS. The secondary antibody incubation was performed for 1 h at RT. The DNA was counterstained with SytoxGreen (167 nM) or 1.0 mg/ml DAPI both containing RNase A (0.01 mg/ml), and imaged as described below. Plates were stored in PBS with NaN₃.

## Lysotracker staining

$0.2 \times 10^6$ cells/well were seeded in 96-well plates. The following day, the cells were washed with PBS and replaced with medium containing 500 nM Lysotracker for 4 h at RT. Subsequently, the cells were washed with PBS and fixed in 3:1 methanol:acetic acid. After repeated rinsing with PBS-T, nuclei were counterstained with SytoxGreen (167 nM) containing RNase A (0.01 mg/ml). Plates were sealed with parafilm following the addition of PBS containing sodium azide (1 mM).

## DNA staining for microscopy

In total, $2.5 \times 10^4$ cells/well were seeded in 96-well plates. The positive control was treated with 0.2 μM aphidicolin and 0.73 mM caffeine for 24 h. Cells were fixed in 3:1 methanol:acetic acid. Subsequently, the cells were washed with PBS-T. The DNA was stained with SytoxGreen (167 nM) containing RNase A (0.01 mg/ml). The plates are sealed with parafilm after the addition of PBS containing sodium azide (1 mM).

## DNA combing

Overall, 200,000 cells were pulse-labeled for 30 min with 10 mM CldU 5-chloro-2'-deoxyuridine (CIdU) followed by 30 min of 10 mM CldU 5-iodo-2'deoxyuridine (IdU) according to the Genomic Vision Replication Combing Assay (RCA) Protocol (Paris, France). DNA was extracted using the Genomic Vision FiberPrep Kit for DNA extraction and Fibers were stretched following the manufacturer's instruction using the Genomic Vision RCA Protocol. A Genomic Vision FiberComb device with a stretching factor (SF) of 2 was used. Antibody staining was performed according to the RCA Protocol.

## Microscopy

Microscopy was performed using a semi-automated Zeiss AxioOb-server Z1 (Oberkochen, Germany) equipped with an ASI MS-2000 stage (Applied Scientific Instrumentation, Eugene, USA), a CSU-X1 spinning disk confocal head (Yokogawa) and a Laser Stack with selectable lasers (Intelligent Imaging Innovations, Denver, USA) and the Cool-Snap HQ camera (Roper Scientific). In all, 40× air or 63× oil objectives were used under the control of the software Slidebook6 (Intelligent Imaging Innovations, Denver, CO). Foci analysis was performed automatized using Cell Profiler V4.2.1 (Stirling et al, 2021).

## siRNA transfection

RPE1 and Rtr21 cells were transfected with 53BP1 siRNA (h) (Santa Cruz, sc-37455) or Control siRNA-A (Santa Cruz) by using Lipofectamine™ 2000 (Life Technologies, Thermo Scientific, CA, USA) according to the manufacturer's guidelines. The cells were seeded for clonogenic assay after 24 h, or the proteins were isolated 48 h after transfection for subsequent immunoblotting.

## Lentiviral infection

HEK 293T cells were transfected with the packaging plasmids pHDM-Hgpm2, pHDM-Tat1b, pHDM-VSV-G, and pRC-CMV-Rev1b (kind gift from Alejandro Balazs), and pLVX–Tight-Puro–FoxM1-dNdK (kind gift from Elsa Logarinho), pMMLV-EBFP-P2A-Puro, or pMMLV-mRFP1-P2A-Puro (both were a kind gift from Jason Sheltzer (Addgene plasmid # 160227; http://n2t.net/addgene:160227; RRI-D:Addgene_160227) (Addgene plasmid # 160228; http://n2t.net/addgene:160228 ; RRID:Addgene_160228)) by using Lipofectamine™ 2000 (Life Technologies, Thermo Scientific, CA, USA) according to the manufacturer's guidelines. The recipient cells were infected 48 h post-transfection of HEK 293 cells with the virus supernatant in the presence of 8 μg/ml polybrene according to the manufacturer's guidelines (Sigma-Aldrich, MO, USA). The cells transduced with the pLVX–Tight-Puro–FoxM1-dNdK were selected in 2 μg/ml puromycin (Gibco, Thermo Fisher Scientific, CA, USA).

## Induced chromosome arm-loss

We used the previously described REDACT-TR system (Girish et al, 2023). For target site prediction, we used the E-CRISP gRNA design tool (Heigwer et al, 2014), using reference genome hg38. The corresponding gRNA sequences were cloned into the pX330-U6-Chimeric_BB-CBh-hSpCas9 backbone (gift from Feng Zhang;

Addgene plasmid #42230) using BbsI (NEB) restriction sites. To introduce a selection marker via NHEJ, a fragment cleaved from EF1a_Puro_Telo_v1 (gift from Jason Sheltzer; Addgene plasmid #195138), which encodes a puromycin resistance cassette fused to a telomeric sequence, was used. A fragment produced by KpnI (NEB) and BstZ17l (NEB) digest was extracted from an agarose gel and co-transfected with the pX330 plasmid into the Htr5 cell line using the PEI-transfection method. Two days after transfection, 1 μg/ml puromycin was added for 48 h to select the transfected cells. Single colonies were picked two weeks later. The resulting cell lines were screened by FISH (Metasystems) and validated by sequencing.

## CRISPRMAX™ reagent Cas9 nuclease transfection

HCT116 and Htr5 cells were transfected with the TrueCut™ Cas9 Protein v2 (Thermo Fisher Scientific, A36497) and the TrueGuide ™Synthetic Guide RNAs CRISPR739962_SGM (Thermo Fisher Scientific, A3553) and CRISPR739967_SGM (Thermo Fisher Scientific, A3553) by using Lipofectamine™ CRISPRMAX™ Cas9 Transfection Reagent (Thermo Fisher Scientific, CMAX00008) according to the manufacturer's guidelines.

## Statistical analysis

The R software (version 4.2.1; R Core Team, 2022) was used to perform growth curve quantification as well as multi-omic data analysis of polysomic model cell lines and TCGA and CPTAC patient samples. For visualization, the R packages *ggplot2* (version 3.4.4; Wickham, 2016), *gghx4* (version 0.2.6), ggpubr (version 0.6.0) and *cowplot* (version 1.1.1) were used. For all other experimental data, analysis was done using GraphPad Prism 9.

## Growth curve quantification

The raw luminescence values in the results of the CellTiter-Glo® assays were normalized by subtracting the RLU measured at the initial time points. Then, a linear spline model was fit through the normalized luminescence values separately for each cell line and experimental run using the R package *lspline* (version 1.0). Residual bootstrapping of the fitted models was performed to create $R = 10,000$ bootstrap samples using the *Boot* function of the *car* (version 3.0; Fox and Weisberg, 2019) package. For each bootstrap sample, the area under the curve (AUC) of the linear spline was calculated by summation of the trapezoid areas between the horizontal axis and the line between two adjacent time points. The resulting AUCs were divided by the respective estimates of the wild-type cell lines at passage 0 and averaged over the different experimental runs. Finally, empirical 95% confidence intervals for the average AUC fold change were calculated using the 2.5% and 97.5% quantiles of the bootstrap samples.

## Genomic analysis

Genome-wide comparative genomic hybridization of cell lines (Dataset EV2) was carried out by IMGM laboratories (Martinsried, Germany). Agilent Human Genome CGH Microarrays (4 × 44K format) and SurePrint G3 Human CGH Microarray (4 × 180K format) were used in combination with a Two-Color based hybridization protocol with a commercially available reference gDNA. Signals on the microarrays were extracted using the Agilent DNA Microarray

Scanner. The Agilent Genomic Workbench 7.0 was used on the raw feature extraction data to apply probe filtering, log2 transformation, GC correction and re-centralization to obtain log2 copy number ratios relative to the reference gDNA. Further details are provided with the uploaded data files (see "Data availability").

Low-coverage whole-genome sequencing (lcWGS) of cell lines was performed at the NIG Integrative Genomics Core Unit (Göttingen, Germany) in two batches (see Dataset EV2). The genomic DNA was extracted in-house. Cells were collected with trypsin and centrifuged. The pellet was washed twice with PBS. DNA isolation was performed using the Qiagen DNA easy Blood and Tissue Kit (Hilden, Germany) following the manufacturer's instructions. gDNA from each cell line was used for library preparation and paired-end sequencing (151 bp read length) to an average depth of >1× on an Illumina HiSeq4000 machine. For data from batch 1, raw reads were aligned to the human reference genome (GRCh37/hg19) using the Illumina DRAGEN Bio-IT Platform (Host Software version 05.021.609.3.9.5 and Bio-IT Processor Version 0×04261818) to generate BAM files, and for batch 2, alignment was done using the BWA-MEM algorithm (version 0.7.17; Li and Durbin, 2009) and conversion to BAM format using Samtools htslib (version 1.9; Danecek et al, 2021).

Mini-bulk sequencing was performed at the ERIBA University Medical Center (Groningen, Netherlands) including sample preparation (see Dataset EV2). For sequencing, cells were suspended in media, washed, and pelleted. To generate nuclei, cells were resuspended in cell lysis and staining buffer (100 mM Tris-HCl pH 7.4, 154 mM NaCl, 1 mM $CaCl_2$, 500 μM $MgCl_2$, 0.2% BSA, 0.1% NP-40, 10 μg/mL Hoechst 33358, 2 μg/mL propidium iodide in ultra-pure water). Resulting cell nuclei were gated for G1 phase (as determined by Hoechst and propidium iodide staining) and sorted into wells of 96 wells plates containing freezing buffer on a MoFlo Astrios cell sorter (Beckman Coulter). Thirty nuclei of each were deposited per well. Plates containing nuclei and freezing buffer were stored at −80 °C until further processing. Automated library preparation was then performed as previously described (van den Bos et al, 2018). Libraries were sequenced on a NextSeq 2000 machine (Illumina; up to 77 cycles—dual-index, single end), and aligned to the human reference genome (GRCh38/hg38) using Bowtie2 (version 2.2.4 or 2.3.4.1; (Langmead and Salzberg, 2012)). Duplicate reads were marked with BamUtil (version 1.0.3; Jun, et al, 2015) or Samtools markdup (version 1.9; Danecek et al, 2021).

Deep whole-genome sequencing at a target 30× of cell lines was performed at the NIG Integrative Genomics Core Unit (Göttingen, Germany). Extraction and isolation were performed as described above. gDNA from each cell line was used for library preparation and paired-end sequencing (150 bp read length) to an effective average depth of >20× on an Illumina HiSeq4000 machine. Raw reads were aligned to the human reference genome (GRCh38/hg38) using the Illumina DRAGEN Bio-IT Platform (Host Software version 05.021.609.3.9.5 and Bio-IT Processor Version 0×04261818) to generate BAM files.

## Variant calling and annotation

Variant calling and annotation from deep sequencing data was performed in a miniconda 3 (version 24.5.0) environment with the following Bioconda packages: bcftools (version 1.20; Danecek et al, 2021), samtools (version 1.20; Danecek et al, 2021), htslib (version 1.20; Danecek et al, 2021), ucsc-liftover (version 447) and ensemble-vep

(version 112.0; McLaren et al, 2016). BAM files were converted to binary calling format (BCF) using the bcftools command *mpileup* on all HCT116-derived and all RPE1-derived data separately. Then, files were converted to variant calling format (VCF, version 4.2) using the *call* command. Variants were filtered based on quality by removing variants with a call quality and average mapping quality of less than 30 as well as fraction of 0 quality reads higher than 1%. Variants were filtered based on depth to only include calls with a minimum of four reads in every sample as well as four variant reads and a variant fraction of at least 5% in at least one sample. Furthermore, only calls with a supporting depth of less than twofold the median call depth were retained to exclude technical artifacts. Filtered variants were turned into annotated mutations with vcf2maf (version 1.6.22, Kandoth et al, 2018) running with Ensembl's variant effect predictor (McLaren et al, 2016). Furthermore, the effect of in-frame mutations was inferred by combining SIFT (Ng and Henikoff, 2003) and PolyPhen (Adzhubei et al, 2010) predictions, calling a mutation deleterious if either method produced such a classification. In the end, genes exclusively mutated in the cell lines were determined by filtering for genes without any variants in any sequenced ancestor cell line. We also sequenced diploid HCT116 after in vitro evolution to filter out genes with mutations that microsatellite instable cell lines acquire with prolonged passaging in a polysomy-independent way.

## Extraction and segmentation of genomic copy number ratios

For the aligned reads from lcWGS and minibulk sequencing, the tools from the HMM Copy Utils repository (https://github.com/shahcompbio/hmmcopy_utils) were used for extracting read counts from the BAM files in 50 kb (lcWGS) and 0.75 Mb (minibulk) bins as well as GC and mappability statistics from the reference genomes. The R package *HMMcopy* (version 1.38) was then used to estimate log2 copy number ratios controlling for mappablilty and GC content. Estimated copy number ratios from aCGH, lcWGS, and minibulk sequencing were then normalized by subtracting the values of the respective unevolved wild type. The normalized log2 copy number ratios were then grouped based on genomic location using circular binary segmentation as implemented in the R package *DNAcopy* (version 1.70) and average and median copy number ratios were calculated per segment.

## Estimation of total relative DNA

The amount of total relative autosomal DNA was estimated heuristically as

$$D = \sum_{i=1}^{N} round(R_i)L_i$$

where $N$ is the number of distinct segments of autosomal chromosomes identified by CBS, $L_i$ is the length of segment $i$ in bp and $R_i$ is the approximate difference in total copy number (which is being rounded up to the nearest digit), calculated as:

$$R_i = 2(2^{q_i} - 1)$$

Here $q_i$ is the normalized, log2 copy number ratio of segment $i$ as calculated above. Assuming that the reference copy number is approximately the same for each cell line of interest and the unevolved, wild type, it follows, that:

$$R_i = 2(2^{q_i} - 1) \approx 2\left(2^{\log_2\left(\frac{x_i}{r_i}\right)} - 1\right) = 2\left(\frac{x_i}{r_i} - 1\right) = 2\frac{x_i - r_i}{r_i}$$

where $x_i$ corresponds to the copy number of segments $i$ in the cell line of interest and $r_i$ to the corresponding value in the wild-type control. Now, reasoning that the parental cell lines are diploid, except for a few gains that do not change with the addition of extra chromosomes or in vitro evolution (Fig. 1C), it is assumed that:

$$R_i = 2\frac{x_i - r_i}{r_i} \approx \begin{cases} 0, when\ r_i \approx x_i \\ x_i - 2, when\ r_i \approx 2 \end{cases}$$

Thus, under the above assumptions, $R_i$ is indeed approximately the difference in segment copy number when being rounded to the nearest integer, and D corresponds to the total amount of autosomal DNA relative to wild type.

## Correlation analysis

Calculation and statistical testing of Pearson and Spearman correlation coefficients was performed using the *cor.test* function of the base R *stats* package with default parameters.

## Sample preparation for tandem mass tag (TMT)-based quantitative proteomics

Cultured cells ($1 \times 10^6$ cells) were harvested by trypsinization, centrifuged at 1200 rpm for 3 min, and washed 1× with 1× PBS. Pellets were snap-frozen in liquid nitrogen and stored at $-80$ °C. Sample preparation and labeling of the peptides with TMT isobaric mass tags was performed as per the manufacturer's instructions (Thermo Fisher Scientific). In brief, cells were lysed with provided lysis buffer and DNA was sheared by sonication in a water bath for 20 cycles (30 s on/30 s off). After centrifugation at $12,000 \times g$ for 10 min at 4 °C, protein concentration was determined using the bicinchoninic acid protein assay kit (Thermo Fisher Scientific) (PierceT™ BCA Protein Assay Kit, Thermo, #23225). In all, 50 µg of protein per condition was reduced with 5 mM Tris(2-carboxyethyl)phosphin-hydrochloride T (TCEP) for 1 h at 55 °C and subsequently alkylated with 10 mM iodacetamide for 30 min at 25 °C protected from light. By adding six volumes of acetone, proteins were allowed to precipitate overnight at $-20$ °C. Precipitated samples were resuspended in 100 mM TEAB and digested by incubation with 1.5 µg proteomics-grade trypsin (Sigma-Aldrich #T6567) at 37 °C overnight. 25 µg of peptides per condition were labeled with isobaric tags by incubation for 1 h at RT (TMT 6-plex and 10-plex, Thermo Fisher Scientific, TMTsix/tenplex™ Isobaric Mass Tagging Kit, #90061, #90406). For 10-plex and 6-plex analysis, cell line samples were individually tagged and for each cell line three biological replicates were labeled separately. The reaction was quenched by adding 5% hydroxylamine to the samples and incubating the samples for 15 min at RT. After combining all individually labeled samples per replicate, TMT-labeled peptides were fractionated into 8 fractions using the high pH reverse-phase peptide fractionation kit according to manufacturer's instructions (Thermo Fisher Scientific) (Pierce High pH Reversed-Phase Peptide Fractionation Kit, Thermo Fisher Scientific, #84868) and subsequently dried using vacuum centrifugation.

## Liquid chromatography-tandem mass spectrometry

TMT-labeled peptides were dissolved in 0.5% acetic acid supplemented with 1% of 2% acetonitrile/0.1% TFA pH 2 and analyzed by liquid chromatography-tandem mass spectrometry (LC-MS/MS) with a system of an EASY nano-LC 1200TM system (Thermo Fisher Scientific) and a Q Exactive HF (Thermo Fisher Scientific) connected through a Nanospray Flex Ion Source (Thermo Fisher Scientific). In all, 3 µl of each fraction was separated on a 40-cm heated reversed-phase HPLC column (75 µm inner diameter with PicoTip EmitterTM, Nex Objective) in-house packed with 1.9 µm C18 beads (ReproSil-Pur 120 C18-AQ, Dr. Maisch). Peptides were loaded in 5% buffer A (0.5% aequeous formic acid) and eluted with a 3 h-gradient (5-95% buffer B (80% acetonitrile, 0.5% formic acid) at a constant flow rate of 0.25 µl/ml). Mass spectra were obtained in a data-dependent mode. In brief, each full scan (mass range 375–1400 $m/z$, resolution of 60,000 at $m/z$ of 200, maximum injection time 80 milliseconds, ion target of 3E6) was followed by high-energy collision dissociation based fragmentation (HCD) of the 15 most abundant isotope patterns with a charge state between 2 and 7 (normalized collision energy of 32, an isolation window of 0.7 $m/z$, resolution of 30,000 at $m/z$ of 200, maximum injection time 100 milliseconds, AGC target value of 1E5, fixed first mass of 100 $m/z$ and dynamic exclusion set to 30 s).

Raw MS data were processed with the MaxQuant software (V1.6.3.3). All data was mapped to the human reference proteome database (UniProt: UP000005640) with peptides with an FDR < 1%.

The TMT mass spectrometry dataset three (see Dataset EV2) was acquired from the Proteomics Core Facility (University of Basel, Switzerland). Sample preparation and tandem mass tag labeling, mass spectrometric analysis as well as database searching and protein quantification were all performed as previously described (Vigano et al, 2018). The acquired TMT reporter ion intensities from peptides, that were identified in all replicates, were summed over corresponding protein accession numbers with missing values replaced by half of the lowest measured value and values over technical replicates being averaged.

## Protein and gene annotation

Mapping between UniProt protein identifiers and NCBI and Ensembl gene identifiers as well as chromosome, band and genomic location information was done using the *biomaRt* R package (version 2.52.0; Durinck et al, 2009).

## Differential protein abundance analysis

Identified protein groups were filtered to remove contaminants, reverse hits and proteins identified by site only. Protein groups with missing intensities were removed leaving sets of 6300, 5920, and 6140 robustly identified protein groups in total for TMT experiment one to three respectively (see Dataset EV2). The remaining TMT reporter intensities were log2-transformed, cleaned for batch effects using the R package *limma* (version 3.52.4; Ritchie et al, 2015) and normalized using variance stabilization normalization between samples as implemented in the *vsn* package (version 3.64.0; Huber et al, 2002).

Pair comparisons of protein intensities between cell lines were carried out using weighted linear regression models with observation-specific precision weights from the *voom* method and empirical Bayes moderated t-statistics from the *eBayes* method as implemented in *limma*. The resulting $p$ values were FDR-adjusted according to the Benjamini–Hochberg method. The analysis results from the three TMT experiments were merged using the proteins corresponding Ensemble gene identifiers.

## Estimation of dosage compensation

For a comparative view with the segmented, genomic, log2 copy number ratios, the log2 protein abundance fold changes were grouped based on the genomic location of their corresponding genes using circular binary segmentation as implemented in the R package *DNAcopy* (version 1.70). Chromosome arm-level dosage compensation was calculated as the difference between average genomic log2 copy number ratios and the average log2 abundance fold change of proteins originating from the respective arm, all relative to the unevolved parental cell line.

## Preparation of multi-omics patient datasets

Cancer patient-derived multi-omics and clinical data of TCGA was obtained as collected by Pan-Cancer Atlas project publications (Hoadley et al, 2018, Data ref: Hoadley et al, 2018). Aneuploidy scores and arm-level copy number gain and loss calls as well as whole-genome doubling status were taken as calculated by (Taylor et al, 2018) using the ABSOLUTE algorithm (Carter et al, 2012). Merged and batch-corrected pan-cancer RSEM RNAseq data was taken from (Hoadley et al, 2018) and processed as in the source paper, filtering for genes with valid expression values in at least 60% of samples and applying upper-quartile normalization and log2 transformation. To make the tumor data more comparable to the situation in model cell lines, patient with tumors that have undergone whole-genome doubling were excluded. Filtering further for tumors with both genomic and transcriptomic data as well as matching data of normal tissue, a set of 467 patients with 22 different cancer types was derived. Finally, the processed gene expression values of normal tissues were subtracted from the corresponding primary tumor data, resulting in a set of 14282 genes with normalized expression values (Appendix Fig. S5A; Dataset EV4).

Corresponding data from the CPTAC was obtained from the Pan Cancer Analysis resources (Li et al, 2023; Data ref: Li et al, 2023). Whole exome sequencing copy number variation data as processed by the Washington University team's pipeline was used to obtain segmented log2 copy number ratios as well as ploidy calls derived from ABSOLUTE. Here, we calculated the aneuploidy score as in (Taylor et al, 2018) by determining the number of autosomal chromosome arms with either a copy number gain or loss covering more than 75% of its length. As recommended in (Li et al, 2023), a chromosomal segment was determined to be gained or lost when the corresponding absolute, log2 copy number ratio exceeded 0.2. To filter out tumors that have undergone whole-genome doubling, we set a cut-off of 2.5 for the ABSOLUTE-derived ploidy. This resulted in aneuploidy scores with a comparable distribution as in TCGA patients (Appendix Fig. S5A,B). Pan-Cancer proteome data as processed by the University of Michigan team's pipeline, and then post-processed by the Baylor College of Medicine's pipeline was filtered to only

include patients with data for tumor and normal tissue. Consequently, the processed log2 relative protein abundance values of normal tissues were subtracted from the tumor data. Filtering further for patients with matching genomic and proteomic data and merging all 6 remaining cancer types, we derived a set of 347 patients with normalized protein values mapping to 8404 different genes (Appendix Fig. S5B; Dataset EV4).

## Definition of proteins scaling with chromosome arm gain in cancer

Log2 protein abundance values in CPTAC pan-cancer tumors, processed and normalized to matched normal tissue abundance as described above, were statistically compared between tumors with and without gain of the chromosome arm that the respective protein is encoded on, using Welch's tests. The tests were carried out one-sided to check whether the average difference in a protein's abundance was significantly lower than 3/2, which represents the level of change expected by a protein whose abundance scales with the change in DNA dosage caused by a single gain of the encoding chromosome in a diploid background. Proteins with an FDR-adjusted $P$ value (Benjamini–Hochberg) smaller than 0.05 were defined as not scaling with arm gain. Then, tests were carried out two-sided to identify proteins with significant, positive changes in abundance using the same significance level. Finally, proteins that meet this criterion and were not defined as non-scaling, were classified as scaling with arm gain.

## AADEPT score calculation

Aneuploidy-associated differential expression in tumorigenesis (AADEPT) scores were calculated by determining Spearman rank correlation coefficients between the aneuploidy score in the patient tumors and the mRNA gene expression (TCGA) and protein abundance (CPTAC) values contrasted with corresponding normal tissue samples. Subsequently, the correlation coefficients were tested for statistical significance using the *cor.test* function of the base R *stats* package.

## Enrichment analysis

For enrichment analysis, the previously developed 2D enrichment method developed by (Cox and Mann, 2012) was extended to be applicable to any number of dimensions. Accordingly, log2 protein abundance fold changes of cell line pair comparisons or protein AADEPT scores from CPTAC or TCGA tumors were first ranked and scaled from −1 (lowest) to 1 (highest). Then, for one-dimensional enrichment, the ranked values of proteins belonging to a set were tested for significant mean difference when compared to the respective background distribution of all other quantified proteins using univariate ANOVA as implemented in the *stats* package *anova* function. The observed difference in average protein ranks constitutes the enrichment score. Likewise, for multidimensional enrichment, protein fold change and/or AADEPT score ranks from multiple comparisons were taken as the input for multivariate ANOVA as implemented in the *stats* package's *manova* function to test for overall difference in means relative to the background. This way, gene set enrichment can be tested globally for a set of comparisons of arbitrary dimension. Lastly, FDR-adjustment for simultaneous testing of KEGG

pathways or hallmark gene sets was done using the Benjamini–Hochberg method.

## Hierarchal clustering

Heatmap clustering of proteins and protein sets was performed using agglomerative clustering as implemented in the R *stats* package function *hclust* with default parameters.

## Protein relevance score

The relevance of protein abundance change patterns in evolved polysomic model cell lines was accessed by calculating protein-wise relevance scores (Appendix Fig. S5C). These were derived as follows: First, the abundance fold changes of a protein in all pairs of cell line comparisons were ranked from −1 (most negative) to 1 (most positive). Then, a protein's fold change ranks were grouped based on the type of cell line comparison, either between an unevolved polysomic cell line and its parental wild type (group 1), or between an evolved cell line and its unevolved counterpart. The latter group was further divided into polysomic (group 2) and disomic control evolutions (group 3). Next, the sum of fold change ranks in group 1 and group 2 were added to one another. To account for changes in protein abundance that could be largely due to the effects of prolonged passaging, the maximum fold change rank in group 3 with equal sign to the sum from group 2 (or 0 if there was no such fold change) was subtracted from the sum prior to taking its absolute value. Only those proteins identified in all proteomics datasets were considered for scoring. Finally, empirical $P$ values for each protein's observed relevance score was calculated from 100,000 bootstrap samples of relevance scores derived from random permutation of proteins.

## Overrepresentation analysis

Hypergeometric tests were performed to assess the statistical significance of the overlap between the sets of differentially abundant proteins and the different molecular signature sets, using all identified proteins as a background. FDR-adjustment of $P$ values was done according to Benjamini–Hochberg within signature set categories. The strength of the overrepresentation was calculated as the ratio between the observed and expected overlap, namely: $s = \log_{10}\left(N\frac{q}{km}\right)$, with $q$ being the observed overlap, $k$ the number of differentially abundant proteins, $m$ the size of the signature set, and $N$ the number of measured proteins among the generated proteomic datasets ($N = 7310$).

When indicated, the redundancy between related sets was reduced by applying the affinity propagation algorithm with set size for scoring to cluster similar sets and extract representatives as implemented in the WebGestaltR package (version 0.4.6; Liao et al, 2019).

## PEI transfection

Cells were cultured in six-well plates to 70% confluency. 2 µg per plasmid/fragment were mixed with sterile-filtered 6 µl PEI-transfection reagent (1 mg/ml) in 700 µl serum-free medium and incubated for 30 min at room temperature. The mixture was then added drop-wise to the cells and incubated at 37 °C. 6 h after

transfection, the cells were washed to remove PEI. Antibiotic selection started 48 h after transfection.

## Patient survival analysis

The analysis of the overall survival of patients included in the TCGA was performed using Cox's proportional hazards model as implemented in the *survival* R package (version 3.3-1; Therneau and Grambsch, 2000) function *coxph*. All patients were stratified based on the cooccurrence of chromosome 5 arm-level gain or loss events, taking patients with copy number neutral chromosome 5 as the reference group. The model was fit using overall survival events and days as the outcome and the strata as well as patient age at initial diagnosis as covariates. Wald test-derived *P* values were taken as indicators of the statistical significance of each strata's hazard ratios. The predicted survival curves for each stratum and for a patient of average age (58.7 years) were visualized using the *survminer* R package (version 0.4.9).

## Data availability

All datasets produced in this study are available in the following databases: Genomic data from aCGH experiments: GEO GSE254936 (https://www.ncbi.nlm.nih.gov/geo/query/acc.cgi?acc=GSE254936); Aligned reads from lcWGS, minibulk and deep sequencing: SRA under BioProject PRJNA1073693 (https://www.ncbi.nlm.nih.gov/bioproject/PRJNA1073693/); In-house mass spectrometry proteomics data: PRIDE PXD049120 (http://www.ebi.ac.uk/pride/archive/projects/PXD049120), PXD049124 (http://www.ebi.ac.uk/pride/archive/projects/PXD049124) and PXD056432 (http://www.ebi.ac.uk/pride/archive/projects/PXD056432); Mass spectrometry data produced at the Proteomics Core Facility, University of Basel: Mass Spectrometry Interactive Virtual Environment (MassIVE) MSV000094028 (https://massive.ucsd.edu/ProteoSAFe/QueryMSV?id=MSV000094028).
Scripts used to analyze the data and generate the figures are available upon request.

The source data of this paper are collected in the following database record: biostudies:S-SCDT-10_1038-S44318-025-00372-w.

## Peer review information

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

## Acknowledgements

The initial observation of the adaptation and improved proliferation of polysomic cells after prolonged passaging was made by Silvia Stingele and Sara Schunter. The authors thank Carina Heinrich and other students of the Master Program Molecular Cell Biology at the RPTU Kaiserslautern-Landau for their help with the experiments. The next-generation sequencing was provided by the NGS Integrative Genomics Core Unit at the Institute of Pathology, University Medical Center Gottingen, and by the Research Sequencing Facility team, ERIBA, University Medical Centre Groningen. The proteomics data were obtained by Center for MS Analytics, RPTU Kaiserslautern-Landau and the Proteomics Core Facility, University of Basel. The authors thank Elsa Logarinho for providing the FOXM1 overexpressing construct, and Jason Sheltzer for providing the ReDACT-TR and competition assay plasmids (via Addgene). This project was funded by the FOR2800 Chromosome Instability: "Cross-talk of DNA replication stress and mitotic dysfunction" (DFG) to ZS and MR.

## Author contributions

**Jan-Eric Bökenkamp**: Resources; Data curation; Software; Formal analysis; Investigation; Visualization; Methodology; Writing—original draft; Writing—review and editing. **Kristina Keuper**: Supervision; Validation; Investigation; Methodology; Writing—original draft; Writing—review and editing. **Stefan Redel**: Validation; Investigation; Methodology; Writing—original draft; Writing—review and editing. **Karen Barthel**: Validation; Investigation; Methodology; Writing—original draft; Writing—review and editing. **Leah Johnson**: Validation; Investigation; Methodology; Writing—original draft; Writing—review and editing. **Amelie Becker**: Investigation; Methodology. **Angela Wieland**: Investigation; Methodology. **Markus Räschle**: Investigation; Methodology. **Zuzana Storchová**:

Conceptualization; Supervision; Funding acquisition; Methodology; Writing—original draft; Project administration; Writing—review and editing.

Source data underlying figure panels in this paper may have individual authorship assigned. Where available, figure panel/source data authorship is listed in the following database record: biostudies:S-SCDT-10_1038-S44318-025-00372-w.

## Funding

## Disclosure and competing interests statement
The authors declare no competing interests.

# Expanded View Figures

**Figure EV1.  Gene expression changes of individual proteins after in vitro evolution and during tumorigenesis.**

(**A**) Venn diagrams visualizing the overlap in significantly over- and underabundant proteins (FDR < 0.05) for cell line comparisons between polysomic cell lines and the parental WT (p0 vs. WT) and between polysomic cell lines before and after evolution (p50 vs. p0). (**B**) Volcano plots showing changes in relative protein abundance after in vitro evolution of polysomic and WT cell lines ($n = 3$ biological replicates per cell line). Proteins significantly over- (red) and underabundant (blue) in all adapted polysomic cell lines are highlighted next to the top 5 most over- and underabundant proteins per individual comparison. (**C**) Volcano plots of AADEPT scores for all measured genes on transcript- (TCGA) and protein-level (CPTAC) derived from 442 to 467 and 39 to 247 cancer patients respectively. Genes with top 10 lowest and highest AADEPT scores as well as genes of proteins in the model cell line overlap are highlighted.

▶

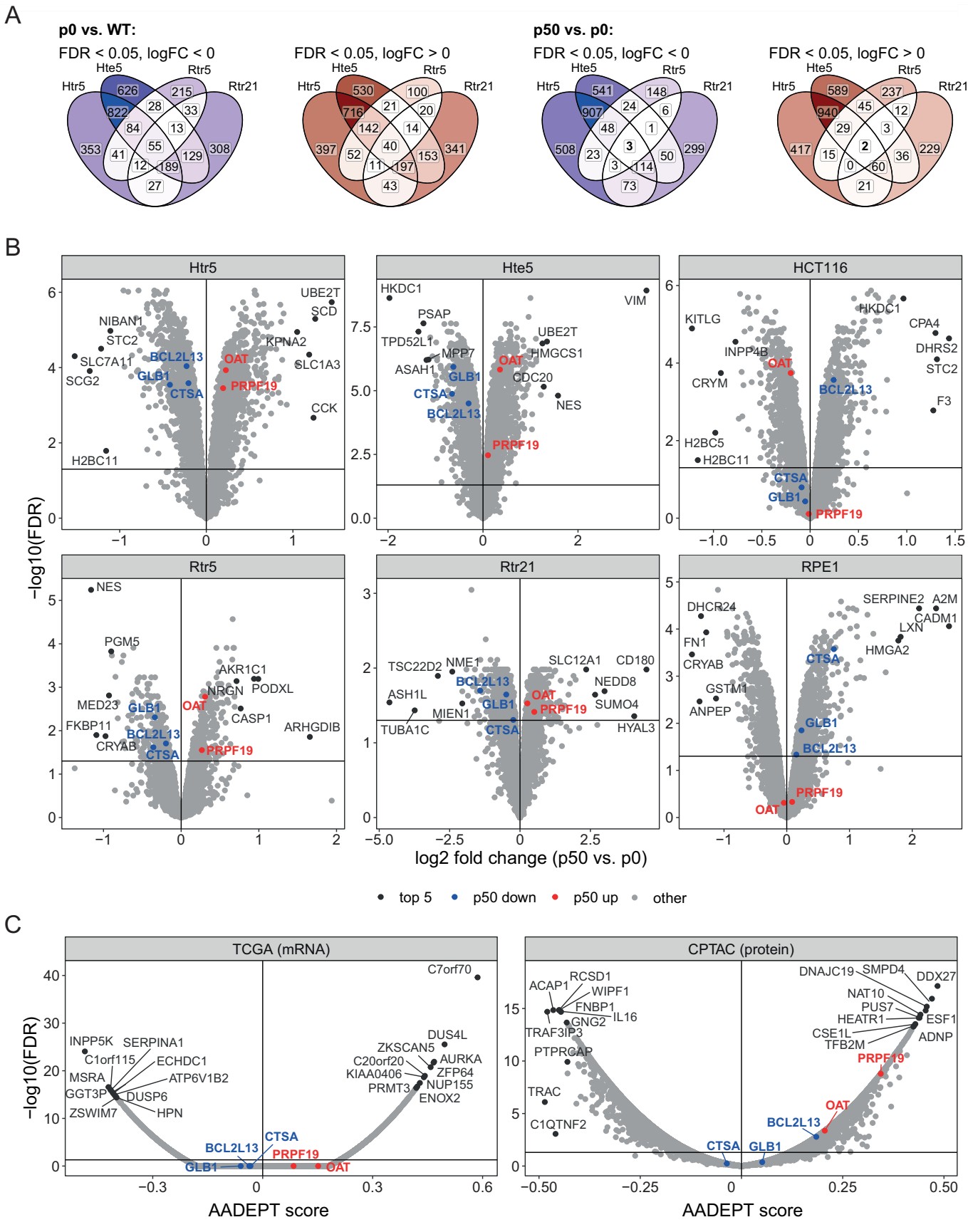

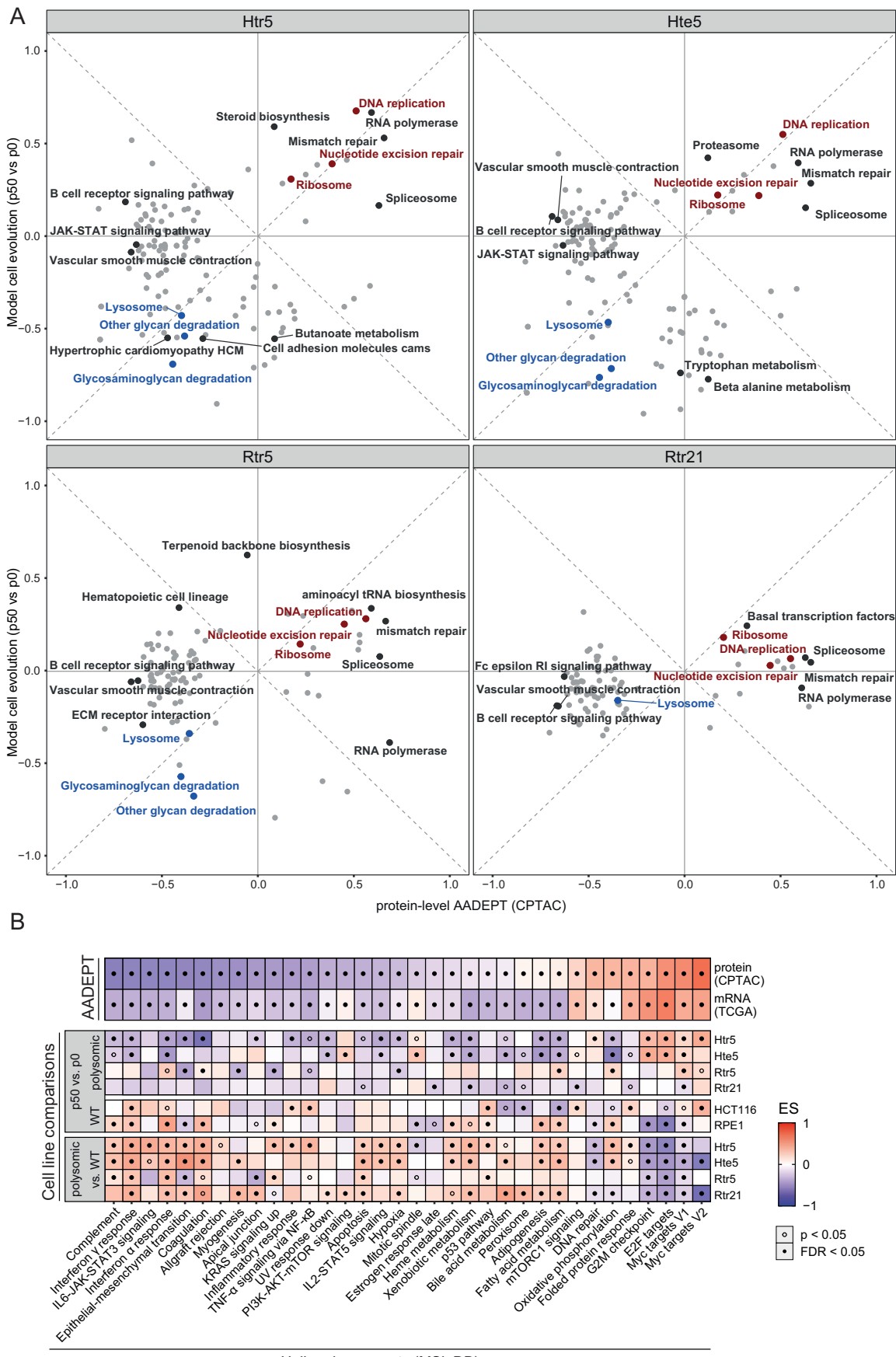

**Figure EV2. Gene expression changes of individual pathways after in vitro evolution and during tumorigenesis.**

(A) 2D enrichment of KEGG pathways comparing protein-level AADEPT scores (*x* axis) with fold changes in all polysomic cell lines after adaptation (*y* axis), showing enrichment scores for pathways with FDR < 0.05 and highlighting those with shared enrichment in adapted polysomic cell lines (blue—negative, red—positive) as well as those with top 3 enrichment scores in both direction of either axis and with more than 10 measured proteins. (B) Enrichment scores of Hallmark gene sets for AADEPT scores (above) and for protein abundance changes between cell line comparisons (below). Hallmark gene sets with FDR < 0.05 in a 2D enrichment analysis of transcript- and protein-level AADEPT scores are shown and sorted by degree of enrichment with protein-level AADEPT scores. The statistical significance of enrichment for individual gene sets was calculated using univariate ANOVA (see "Methods") and is indicated by the points within each tile.

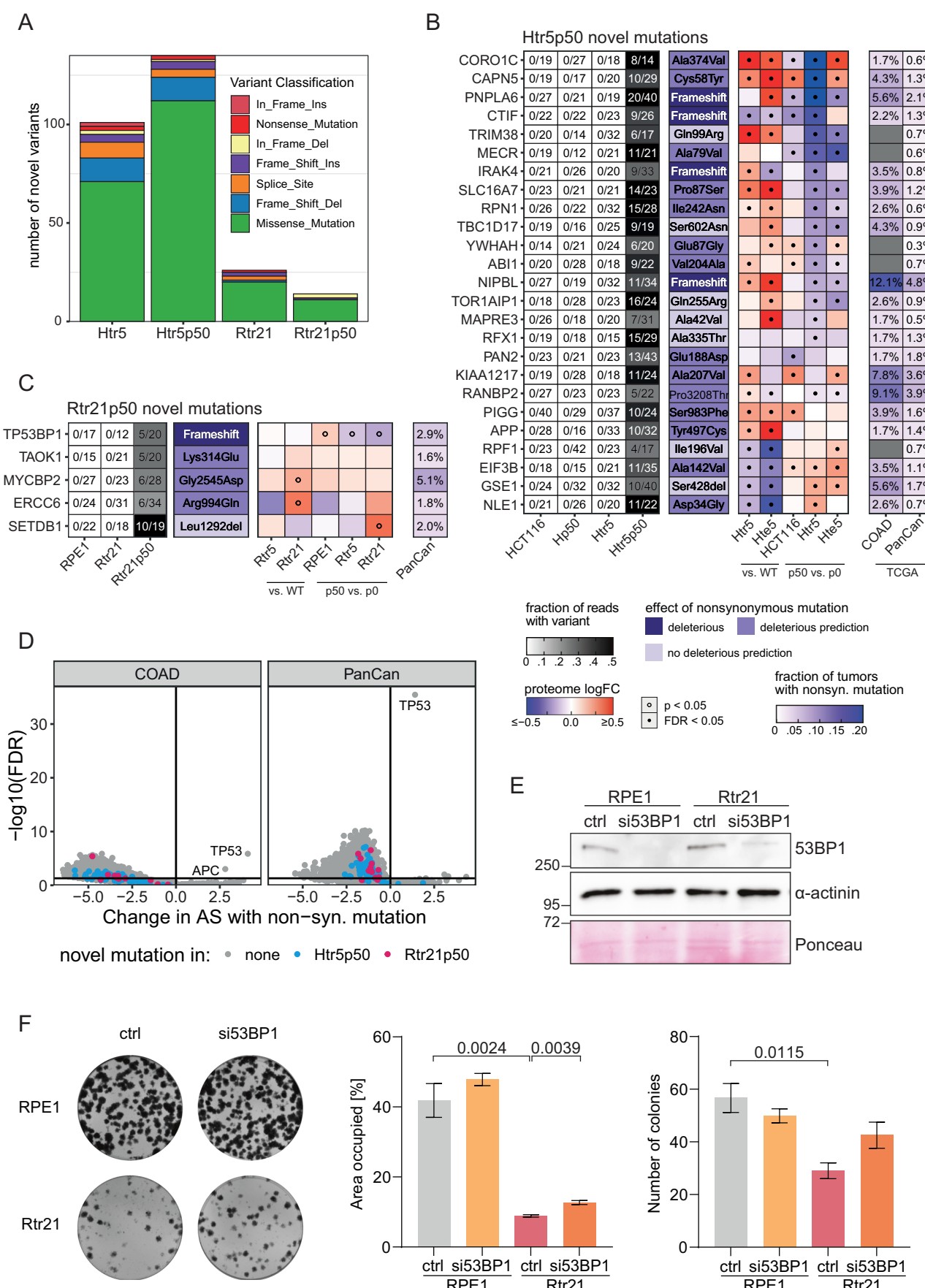

◀  **Figure EV3.   Novel point mutations after in vitro evolution.**

(**A**) Number of variants with classification per cell lines, excluding parental variants or those observed during evolution of diploid HCT116. (**B**) Fraction of variant reads per cell line for genes exclusively mutated in evolved Htr5 together with combined SIFT and PolyPhen variant effect prediction, changes in protein abundance, and prevalence of nonsynonymous mutations in TCGA colon (COAD) and pan-cancer tumors. Statistical significance for protein abundance changes between cell lines was determined using empirical Bayes moderated Student's *t* tests (see "Methods"). (**C**) Same as (**B**), but for genes exclusively mutated in evolved Rtr21. Legends, including numbered scale bars for heatmaps, are the same as in (**B**). (**D**) Volcano plot with size and significance of difference in average aneuploidy score between tumors with and without nonsynonymous mutations in a respective gene in colon and pan-cancer tumors. Only genes mutated in at least four samples were considered. *P* values were derived using Wilcoxon rank sum test of aneuploidy scores between the groups. (**E**) Representative immunoblot of 53BP1 in the RPE1 and unevolved Rtr21 48 h after transfection with either siRNA against 53BP1 or scrambled control siRNA. (**F**) Representative images of clonogenic assay of RPE1 and Rtr21 after transfection with either siRNA against 53BP1 or scrambled control siRNA. Quantification of the percentage of area covered by cells and the number of colonies in the clonogenic assay is shown ($n = 1$ for control cells, $n = 3$ for depleted cells, three technical replicates each), mean with SEM is shown. *P* values were calculated using unpaired Student's *t* test.

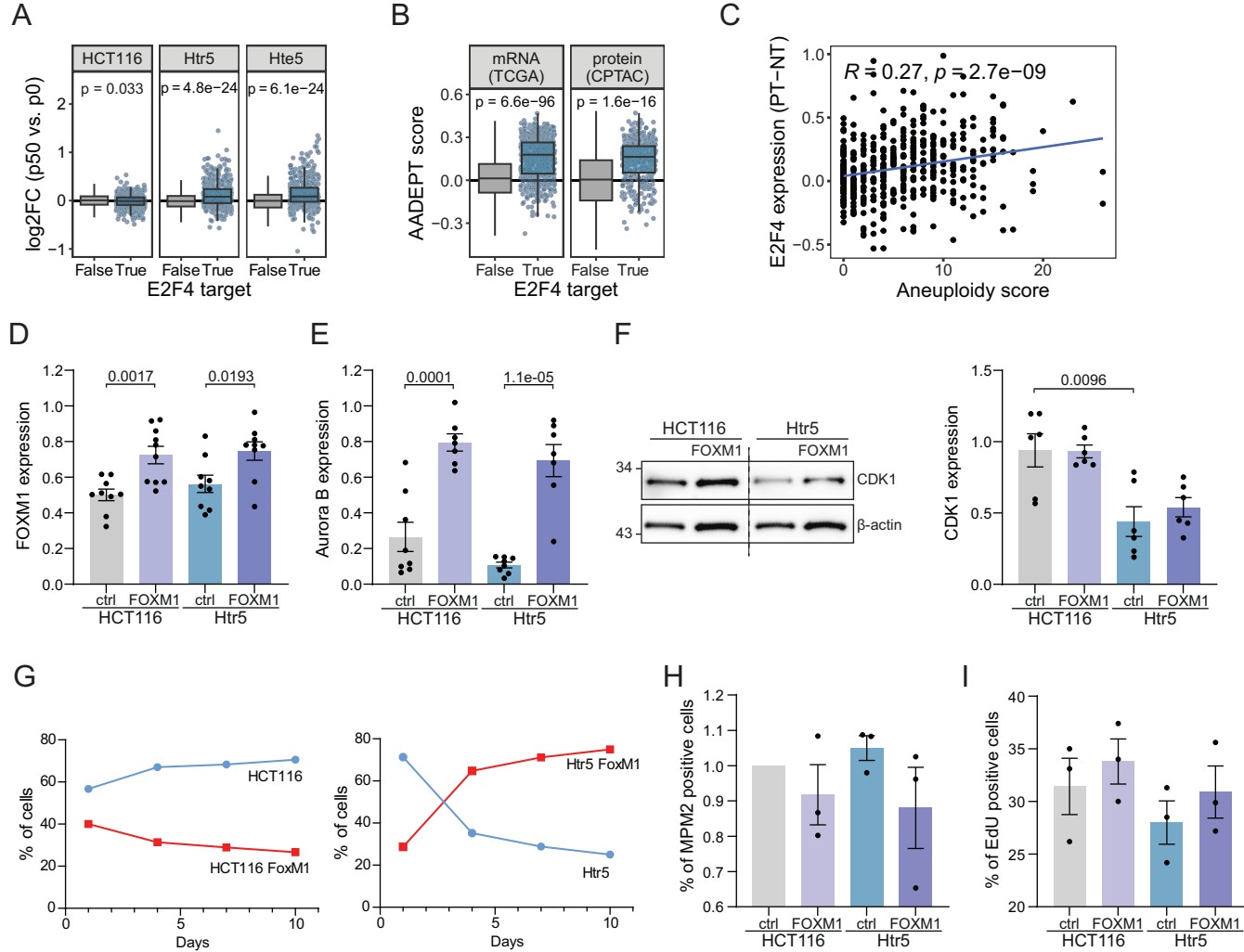

**Figure EV4. FOXM1 and E2F4 dependent changes after in vitro evolution and in cancer.**

(A) Protein abundance fold changes of all E2F4 targets ($n = 484$) in evolved model cell lines, tested against all other proteins using Welch's $t$ tests. (B) Transcript- (TCGA) and protein-level (CPTAC) AADEPT scores of E2F4 target genes ($n = 648$, $n = 406$ respectively) tested against all other proteins using Welch's $t$ tests. (C) Spearman correlation coefficient between TCGA patient ($N = 467$) tumor aneuploidy score and E2F4 gene expression relative to normal tissue. (D) Quantification of immunoblotting of FOXM1 overexpression cells of three biological replicates with 3 to 4 technical replicates each. (E) Quantification of immunoblotting of Aurora kinase B in FOXM1 overexpressing cells of three biological replicates with 2 to 3 technical replicates. (F) Representative immunoblot of CDK1 in FOXM1 overexpressing cells and quantification of three biological replicates with two technical replicates each. (G) Quantification of the RFP- and BFP-positive cell fraction in competition assay. Representative experiment. (H) The fraction of phospho-MPM-2 positive cells determined by flow cytometry. Mean of 3 biological replicates with 100,000 cells tested for each. (I) The fraction of EdU-positive cells determined by flow cytometry. Mean of 3 biological replicates with 100,000 cells tested for each. Data information: Boxplots represent the 25th and 75th percentile with the median. The whiskers extend from the upper and lower bound of the box to the largest and smallest value no further than 1.5 * IQR (inter-quartile range) from the respective bound (Tukey method). Bar plots show mean with SEM. If not specified otherwise, $p$ values were calculated using unpaired Student's $t$ test.

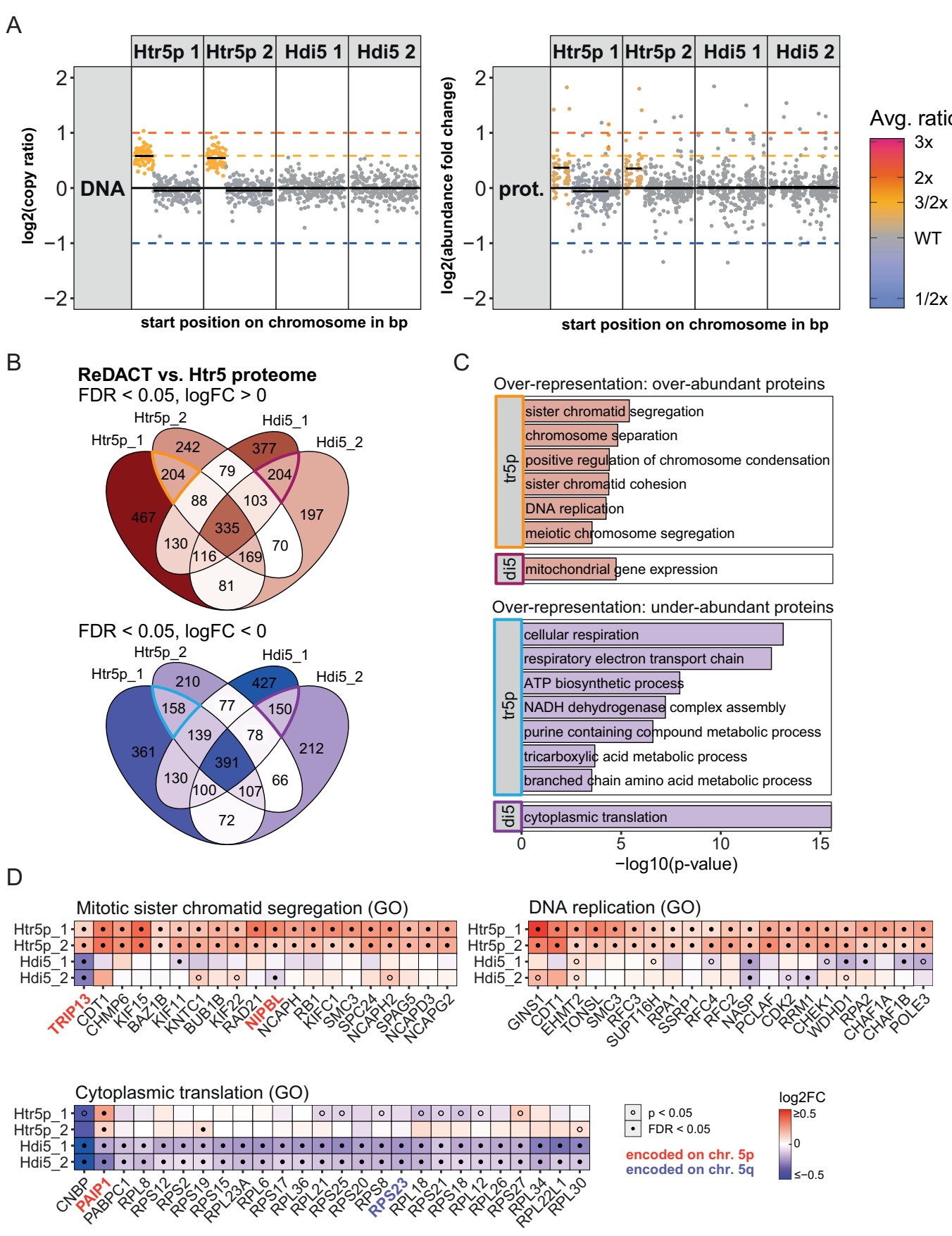

◄ **Figure EV5. Differential protein abundance in trisomy 5 ReDACT clones.**

(A) Average DNA copy number ratios of genomic bins (left) and abundance fold changes of proteins (right) relative to disomic HCT116. Only genomic bins and proteins located or encoded on chromosome 5 are shown for Htr5p and Hdi5 ReDACT clones in order of genomic start position. (B) Overlap of over- (top) and underabundant (bottom) proteins compared to parental Htr5 between Htr5p and Hdi5 clones. Proteins exclusive to Htr5p and Hdi5 are highlighted respectively. (C) Biological processes (Gene Ontology Consortium et al, 2023, Data ref: Liberzon et al, 2023) over-represented by the sets of proteins highlighted in (A). The redundancy of results was reduced using affinity propagation (see "Methods"). (D) Cell line-wise abundance fold changes of proteins which are involved in selected biological processes from the results of (B) and which are part of the respective set of exclusively deregulated proteins of (A). Statistical significance for protein abundance changes between cell lines was determined using empirical Bayes moderated Student's $t$ tests (see "Methods").

