## [Peer Review File · The EMBO Journal]

Proteogenomic analysis reveals adaptive strategies for alleviating the consequences of aneuploidy in cancer

Zuzana Storchova, Jan-Eric Bökenkamp, Kristina Keuper, Stefan Redel, Karen Barthel, Leah Johnson, Amelie Becker, Angela Wieland, and Markus Räschle

Corresponding author(s): Zuzana Storchova (storchova@bio.uni-kl.de)

Review Timeline:

Submission Date:	15th Feb 24
Editorial Decision:	27th May 24
Revision Received:	4th Oct 24
Editorial Decision:	9th Dec 24
Revision Received:	17th Jan 25
Accepted:	21st Jan 25

Editor: Hartmut Vodermaier

Transaction Report:

Dear Zuzana,

Thank you again for submitting your proteogenomic analysis of cellular adaptation to aneuploidy for our consideration, and apologies for the delay in getting back to you with the outcome of its evaluation. We sent the work to three expert referees, who have now returned the reports copied below. As you will see, all referees appreciate the potential interest of your results, but only referee 3 is at this point already strongly supportive of publication. Referees 1 and 2, on the other hand, both raise a number of substantive issues, and find that additional data and experimental extensions would be required to provide a sufficiently major advance in our understanding. Since it is not clear if and how these issues could be adequately addressed during a regular round of major revision, I would at this point invite you to discuss the reports with your coworkers and to send me a tentative point-by-point response, detailing how you might envision addressing the key concerns of the referees. Subsequently, I would be happy to discuss these revision plans with you directly, in order to find out whether a major revision for The EMBO Journal would seem realistic, or whether a less substantively revised version might alternatively be suitable for our sister journal EMBO Reports, as proposed by referee 1. To expedite the final decision, it would be great if you could back at your earliest convenience, ideally within the next 2 weeks. Looking forward to hearing from you,

With best regards,

Hartmut

Referee #1 (Report for Author)

This is an interesting but incomplete paper from the Storchova Lab regarding the evolution of aneuploid cells. The authors try to take on an important discrepancy - most cancers are aneuploid, but if you just generate aneuploidy in the lab, it hinders growth. So, how do you get from an antiproliferative aneuploidy to an aneuploid cancer that grows robustly?

My top-line conclusion is that with a lot of additional work, this manuscript could be appropriate for EMBO Journal, but I think that it's also a good candidate for transfer to EMBO reports.

1) Prior literature on the evolution of aneuploid cells. Two big papers from the Amon and Torres groups previously took on this topic, and their results are not cited or discussed here: Torres, Cell 2012 and Sheltzer, Cancer Cell 2017. In the Torres paper, the authors evolved aneuploid yeast, and then did deep sequencing on the evolved lines that revealed an abundance of mutations in protein-homeostasis and degradation genes. The authors then showed that these mutations helped deal

with the aneuploidy-induced protein imbalances. In light of this previous work, I think that it's important for the authors to perform whole-exome sequencing on their evolved cells to determine whether point mutations are contributing to the evolved fitness. I can imagine that this would be difficult in the highly mutagenic HCT116 cells but it should be feasible in RPE1, which I expect will have few background mutations? Without mutation data - and follow-up work to test any hits - this story is not complete.

Regarding the Seltzer paper, the results are similar to what is reported here, and in fact seem to cover many of the exact same cell lines. In that paper, the authors evolved aneuploid cells, and find that many of them display karyotypic changes that seem to enhance fitness. How do these current results compare to that paper? A direct comparison seems feasible as they seem to be the exact same cell lines.

2) Cause vs. correlation in the interpretation of results. The authors spend most of this paper discussing the upregulation of cell cycle genes in the evolved aneuploid cells. The authors call this "adaptation" to aneuploidy. The genes they highlight are the MCM's, Ki67, CDK1, Aurora B, and similar - but these are cell cycle genes that are expressed at higher levels in proliferating cells! The upregulation of many of these genes is likely just a consequence of the improved proliferation and not a driver of adaptation itself. The problem of inferring "cause" vs. "correlation" with cell cycle gene signatures was dealt with in Venet et al. Plos Comp Bio 2011.

This issue confounds the interpretation of the FOXM1 results. FOXM1 is a cell cycle transcription factor. Over-expressing FOXM1 accelerates the S-G2-M transition. So it isn't surprising that FOXM1 over-expression enhances the growth of the aneuploid cells. But, consider what the authors write in the abstract: "We identified E2F4 and FOXM1 as transcription factors required for adaptation to aneuploidy in vitro...". Their results do not support the claim that FOXM1 is "required" for adaptation to aneuploidy. The missing experiment is to delete FOXM1 in the aneuploid cells and then show that after the same number of population doublings, the aneuploid cells don't grow any better. In the absence of that experiment, to be blunt, showing that over-expressing a cell cycle transcription factor enhances cell cycle progression is a trivial result.

3) Dosage compensation in aneuploid cells. I think that this topic deserves more attention. It seems like the authors have generated isogenic or near-isogenic cell lines that harbor the same aneuploidy but exhibit differing levels of dosage compensation. This may be the first time that such a system has been created, and it could be used to really dive into how dosage compensation happens in cancer. Returning to point #1 above, I wonder if the "evolved" cells have mutations that enhance protein turnover? If not, how is the increase in dosage compensation occurring? Is translation affected? This could be tested by ribosome profiling. Or is protein turnover affected? This could be tested using pulse-chase labeling experiments. Getting some insight here would significantly improve the paper.

The analysis of dosage-compensation is rather superficial. I see that in some cell lines, there is chromosome-wide dosage compensation while in others the signal is more modest. I think that it is worth looking at a recent bioRxiv paper from the Fojjier Lab (Rendo et al., bioRxiv 2023), which

suggested that there are certain "toxic" genes that must be dosage compensated when present on an aneuploidy. Perhaps there are one or a few genes on each chromosome (maybe the ones highlighted by Rendo et al) that are being compensated?

Or, I wonder, does the expression pattern from aneuploid chromosomes become more "tumor-like" over time? The authors could perform an "AADEPT"-like analysis, but focused specifically on which genes are over-expressed or compensated on gained chromosomes in tumors. Maybe in their engineered aneuploidy system, you get a raw over-expression of most genes, but over time you get evolution towards a balanced pattern that allows aneuploid tumors to grow robustly.

Referee #2 (Report for Author)

Summary: In this manuscript, the authors address the aneuploidy paradox (the fact that aneuploidy has a negative effect on cellular fitness but is a hallmark of cancer, a disease of uncontrolled cell growth) by aiming to identify expression changes that alleviate the negative consequences of aneuploidy on cell proliferation. The authors show that aneuploid cell lines created via microcell mediated chromosome transfer initially have a reduced proliferation rate compared to their parental cell lines, but allowing the cells to "evolve" over 50 passages resulted in a partial recovery of this fitness defect. Using proteogenomic analysis of the aneuploid cell lines before and after evolution identified expression changes in DNA replication and lysosomal genes, changes that the authors show are also observed in aneuploid tumors. The authors show that increased activity of the transcription factor FOXM1 is a common characteristic of the evolved aneuploid cells and highly aneuploid tumors and experimentally show that increasing FOXM1 expression can improve the proliferation of unevolved aneuploid cells. Finally, the authors show that the partial reversal of one of the studied trisomies (chr 5) is also associated with improved proliferation of cells in vitro and worse survival in cancer patients.

Overall impressions: Altogether, this manuscript makes the novel finding that the reversal of common aneuploidy-associated gene expression changes in DNA replication and lysosomal stress are associated with improved proliferation. This grants exciting insight into how aneuploidy can be tolerated and therefore so widely observed in tumors, potentially aiding the field in explaining the aneuploidy paradox in cancer. However, additional data is required to support the authors' conclusions.

Major points:

1. The data shown in Figure 5D, E, F while very interesting would require additional data to confirm the conclusions. It is quite surprising the fact that deletion of the whole chr5 leads to almost zero colonies (as per figure and quantification). Since one of the clones that lost chr5 seems also to have lost a telomeric portion of 8q where likely MYC is located, this could be a reason why this clone forms so few colonies. While this would not explain the behavior of the other clones, it suggests that the experiment would benefit from more clones (2-3 more) that lost chromosome 5 and ideally also 5p only. A more clear genomic profile including focal events should be added to the figure 5. Did the clones lose the same chromosome 5 (or 5p)? (paternal, maternal, the one added

through MMCT) Deep whole genome sequencing should be able to address this point.

2. Supp Figure 2E: Why doesn't p53 signaling increase between WT and polysomic cells? Is it because the aneuploidy was introduced via MMCT instead of missegregation, or are single gains not enough to activate p53? Others showed these cell lines do activate the pathway when aneuploid. See <https://doi.org/10.1016/j.celrep.2021.108892>

3. In general, there needs to be more details/explanation of the connection between the markers of DNA replication stress investigated (DNA combing experiments, CHK1 signaling, γ H2AX) and the effect of FOXM1 overexpression/5q loss as they relate to proliferation rate (Supplementary Figure 7, Fig. 3, supp. Fig. 10). It is unclear to what degree/what kind of DNA damage or instability is different between WT, aneuploid, and evolved cells and how FOXM1 overexpression/5q loss might be contributing.

a. When making the conclusion "We conclude that one of the strongest changes upon in vitro evolution of cells with additional chromosomes is an increased expression of DNA repair and replication factors and reduced genomic instability, which is reflected by reduced checkpoint signaling and decreased DNA damage," Why isn't γ -H2AX staining shown at this point? It is unclear how a change of +/-1% in anaphase bridges is supposed to contribute to the argument.

b. What would be the significance of change in inter-origin distance?

4. In general, there are a lot of assumptions that the observed changes in gene expression (especially in the tumor data) are related to the recovery of proliferation after evolution before showing causation. Did the authors consider or look at any other phenotypes that could differ before and after evolution and potentially contribute to malignancy in vivo (growth under different conditions, migration, etc)?

5. Fig. 4H and 5E: For the assessment of the effects of FOXM1 overexpression and 5q loss on proliferation of Htr5, why was a clonogenic assay (10 days) used instead of the MTT assay as used in figure 1 (4 days)? Was the effect on proliferation not noticeable after 4 days?

6. What is the proposed mechanism of FOXM1 restoring proliferation in evolved aneuploid cells? Is there any phenotype that could be compared before and after evolution (time in metaphase?) to further support it as the contributing factor to change in proliferation rate?

7. How frequent the loss of 5q was after evolution? Can the authors perform FISH or single cell sequencing on the evolved population? Can the competition assay be performed with the Htr5 and Htr5p cells? This would further support that trisomy of 5q gives cells a disadvantage.

8. The findings that the Hdi5 clones have reduced, rather than improved proliferation are very interesting and unexpected. It would be interesting to know if these cells that were reverted to the disomic state for chr 5 have expression patterns more closely resembling the trisomic cells or the parental HCT116 cell line. Can the authors propose an explanation for how the cells adapted to "depend" on chr 5 trisomy for proliferation, yet also evolve to minimize the stress induced by the gain over time?

Minor points:

1. Fig. 4I,J: Replicate information is missing for the FOXM1 competition assays.

2. Fig. 3H: Caption refers to anaphase bridges but the figure shows micronuclei percentages.

3. Consider citing Sheltzer 2017 in the background of trisomies causing proliferative defects that can be reversed through evolution/karyotype changes <http://dx.doi.org/10.1016/j.ccell.2016.12.004>

Referee #3 (Report for Author)

Aneuploidy is a hallmark of cancer, a disease characterised by uncontrolled proliferation, yet it is very detrimental in untransformed cells. This suggests that aneuploid cancer cells can tolerate aneuploidy-associated cellular stresses. The study by Boekenkamp et al. identify adaptation strategies utilised by aneuploid cells to cope with the cellular stresses associated with the presence of an extra chromosome. They show that the enhanced proliferation of aneuploid cells was linked, among other factors, to diminished DNA damage and genomic instability, decreased replication stress, and modified expression of factors implicated in DNA replication and lysosomal degradation. These pathways and genes identified show a close correlation with the deregulated pathways observed in aneuploid cancers. The authors made several interesting observations, the data presented are convincing and the experiments well-done. The paper is well written, it reads well and the main conclusions are compelling and novel. Thus, the paper is suitable for publication in The EMBO Journal. This Reviewer has only minor points, listed below, that can be helpful in improving the work:

In the introduction, when referring to stresses present in cells with constitutive gain of an extra chromosome, it would be helpful to cite: <https://doi.org/10.1101/2023.01.27.525822>.
<https://doi.org/10.1101/2023.01.27.525826>

When discussing the selective advantage provided by aneuploidy, it would be worth citing: PMID: 34352223 34352222 34266888 34266887

This study focuses on stable aneuploid clones. It would be worth trying to induce chromosome mis-segregation in HCT116 expressing or not FOXM1-dNdK and see whether this is enough to rescue also the proliferation defects observed upon acute induction of aneuploidy.

Figure 4J lacks error bars.

Dear Hartmut,

Thank you very much for your personal assessment of our manuscript and the corresponding reviews. We were very pleased to see that all reviewers considered our manuscript interesting (Reviewer #1, 3), with novel findings (Reviewer #2, 3), and suitable for publication in *The EMBO Journal* (Reviewer #1, 3). Additionally, in particular reviewers 1 and 2 raised important questions that need to be addressed should the manuscript be accepted. We appreciate this constructive criticism, which largely tackles important points and will improve our manuscript. As you suggested, we prepared a point-by-point response where we summarize our detailed plan for the revision of the manuscript, summarizing the conduct of individual experiments and the expected results, including an assessment of their feasibility. We are confident that we will be able to address the critical concerns, remarks, and suggestions in approximately three months. We are convinced that the added experiments will significantly improve the manuscript and thus make it suitable for publication in *The EMBO Journal*.

Referee #1 (Report for Author)

This is an interesting but incomplete paper from the Storchova Lab regarding the evolution of aneuploid cells. The authors try to take on an important discrepancy - most cancers are aneuploid, but if you just generate aneuploidy in the lab, it hinders growth. So, how do you get from an antiproliferative aneuploidy to an aneuploid cancer that grows robustly?

My top-line conclusion is that with a lot of additional work, this manuscript could be appropriate for EMBO Journal, but I think that it's also a good candidate for transfer to EMBO reports.

RE> We are grateful for the reviewer's positive evaluation of our manuscript, acknowledging its interest and suitability for publication in the EMBO Journal. We would like to thank the expert for the constructive criticism and are confident that we can address most of the raised concerns.

1) Prior literature on the evolution of aneuploid cells. Two big papers from the Amon and Torres groups previously took on this topic, and their results are not cited or discussed here: Torres, Cell 2012 and Sheltzer, Cancer Cell 2017. In the Torres paper, the authors evolved aneuploid yeast, and then did deep sequencing on the evolved lines that revealed an abundance of mutations in protein-homeostasis and degradation genes. The authors then showed that these mutations helped deal with the aneuploidy-induced protein imbalances. In light of this previous work, I think that it's important for the authors to perform whole-exome sequencing on their evolved cells to determine whether point mutations are contributing to the evolved fitness. I can imagine that this would be difficult in the highly mutagenic HCT116 cells but it should be feasible in RPE1, which I expect will have few background mutations? Without mutation data - and follow-up work to test any hits - this story is not complete.

RE> The reviewer suggests evaluating point mutations in the evolved clones to identify mutations in individual genes which contribute to improved proliferation. This is a valid idea and in fact, we have anticipated this and recently performed whole genome sequencing at 30x of evolved and unevolved Rtr21 (trisomy 21), HCT116 (parent), and Htr5 (trisomy 5), respectively. While indeed HCT116 with its mutagenic features due to MMR deficiency will accumulate more mutations, clonal selection of favorable genotypes should lead to enrichment of specific mutations. Thus, we believe that both RPE1 and HCT116 derived cell lines will be useful for this analysis. The data has arrived, and we are already in the process of analyzing it. We thus and expect to be able to conclude these analyses well within the time frame of the revision.

We apologize for omitting some of these important references, which will all be added and discussed in the revised version.

Regarding the Seltzer paper, the results are similar to what is reported here, and in fact seem to cover many of the exact same cell lines. In that paper, the authors evolved aneuploid cells, and find that many of them display karyotypic changes that seem to enhance fitness. How do these current

results compare to that paper? A direct comparison seems feasible as they seem to be the exact same cell lines.

RE> It might be indeed very interesting to compare our results to those described by Sheltzer et al 2017 particularly, because the two studies compare evolution of the same aneuploid clones in a very different setting. While Sheltzer et al. performed evolution experiments in an *in vivo* mouse xenograft model, in our study we describe the molecular and phenotypic changes after evolution of aneuploid cells *in vitro*.

One notable difference between the two studies is the frequent loss of the extra chromosome during *in vivo* evolution of ex xenograft aneuploid cells, while in our study all aneuploid cells retained the full or at least large parts of the extra chromosome. It should be noted that there we did not use any selection to maintain the presence of the extra chromosome in the course of *in vitro* evolution. Thus, our study brings an important conclusion, namely that human cells adapt to aneuploidy by mechanisms that do not solely rely on the loss of the extra chromosome.

As we in fact provided the aneuploid HCT116 cell lines to the Sheltzer et al. Cancer Cell 2017 study, we also obtained the evolved cell lines from the *in vivo* xenograft study from the Sheltzer lab and already performed many of the exact same phenotypic analyses. However, because these cells lost the extra chromosome (Sheltzer et al, 2017, Fig.7B: trisomy of chr. 3 was lost in 3 out of 3 cases, accompanied by a partial gain of chr. 21 (2/3), tetrasomy of chr. 5 was lost (3/3), and so on), we decided against including these analyses into the current manuscript. We however would be happy to discuss these results with the editor and eventually include some of the results to the manuscript. Independent of this we will add a paragraph discussing the results of the two studies and cite Sheltzer et al. as suggested.

2) Cause vs. correlation in the interpretation of results. The authors spend most of this paper discussing the upregulation of cell cycle genes in the evolved aneuploid cells. The authors call this "adaptation" to aneuploidy. The genes they highlight are the MCM's, Ki67, CDK1, Aurora B, and similar - but these are cell cycle genes that are expressed at higher levels in proliferating cells! The upregulation of many of these genes is likely just a consequence of the improved proliferation and not a driver of adaptation itself. The problem of inferring "cause" vs. "correlation" with cell cycle gene signatures was dealt with in Venet et al. Plos Comp Bio 2011.

RE> We understand reviewers concern that the upregulation of DNA replication proteins might be a proteomic signature reflecting the enhanced proliferation of the aneuploid cells after evolution. We thank the reviewer for pointing this out and will adjust the interpretation. However, we would like to highlight that in our manuscript we do not want to derive an "adaptation signature". We demonstrate that aneuploid cells can improve proliferation despite maintaining their aberrant chromosomal content – and this is obviously associated with increased expression of genes relevant to proliferation – and that this improved proliferation is accompanied by reduced genomic instability and DNA damage. Additionally, we show that the gene expression changes correlate with the gene expression changes in cancer cells, and the correlation is stronger in aneuploid cancers.

This issue confounds the interpretation of the FOXM1 results. FOXM1 is a cell cycle transcription factor. Over-expressing FOXM1 accelerates the S-G2-M transition. So it isn't surprising that FOXM1 over-expression enhances the growth of the aneuploid cells. But, consider what the authors write in the abstract: "We identified E2F4 and FOXM1 as transcription factors required for adaptation to aneuploidy *in vitro*...". Their results do not support the claim that FOXM1 is "required" for adaptation to aneuploidy. The missing experiment is to delete FOXM1 in the aneuploid cells and then show that after the same number of population doublings, the aneuploid cells don't grow any better. In the absence of that experiment, to be blunt, showing that over-expressing a cell cycle transcription factor enhances cell cycle progression is a trivial result.

RE> Our observation, that FoxM1 expression in wildtype cells does not lead to enhanced proliferation clearly shows that this is not as trivial as put by the reviewer. We show that FOXM1 overexpression selectively improves proliferation in aneuploid, while it rather impairs proliferation in the parent HCT116 cells (Fig. 4H). In addition, only aneuploid cells profited from the overexpression of FOXM1 in the competition experiment (Fig. 4J). While FOXM1 is a general cell cycle transcription regulator, other transcription factors that enhance cell cycle progression were not identified as contributing to the improved proliferation of aneuploid cells. If this would be just simply enhancing cell cycle progression, any of the other factors could contribute as well. Similarly, the expression of FOXM1 and its targets correlates with aneuploidy in time (AADEPT score, Fig. 4E), while other proliferation-related transcription factors do not show this association. We will also explain this link better in our Discussion and will tone down that statement in the Abstract that FOXM1 is required for adaptation.

While the idea of conducting an evolution experiment with FoxM1 knock-out aneuploid cells appears feasible and interesting, the outcome and interpretation might be confounded by additional phenotypes caused by FoxM1 loss. While we have the expertise to generate knock-out cell lines, we find that the experiment's technical feasibility is overshadowed by its uncertain outcome, extensive procedure, and cost. Consequently, we deem it outside the manuscript's scope, as its inclusion would only marginally bolster the conclusiveness of our findings.

Should however the editor and reviewer insist, the experiment with FOXM1 deletion can be performed. To this end, we will use CRISPR/Cas9, which is well established in my laboratory, to delete FOXM1 in parental and aneuploid cell lines. Selection of the clones with and without FOXM1 and their further passaging to achieve comparable number of population doubling as in the *in vitro* evolution will then allow to identify whether loss of FOXM1 reduced the ability of trisomic cells to improve their proliferation. This procedure would be similar to the "colony selection" clones (see manuscript fig. 1A, B, C) from our *in vitro* evolution experiment. Nevertheless, we consider this experiment difficult to interpret due to complex phenotypes of FOXM1 loss, and thus unlikely to provide significant insight.

3) Dosage compensation in aneuploid cells. I think that this topic deserves more attention. It seems like the authors have generated isogenic or near-isogenic cell lines that harbor the same aneuploidy but exhibit differing levels of dosage compensation. This may be the first time that such a system has been created, and it could be used to really dive into how dosage compensation happens in cancer. Returning to point #1 above, I wonder if the "evolved" cells have mutations that enhance protein turnover? If not, how is the increase in dosage compensation occurring? Is translation affected? This could be tested by ribosome profiling. Or is protein turnover affected? This could be tested using pulse-chase labeling experiments. Getting some insight here would significantly improve the paper. The analysis of dosage-compensation is rather superficial. I see that in some cell lines, there is chromosome-wide dosage compensation while in others the signal is more modest. I think that it is worth looking at a recent bioRxiv paper from the Fojijier Lab (Rendo et al., bioRxiv 2023), which suggested that there are certain "toxic" genes that must be dosage compensated when present on an aneuploidy. Perhaps there are one or a few genes on each chromosome (maybe the ones highlighted by Rendo et al) that are being compensated?

RE> We agree with the reviewer that the dosage compensation is a very interesting effect which can potentially explain some of the observed improvements in polysomic cells. As such, the adapted cell lines provide an exciting opportunity to evaluate the effects of dosage compensation on cell proliferation at the level of each individual protein. Given the availability of proteomics data, we will reanalyze the data during the revisions with a focus on dosage compensation of specific genes (most overabundant or generally toxic/ tumor suppressive) located on the extra chromosomes before and after evolution. In fact, we have already initiated this analysis.

We would like to note that while the preprint from Fojijier lab (Rendo et al, 2023) is highly interesting, it focuses on dosage compensation on RNA level. In our model systems we observed that dosage compensation is much stronger on the level of proteins than transcripts (Stingele et al, 2012, Selbach et al, 2016, Chunduri et al, 2021). Nevertheless, we are thankful to the reviewer to point this out and

will analyze the degree of compensation for the genes of concern wherever proteomic data will be available (considering that transcripts are much more comprehensively measured than proteins). The mechanism of dosage compensation is still poorly understood. There is evidence for increased protein turnover, particularly via an upregulation of proteasomal degradation. However, additional mechanisms are likely to contribute to dosage compensation as well. We have already performed analysis of proteasome activity in HCT116 cells carrying extra copy of chromosome 5 (See fig. R1). Encouragingly, after *in vitro* evolution proteasomal activity was significantly increased in these cell lines. For the revision we plan to perform proteasomal activity assays with additional pairs of evolved and non-evolved aneuploid cell lines both in the HCT116 and RPE background.

Or, I wonder, does the expression pattern from aneuploid chromosomes become more "tumor-like" over time? The authors could perform an "AADEPT"-like analysis, but focused specifically on which genes are over-expressed or compensated on gained chromosomes in tumors. Maybe in their engineered aneuploidy system, you get a raw over-expression of most genes, but over time you get evolution towards a balanced pattern that allows aneuploid tumors to grow robustly.

RE> We thank the reviewer for this interesting idea. We will assess the profiles of deregulated proteins encoded on the extra chromosomes before and after evolution and compare how they are expressed in corresponding aneuploid cancers of the CPTAC and TCGA.

Referee #2 (Report for Author)

Summary: In this manuscript, the authors address the aneuploidy paradox (the fact that aneuploidy has a negative effect on cellular fitness but is a hallmark of cancer, a disease of uncontrolled cell growth) by aiming to identify expression changes that alleviate the negative consequences of aneuploidy on cell proliferation. The authors show that aneuploid cell lines created via microcell mediated chromosome transfer initially have a reduced proliferation rate compared to their parental cell lines, but allowing the cells to "evolve" over 50 passages resulted in a partial recovery of this fitness defect. Using proteogenomic analysis of the aneuploid cell lines before and after evolution identified expression changes in DNA replication and lysosomal genes, changes that the authors show are also observed in aneuploid tumors. The authors show that increased activity of the

transcription factor FOXM1 is a common characteristic of the evolved aneuploid cells and highly aneuploid tumors and experimentally show that increasing FOXM1 expression can improve the proliferation of unevolved aneuploid cells. Finally, the authors show that the partial reversal of one of the studied trisomies (chr 5) is also associated with improved proliferation of cells in vitro and worse survival in cancer patients.

Overall impressions: Altogether, this manuscript makes the novel finding that the reversal of common aneuploidy-associated gene expression changes in DNA replication and lysosomal stress are associated with improved proliferation. This grants exciting insight into how aneuploidy can be tolerated and therefore so widely observed in tumors, potentially aiding the field in explaining the aneuploidy paradox in cancer. However, additional data is required to support the authors' conclusions.

RE> We thank the reviewer for this positive evaluation as well as for the suggestions how to improve the manuscript.

Major points:

1. The data shown in Figure 5D, E, F while very interesting would require additional data to confirm the conclusions. It is quite surprising the fact that deletion of the whole chr5 leads to almost zero colonies (as per figure and quantification). Since one of the clones that lost chr5 seems also to have lost a telomeric portion of 8q where likely MYC is located, this could be a reason why this clone forms so few colonies. While this would not explain the behavior of the other clones, it suggests that the experiment would benefit from more clones (2-3 more) that lost chromosome 5 and ideally also 5p only. A more clear genomic profile including focal events should be added to the figure 5. Did the clones lose the same chromosome 5 (or 5p)? (paternal, maternal, the one added through MMCT) Deep whole genome sequencing should be able to address this point.

RE> We agree with the reviewer that the findings with the loss of chromosome 5 are very surprising. We appreciate the suggestion with the chr. 8q. Indeed, quick reassessment of the sequencing data revealed that the third copy of MYC gene is lost in this cell line. For validation we will conduct immunoblotting to evaluate the expression of MYC in these clones. Moreover, we have several additional clones from the REDACT experiment which were not fully analyzed yet. We will analyze these clones in order to obtain more clones with the specific karyotypic change and add the results to the manuscript.

While we can add additional genomic profile and deeper sequencing, we are not sure how the knowledge which chromosome 5 (maternal, paternal, or exogenous) was lost would contribute to understanding of the consequence of chromosome gains and losses. In our view this costly and time-consuming experiment does not bring any additional insight. However, as elaborated under point 8 below, we will perform a proteomic analysis of these clones to further characterize the changes that lead to altered cell growth. My laboratory has extensive experience in proteomics and we do not foresee any problem or time constrains in obtaining and analyzing this data.

2. Supp Figure 2E: Why doesn't p53 signaling increase between WT and polysomic cells? Is it because the aneuploidy was introduced via MMCT instead of missegregation, or are single gains not enough to activate p53? Others showed these cell lines do activate the pathway when aneuploid. See

<https://doi.org/10.1016/j.celrep.2021.108892>

RE> The main difference is that p53 is activated after acute missegregation. The mentioned paper, and several others, e.g., from Medema, Amon and Santaguida lab show that acute aneuploidy, 24 h – 48 h after chromosome missegregation, leads to the activation of the p53 pathway. In contrast, constitutive aneuploidy (regardless of whether generated via MMCT or chromosome missegregation) does not activate this pathway. We will add this important point to the discussion.

3. In general, there needs to be more details/explanation of the connection between the markers of DNA replication stress investigated (DNA combing experiments, CHK1 signaling, γH2AX) and the

effect of FOXM1 overexpression/5q loss as they relate to proliferation rate (Supplementary Figure 7, Fig. 3, supp. Fig. 10). It is unclear to what degree/what kind of DNA damage or instability is different between WT, aneuploid, and evolved cells and how FOXM1 overexpression/5q loss might be contributing.

RE> We agree with the reviewer and will add analyses to address these important points. To this end, we will analyze DNA damage and replication stress also in the cell lines overexpressing FOXM1, as well as in the engineered 5q loss clones.

a. When making the conclusion "We conclude that one of the strongest changes upon in vitro evolution of cells with additional chromosomes is an increased expression of DNA repair and replication factors and reduced genomic instability, which is reflected by reduced checkpoint signaling and decreased DNA damage," Why isn't γ -H2AX staining shown at this point? It is unclear how a change of +/-1% in anaphase bridges is supposed to contribute to the argument.

RE> We have data available for γ -H2AX in evolved and unevolved cell lines, which we are happy to add to the manuscript (see an example in Fig. R2). The data show reduced γ -H2AX staining after evolution. We will now also test the cells overexpressing FOXM1 and 5q loss. My laboratory has an extensive experience with these experiments, and we will be able to collect the necessary data quickly. Below is an example of already available data.

b. What would be the significance of change in inter-origin distance?

RE> We believe that this reflects the MCM levels. It has been previously demonstrated that reduced MCM abundance results in increased inter-origin distance, and in turn in replication stress, as the areas to be replicated from one origin are larger (e.g., Ge et al, 2007). This is indeed what we observe in polysomic cells. In contrast, upon evolution, the abundance of MCM increases and the distance is reduced, which in turn fits well with the reduced replication stress.

4. In general, there are a lot of assumptions that the observed changes in gene expression (especially in the tumor data) are related to the recovery of proliferation after evolution before showing causation. Did the authors consider or look at any other phenotypes that could differ before and after evolution and potentially contribute to malignancy in vivo (growth under different conditions, migration, etc)?

RE> We thank the reviewer for this interesting idea. We have not performed any similar experiments in this context; however, we have previous experience with wound healing and migration assays, which we plan to use for this evaluation and will add the data to the manuscript.

5. Fig. 4H and 5E: For the assessment of the effects of FOXM1 overexpression and 5q loss on proliferation of Htr5, why was a clonogenic assay (10 days) used instead of the MTT assay as used in figure 1 (4 days)? Was the effect on proliferation not noticeable after 4 days?

RE> There was no specific reason. We are happy to perform additional proliferation assays using the MTT assay.

6. What is the proposed mechanism of FOXM1 restoring proliferation in evolved aneuploid cells? Is there any phenotype that could be compared before and after evolution (time in metaphase?) to further support it as the contributing factor to change in proliferation rate?

RE> This is an important point, and we thank for this suggestion. We have previously performed flow cytometry with these cells, and the DAPI-based cell cycle profile did show a small increase of G2/M fraction in Htr5 FOXM1, but not in HCT116 FOXM1. We plan to add the analysis in combination with EdU staining and with a marker for mitotic cells (MPM2 phosphorylation) to obtain a full evaluation of the cell cycle. My group has extensive experience with cell cycle analysis and this data should be collected soon.

7. How frequent the loss of 5q was after evolution? Can the authors perform FISH or single cell sequencing on the evolved population? Can the competition assay be performed with the Htr5 and Htr5p cells? This would further support that trisomy of 5q gives cells a disadvantage.

RE> We thank the reviewer for this interesting idea. We will extend our FISH/chromosome paints to the evolved cell lines to obtain a better overview of the frequency of the specific karyotypes. The competition assay for Htr5 and Htr5p is a nice idea, and we will add it to the manuscript.

8. The findings that the Hdi5 clones have reduced, rather than improved proliferation are very interesting and unexpected. It would be interesting to know if these cells that were reverted to the disomic state for chr 5 have expression patterns more closely resembling the trisomic cells or the parental HCT116 cell line. Can the authors propose an explanation for how the cells adapted to "depend" on chr 5 trisomy for proliferation, yet also evolve to minimize the stress induced by the gain over time?

RE> We share the reviewer's curiosity about the underlying mechanisms. To address the question, we plan to perform proteomics analysis in these cells and compare it with the parental and evolved trisomies. We have extensive experience with global proteomics and can collect this data soon.

Minor points:

1. Fig. 4I,J: Replicate information is missing for the FOXM1 competition assays.

RE> Instead of triplicates we have performed a label swap experiment, which showed the same trend (see Sup. Fig. 9). In each of these experiments, we have evaluated the ratio of FoxM1 OE to untreated control cells at three different time points, which all show the same trend. For each datapoint a population of 50'000 cells was evaluated. However, if the editorial team insists, we are happy to add two additional replicates of the experiment shown in Fig. 4J, even though this seems to be beyond the state-of-the-art required for this type of experiments (see for example Girish et al., Science 2023).

2. Fig. 3H: Caption refers to anaphase bridges but the figure shows micronuclei percentages.

RE> We apologize for this error which will be corrected in the revised manuscript.

3. Consider citing Sheltzer 2017 in the background of trisomies causing proliferative defects that can be reversed through evolution/karyotype changes <http://dx.doi.org/10.1016/j.ccell.2016.12.004>

RE> We will add the missing citations and expand/correct the figure caption as suggested.

Referee #3 (Report for Author)

Aneuploidy is a hallmark of cancer, a disease characterised by uncontrolled proliferation, yet it is very detrimental in untransformed cells. This suggests that aneuploid cancer cells can tolerate aneuploidy-associated cellular stresses. The study by Boekenkamp et al. identify adaptation strategies utilised by aneuploid cells to cope with the cellular stresses associated with the presence of an extra chromosome. They show that the enhanced proliferation of aneuploid cells was linked, among other factors, to diminished DNA damage and genomic instability, decreased replication stress, and modified expression of factors implicated in DNA replication and lysosomal degradation. These pathways and genes identified show a close correlation with the deregulated pathways observed in aneuploid cancers. The authors made several interesting observations, the data presented are convincing and the experiments well-done. The paper is well written, it reads well and the main conclusions are compelling and novel. Thus, the paper is suitable for publication in The EMBO Journal. This Reviewer has only minor points, listed below, that can be helpful in improving the work:

RE> We thank the reviewer for this very positive evaluation.

In the introduction, when referring to stresses present in cells with constitutive gain of an extra chromosome, it would be helpful to cite: <https://doi.org/10.1101/2023.01.27.525822>.
<https://doi.org/10.1101/2023.01.27.525826>

When discussing the selective advantage provided by aneuploidy, it would be worth citing: PMID: 34352223 34352222 34266888 34266887

RE> We will add a short discussion of these papers and include the references as suggested.

This study focuses on stable aneuploid clones. It would be worth trying to induce chromosome mis-segregation in HCT116 expressing or not FOXM1-dNdK and see whether this is enough to rescue also the proliferation defects observed upon acute induction of aneuploidy.

RE> We agree with the reviewer that this is a very interesting topic. However, the situation immediately after chromosome missegregation is much more complex because various chromosomes are missegregated, and gains, losses and chromosome breakages occur (see Soto et al, 2017, Santaguida et al, 2017). Moreover, the p53 pathways is activated immediately after chromosome missegregation. We are convinced that this experiment requires its own study and is beyond the scope of this manuscript.

Figure 4J lacks error bars.

RE> Instead of triplicates we have performed a label swap experiment, which showed the same trend (see Sup. Fig. 9). In each of these experiments, we have evaluated the ratio of FoxM1 OE to untreated control cells at three different time points, which all show the same trend. For each datapoint a population of 50'000 cells was evaluated. However, if the editorial team insists, we are happy to add two additional replicates of the experiment shown in Fig. 4J, even though this seems to be beyond the state-of-the-art required for this type of experiments (see for example Girish et al., Science 2023).

Dr. Zuzana Storchova
RPTU Kaiserslautern
Molecular Genetics
Paul Ehrlich Strasse 24
Kaiserslautern 67663
Germany

27th May 2024

Re: EMBOJ-2024-116981
Proteogenomic analysis reveals adaptive strategies to alleviate the consequences of aneuploidy

Dear Zuzana,

Thank you for your detailed revision plan and tentative point-by-point responses to the reviews on your recent EMBO journal submission. I have now had a chance to go through them, and I was pleased to see that you seem to be in a good position to address the most salient issues raised by the referees. In particular, analyses of genome sequencing data, and follow-up on dosage compensation and on the chromosome-5-related results should be valuable additions. I also agree that studying consequences of acute chromosome missegregation is not directly within the scope of this study. Regarding causal involvement of FOXM1, I appreciate your clarifications, and that the proposed deletion experiment may not provide very concrete understanding - so I would at present not insist on its inclusion during this revision. Possibly including data from xenograft-derived cell lines (response to ref 1) may potentially be helpful, and I would suggest discussing any such data in the response letter at the time of resubmission, so we could (with the input of the referees) decide whether or not some of them may be added to a final revised version.

Please keep in mind that it is our policy to allow only a single round of (major) revision, and do update me should there be any unexpected problems with the revisions, or should you require an extension beyond the default 3-months deadline. As always, competing manuscript published during the course of this revision will not affect our final decision on your study. Finally, please note the detailed information and guidelines on how to prepare a revision below (and in our online Guide to Authors) - closely adhering to them shall greatly facilitate the editorial process at the time of resubmission.

Thank you again for the opportunity to consider this work, and I look forward to receiving your revision in due time.

With kind regards,

Hartmut

3) Revised manuscript text (including main tables, and figure legends for main and EV figures) has to be submitted as editable

text file (e.g., .docx format). We encourage highlighting of changes (e.g., via text color) for the referees' reference.

4) Each main and each Expanded View (EV) figure should be uploaded as individual production-quality files (preferably in .eps, .tif, .jpg formats). For suggestions on figure preparation/layout, please refer to our Figure Preparation Guidelines:

8) Please note that supplementary information at EMBO Press has been superseded by the 'Expanded View' for inclusion of additional figures, tables, movies or datasets; with up to five EV Figures being typeset and directly accessible in the HTML version of the article. For details and guidance, please refer to:

embopress.org/page/journal/14602075/authorguide#expandedview

9) Digital image enhancement is acceptable practice, as long as it accurately represents the original data and conforms to community standards. If a figure has been subjected to significant electronic manipulation, this must be clearly noted in the figure legend and/or the 'Materials and Methods' section. The editors reserve the right to request original versions of figures and the original images that were used to assemble the figure. Finally, we generally encourage uploading of numerical as well as gel/blot image source data; for details see: embopress.org/page/journal/14602075/authorguide#sourcedata

At EMBO Press, we ask authors to provide source data for the main manuscript figures. Our source data coordinator will contact you to discuss which figure panels we would need source data for and will also provide you with helpful tips on how to upload and organize the files.

Further information is available in our Guide For Authors:

In the interest of ensuring the conceptual advance provided by the work, we recommend submitting a revision within 3 months (25th Aug 2024). Please discuss the revision progress ahead of this time with the editor if you require more time to complete the revisions. Use the link below to submit your revision:

Link Not Available

Referee #1:

This is an interesting but incomplete paper from the Storchova Lab regarding the evolution of aneuploid cells. The authors try to take on an important discrepancy - most cancers are aneuploid, but if you just generate aneuploidy in the lab, it hinders growth. So, how do you get from an antiproliferative aneuploidy to an aneuploid cancer that grows robustly?

My top-line conclusion is that with a lot of additional work, this manuscript could be appropriate for EMBO Journal, but I think that it's also a good candidate for transfer to EMBO reports.

1) Prior literature on the evolution of aneuploid cells. Two big papers from the Amon and Torres groups previously took on this topic, and their results are not cited or discussed here: Torres, Cell 2012 and Sheltzer, Cancer Cell 2017. In the Torres paper, the authors evolved aneuploid yeast, and then did deep sequencing on the evolved lines that revealed an abundance of mutations in protein-homeostasis and degradation genes. The authors then showed that these mutations helped deal with the aneuploidy-induced protein imbalances. In light of this previous work, I think that it's important for the authors to perform whole-exome sequencing on their evolved cells to determine whether point mutations are contributing to the evolved fitness. I can imagine that this would be difficult in the highly mutagenic HCT116 cells but it should be feasible in RPE1, which I expect will have few background mutations? Without mutation data - and follow-up work to test any hits - this story is not complete.

Regarding the Seltzer paper, the results are similar to what is reported here, and in fact seem to cover many of the exact same

cell lines. In that paper, the authors evolved aneuploid cells, and find that many of them display karyotypic changes that seem to enhance fitness. How do these current results compare to that paper? A direct comparison seems feasible as they seem to be the exact same cell lines.

2) Cause vs. correlation in the interpretation of results. The authors spend most of this paper discussing the upregulation of cell cycle genes in the evolved aneuploid cells. The authors call this "adaptation" to aneuploidy. The genes they highlight are the MCM's, Ki67, CDK1, Aurora B, and similar - but these are cell cycle genes that are expressed at higher levels in proliferating cells! The upregulation of many of these genes is likely just a consequence of the improved proliferation and not a driver of adaptation itself. The problem of inferring "cause" vs. "correlation" with cell cycle gene signatures was dealt with in Venet et al. Plos Comp Bio 2011.

This issue confounds the interpretation of the FOXM1 results. FOXM1 is a cell cycle transcription factor. Over-expressing FOXM1 accelerates the S-G2-M transition. So it isn't surprising that FOXM1 over-expression enhances the growth of the aneuploid cells. But, consider what the authors write in the abstract: "We identified E2F4 and FOXM1 as transcription factors required for adaptation to aneuploidy in vitro...". Their results do not support the claim that FOXM1 is "required" for adaptation to aneuploidy. The missing experiment is to delete FOXM1 in the aneuploid cells and then show that after the same number of population doublings, the aneuploid cells don't grow any better. In the absence of that experiment, to be blunt, showing that over-expressing a cell cycle transcription factor enhances cell cycle progression is a trivial result.

3) Dosage compensation in aneuploid cells. I think that this topic deserves more attention. It seems like the authors have generated isogenic or near-isogenic cell lines that harbor the same aneuploidy but exhibit differing levels of dosage compensation. This may be the first time that such a system has been created, and it could be used to really dive into how dosage compensation happens in cancer. Returning to point #1 above, I wonder if the "evolved" cells have mutations that enhance protein turnover? If not, how is the increase in dosage compensation occurring? Is translation affected? This could be tested by ribosome profiling. Or is protein turnover affected? This could be tested using pulse-chase labeling experiments. Getting some insight here would significantly improve the paper.

The analysis of dosage-compensation is rather superficial. I see that in some cell lines, there is chromosome-wide dosage compensation while in others the signal is more modest. I think that it is worth looking at a recent bioRxiv paper from the Fojjier Lab (Rendo et al., bioRxiv 2023), which suggested that there are certain "toxic" genes that must be dosage compensated when present on an aneuploidy. Perhaps there are one or a few genes on each chromosome (maybe the ones highlighted by Rendo et al) that are being compensated?

Or, I wonder, does the expression pattern from aneuploid chromosomes become more "tumor-like" over time? The authors could perform an "AADEPT"-like analysis, but focused specifically on which genes are over-expressed or compensated on gained chromosomes in tumors. Maybe in their engineered aneuploidy system, you get a raw over-expression of most genes, but over time you get evolution towards a balanced pattern that allows aneuploid tumors to grow robustly.

Referee #2:

Summary: In this manuscript, the authors address the aneuploidy paradox (the fact that aneuploidy has a negative effect on cellular fitness but is a hallmark of cancer, a disease of uncontrolled cell growth) by aiming to identify expression changes that alleviate the negative consequences of aneuploidy on cell proliferation. The authors show that aneuploid cell lines created via microcell mediated chromosome transfer initially have a reduced proliferation rate compared to their parental cell lines, but allowing the cells to "evolve" over 50 passages resulted in a partial recovery of this fitness defect. Using proteogenomic analysis of the aneuploid cell lines before and after evolution identified expression changes in DNA replication and lysosomal genes, changes that the authors show are also observed in aneuploid tumors. The authors show that increased activity of the transcription factor FOXM1 is a common characteristic of the evolved aneuploid cells and highly aneuploid tumors and experimentally show that increasing FOXM1 expression can improve the proliferation of unevolved aneuploid cells. Finally, the authors show that the partial reversal of one of the studied trisomies (chr 5) is also associated with improved proliferation of cells in vitro and worse survival in cancer patients.

Overall impressions: Altogether, this manuscript makes the novel finding that the reversal of common aneuploidy-associated gene expression changes in DNA replication and lysosomal stress are associated with improved proliferation. This grants exciting insight into how aneuploidy can be tolerated and therefore so widely observed in tumors, potentially aiding the field in explaining the aneuploidy paradox in cancer. However, additional data is required to support the authors' conclusions.

Major points:

1. The data shown in Figure 5D, E, F while very interesting would require additional data to confirm the conclusions. It is quite surprising the fact that deletion of the whole chr5 leads to almost zero colonies (as per figure and quantification). Since one of the clones that lost chr5 seems also to have lost a telomeric portion of 8q where likely MYC is located, this could be a reason

why this clone forms so few colonies. While this would not explain the behavior of the other clones, it suggests that the experiment would benefit from more clones (2-3 more) that lost chromosome 5 and ideally also 5p only. A more clear genomic profile including focal events should be added to the figure 5. Did the clones lose the same chromosome 5 (or 5p)? (paternal, maternal, the one added through MMCT) Deep whole genome sequencing should be able to address this point.

2. Supp Figure 2E: Why doesn't p53 signaling increase between WT and polysomic cells? Is it because the aneuploidy was introduced via MMCT instead of missegregation, or are single gains not enough to activate p53? Others showed these cell lines do activate the pathway when aneuploid. See <https://doi.org/10.1016/j.celrep.2021.108892>

3. In general, there needs to be more details/explanation of the connection between the markers of DNA replication stress investigated (DNA combing experiments, CHK1 signaling, γ H2AX) and the effect of FOXM1 overexpression/5q loss as they relate to proliferation rate (Supplementary Figure 7, Fig. 3, supp. Fig. 10). It is unclear to what degree/what kind of DNA damage or instability is different between WT, aneuploid, and evolved cells and how FOXM1 overexpression/5q loss might be contributing.

a. When making the conclusion "We conclude that one of the strongest changes upon in vitro evolution of cells with additional chromosomes is an increased expression of DNA repair and replication factors and reduced genomic instability, which is reflected by reduced checkpoint signaling and decreased DNA damage," Why isn't γ -H2AX staining shown at this point? It is unclear how a change of +/-1% in anaphase bridges is supposed to contribute to the argument.

b. What would be the significance of change in inter-origin distance?

4. In general, there are a lot of assumptions that the observed changes in gene expression (especially in the tumor data) are related to the recovery of proliferation after evolution before showing causation. Did the authors consider or look at any other phenotypes that could differ before and after evolution and potentially contribute to malignancy in vivo (growth under different conditions, migration, etc)?

5. Fig. 4H and 5E: For the assessment of the effects of FOXM1 overexpression and 5q loss on proliferation of Htr5, why was a clonogenic assay (10 days) used instead of the MTT assay as used in figure 1 (4 days)? Was the effect on proliferation not noticeable after 4 days?

6. What is the proposed mechanism of FOXM1 restoring proliferation in evolved aneuploid cells? Is there any phenotype that could be compared before and after evolution (time in metaphase?) to further support it as the contributing factor to change in proliferation rate?

7. How frequent the loss of 5q was after evolution? Can the authors perform FISH or single cell sequencing on the evolved population? Can the competition assay be performed with the Htr5 and Htr5p cells? This would further support that trisomy of 5q gives cells a disadvantage.

8. The findings that the Hdi5 clones have reduced, rather than improved proliferation are very interesting and unexpected. It would be interesting to know if these cells that were reverted to the disomic state for chr 5 have expression patterns more closely resembling the trisomic cells or the parental HCT116 cell line. Can the authors propose an explanation for how the cells adapted to "depend" on chr 5 trisomy for proliferation, yet also evolve to minimize the stress induced by the gain over time?

Minor points:

1. Fig. 4I,J: Replicate information is missing for the FOXM1 competition assays.

2. Fig. 3H: Caption refers to anaphase bridges but the figure shows micronuclei percentages.

3. Consider citing Sheltzer 2017 in the background of trisomies causing proliferative defects that can be reversed through evolution/karyotype changes <http://dx.doi.org/10.1016/j.ccell.2016.12.004>

Referee #3:

Aneuploidy is a hallmark of cancer, a disease characterised by uncontrolled proliferation, yet it is very detrimental in untransformed cells. This suggests that aneuploid cancer cells can tolerate aneuploidy-associated cellular stresses. The study by Boekenkamp et al. identify adaptation strategies utilised by aneuploid cells to cope with the cellular stresses associated with the presence of an extra chromosome. They show that the enhanced proliferation of aneuploid cells was linked, among other factors, to diminished DNA damage and genomic instability, decreased replication stress, and modified expression of factors implicated in DNA replication and lysosomal degradation. These pathways and genes identified show a close correlation with the deregulated pathways observed in aneuploid cancers. The authors made several interesting observations, the data presented are convincing and the experiments well-done. The paper is well written, it reads well and the main conclusions are compelling and novel. Thus, the paper is suitable for publication in The EMBO Journal. This Reviewer has only minor points, listed below, that can be helpful in improving the work:

In the introduction, when referring to stresses present in cells with constitutive gain of an extra chromosome, it would be helpful to cite: <https://doi.org/10.1101/2023.01.27.525822>. <https://doi.org/10.1101/2023.01.27.525826>

When discussing the selective advantage provided by aneuploidy, it would be worth citing: PMID: 34352223 34352222 34266888 34266887

This study focuses on stable aneuploid clones. It would be worth trying to induce chromosome mis-segregation in HCT116 expressing or not FOXM1-dNdK and see whether this is enough to rescue also the proliferation defects observed upon acute induction of aneuploidy.

Figure 4J lacks error bars.

Response to the reviewers

We were very pleased to see that the reviewers considered our manuscript interesting (Reviewer #1, 3), with novel findings (Reviewer #2, 3), and suitable for publication in *The EMBO Journal* (Reviewer #1, 3). However, in particular reviewers 1 and 2 raised several important questions that needed to be addressed should the manuscript be accepted. We appreciate this constructive criticism, which pointed out important aspects and helped to improve our manuscript. Below is a point-by-point response, where we summarize the response to reviewers' queries and individual experiments and the results that we added to the revised manuscript. We are confident that the new data answered the critical concerns, remarks, and suggestions, and significantly improved the manuscript and thus made it suitable for publication in *The EMBO Journal*.

Referee #1 (Report for Author)

This is an interesting but incomplete paper from the Storchova Lab regarding the evolution of aneuploid cells. The authors try to take on an important discrepancy - most cancers are aneuploid, but if you just generate aneuploidy in the lab, it hinders growth. So, how do you get from an antiproliferative aneuploidy to an aneuploid cancer that grows robustly?

My top-line conclusion is that with a lot of additional work, this manuscript could be appropriate for EMBO Journal, but I think that it's also a good candidate for transfer to EMBO reports.

RE> We are grateful for the reviewer's positive evaluation of our manuscript, acknowledging its interest and suitability for publication in the EMBO Journal. We would like to thank the expert for the constructive criticism and are confident that we were able to address most of the raised concerns.

1) Prior literature on the evolution of aneuploid cells. Two big papers from the Amon and Torres groups previously took on this topic, and their results are not cited or discussed here: Torres, Cell 2012 and Sheltzer, Cancer Cell 2017. In the Torres paper, the authors evolved aneuploid yeast, and then did deep sequencing on the evolved lines that revealed an abundance of mutations in protein-homeostasis and degradation genes. The authors then showed that these mutations helped deal with the aneuploidy-induced protein imbalances. In light of this previous work, I think that it's important for the authors to perform whole-exome sequencing on their evolved cells to determine whether point mutations are contributing to the evolved fitness. I can imagine that this would be difficult in the highly mutagenic HCT116 cells but it should be feasible in RPE1, which I expect will have few background mutations? Without mutation data - and follow-up work to test any hits - this story is not complete.

RE> The reviewer suggests evaluating point mutations in the evolved clones to identify mutations in individual genes which contribute to improved proliferation. This is a valid idea, and thus we have recently performed whole genome sequencing at 30x of evolved and unevolved Rtr21 (trisomy 21), HCT116 (parent), and Htr5 (trisomy 5), respectively. As the reviewer noted, the mismatch repair deficient Htr5 cells accumulated more genetic variants than Rtr21 (135 vs. 14) even when subtracting mutations observed also in evolved HCT116. By using the acquired proteome data, we confirmed the downregulation of most of the mutated genes. However, by comparison with data from TCGA colon and pan-cancer tumors we found that none of the identified mutations was associated with a higher degree of aneuploidy, putting their specificity as aneuploidy-tolerating mutations into question. Despite this, we decided to follow up on TP53BP1, which was the only gene with a deleterious

heterozygotic mutation and a significant protein abundance reduction after evolution in Rtr21. We validated that depletion of TP53BP1 indeed leads to improved proliferation in trisomy cells even without evolution. The novel data are summarized in Figure EV4 and Table EV7, as well as in a new chapter “**Gene mutations in evolved polysomic cells**”, page 10. While our new data suggest that there are individual mutations which may improve the proliferation in evolved polysomic cells lines, the lack of similarities between the two cell lines, and with aneuploid tumors provide further support to our decision to focus rather on the general pathway deregulation shared among evolved polysomes and cancers.

Regarding the Seltzer paper, the results are similar to what is reported here, and in fact seem to cover many of the exact same cell lines. In that paper, the authors evolved aneuploid cells, and find that many of them display karyotypic changes that seem to enhance fitness. How do these current results compare to that paper? A direct comparison seems feasible as they seem to be the exact same cell lines.

RE> The idea to compare our results to those described by Sheltzer et al 2017 is interesting indeed. However, it should be noted that the two studies compared evolution of the same trisomic cell lines in very different settings. While Sheltzer et al. performed evolution experiments in an *in vivo* mouse xenograft model, in our study we describe the molecular and phenotypic changes after evolution of aneuploid cells *in vitro*. This is probably the reason for the one important difference between these two settings, which is the frequent loss of the extra chromosome during *in vivo* evolution of post-xenograft aneuploid cells (see Sheltzer et al, 2017, Figure 7B), where ALL analyzed xenografts lost the extra chromosome during the *in vivo* growth (and some gained new aberrations). In contrast, all aneuploid cells retained either full or at least large parts of the extra chromosome in our study despite the fact that we did not use any selection to maintain the presence of the extra chromosome during *in vitro* evolution. This striking difference suggest that the selection *in vivo* against aneuploidy is stronger than *in vitro*, and in our opinion excludes the possibility of direct comparison, since the model conditions and the mechanisms of adaptation to aneuploidy seem very different.

The aneuploid HCT116 cell lines used in Sheltzer et al. Cancer Cell 2017 study were created in our laboratory, and we subsequently obtained the evolved cell lines from the *in vivo* xenograft study. We have previously performed phenotypic analyses, which suggest a slightly reduced replication stress, and variable changes in genomic instability (Figure R1). However, we believe that the loss of the extra chromosome strongly argues that we cannot compare the two sets of cell lines and thus we decided against including these analyses into the current manuscript.

2) Cause vs. correlation in the interpretation of results. The authors spend most of this paper discussing the upregulation of cell cycle genes in the evolved aneuploid cells. The authors call this "adaptation" to aneuploidy. The genes they highlight are the MCM's, Ki67, CDK1, Aurora B, and similar - but these are cell cycle genes that are expressed at higher levels in proliferating cells! The upregulation of many of these genes is likely just a consequence of the improved proliferation and not a driver of adaptation itself. The problem of inferring "cause" vs. "correlation" with cell cycle gene signatures was dealt with in Venet et al. Plos Comp Bio 2011.

RE> We agree with the reviewer's concern that the upregulation of DNA replication proteins other mentioned proteins might be a proteomic signature reflecting the enhanced proliferation of the aneuploid cells after evolution. We thank the reviewer for pointing this out and will adjust the interpretation. However, we would like to highlight that in our manuscript we do not want to derive an "adaptation signature". We demonstrate that aneuploid cells can improve proliferation despite

maintaining their aberrant chromosomal content – and this is obviously associated with increased expression of genes relevant to proliferation – and that this improved proliferation is accompanied by reduced genomic instability and DNA damage. Additionally, we show that the gene expression changes correlate with the gene expression changes in cancer cells, and that the correlation is stronger in aneuploid cancers than in near-diploid cancers.

We now added following statement:

While the enrichment of cell cycle proteins could be a consequence of the improved growth after *in vitro* evolution rather than a cause, it may nevertheless allow insights into the underlying mechanisms.

This issue confounds the interpretation of the FOXM1 results. FOXM1 is a cell cycle transcription factor. Over-expressing FOXM1 accelerates the S-G2-M transition. So it isn't surprising that FOXM1 over-expression enhances the growth of the aneuploid cells. But, consider what the authors write in the abstract: "We identified E2F4 and FOXM1 as transcription factors required for adaptation to aneuploidy *in vitro*...". Their results do not support the claim that FOXM1 is "required" for adaptation to aneuploidy. The missing experiment is to delete FOXM1 in the aneuploid cells and then show that after the same number of population doublings, the aneuploid cells don't grow any better. In the absence of that experiment, to be blunt, showing that over-expressing a cell cycle transcription factor enhances cell cycle progression is a trivial result.

RE> Our observation that FoxM1 expression in wildtype cells does not lead to enhanced proliferation clearly shows that this is not as trivial as put by the reviewer. We show that FOXM1 overexpression selectively improves proliferation in aneuploid cell lines, while it rather impairs proliferation in the parental HCT116 cells (Fig. 4H). In addition, only aneuploid cells profited from the overexpression of FOXM1 in the competition experiment (Fig. 4J). While FOXM1 is indeed a cell cycle transcription regulator with a positive effect on proliferation, other transcription factors that enhance cell cycle progression (e.g., E2F1 or YAP/TAZ coactivators) were not identified as contributing to the improved proliferation of aneuploid cells. Similarly, the expression of FOXM1 and its targets correlates with cancer aneuploidy in time (AADEPT score, Fig. 4E), while other proliferation-related transcription factors do not show this association (e.g., MYC). We now explain this argument better in our Discussion. Moreover, we have changed the statement in abstract as follows:

We identified E2F4 and FOXM1 as transcription factors strongly associated with adaptation to aneuploidy *in vitro*

We appreciate the suggestions of an evolution experiment with FoxM1 knock-out aneuploid cells, although the outcome and interpretation might be confounded by additional phenotypes caused by FoxM1 loss. For this, we have generated knock-out cell lines in wild type and in Htr5. The single cell clones selected after the CRISPR/Cas9-mediated FoxM1 loss were cultured in the same way as one of the modes of *in vitro* evolution (the cs clones, colony selection, Figure 1A, C). Indeed, the loss of FOXM1 reduced the ability of trisomic cells to proliferated in four clones, while the proliferation was not significantly affected in parental cell lines. This clearly shows that FOXM1 is required for proliferation of aneuploid cells at least in this experimental model, but most likely also in cancer, given the high AADEPT score. The new data are now part of Figure 5 and discussed in the manuscript on page 13 - 14.

3) Dosage compensation in aneuploid cells. I think that this topic deserves more attention. It seems like the authors have generated isogenic or near-isogenic cell lines that harbor the same aneuploidy

but exhibit differing levels of dosage compensation. This may be the first time that such a system has been created, and it could be used to really dive into how dosage compensation happens in cancer. Returning to point #1 above, I wonder if the "evolved" cells have mutations that enhance protein turnover? If not, how is the increase in dosage compensation occurring? Is translation affected? This could be tested by ribosome profiling. Or is protein turnover affected? This could be tested using pulse-chase labeling experiments. Getting some insight here would significantly improve the paper. The analysis of dosage-compensation is rather superficial. I see that in some cell lines, there is chromosome-wide dosage compensation while in others the signal is more modest. I think that it is worth looking at a recent bioRxiv paper from the Fojjier Lab (Rendo et al., bioRxiv 2023), which suggested that there are certain "toxic" genes that must be dosage compensated when present on an aneuploidy. Perhaps there are one or a few genes on each chromosome (maybe the ones highlighted by Rendo et al) that are being compensated?

RE> We agree with the reviewer that the dosage compensation is a very interesting effect which can potentially explain some of the observed improvements in polysomic cells, but it should be noted that there are no data suggesting that high degree of dosage compensation is associated with improved proliferation of human aneuploid cells. The adapted cell lines provide an exciting opportunity to evaluate the effects of dosage compensation on cell proliferation at the level of each individual protein, although it should be noted that the differences in dosage compensation are minor and not general among all cell lines. Given the availability of proteomics data, we reanalyzed the data with a focus on dosage compensation of specific genes (most overabundant or generally toxic/ tumor suppressive) located on the extra chromosomes before and after evolution. We found no evidence that the degree of dosage compensation is higher in cells with improved proliferation.

The mechanism of dosage compensation is still poorly understood. We have previously shown that trisomy leads to increased protein turnover, particularly via an upregulation of proteasomal degradation (Donnelly et al, Embo J, 2014), and that the proteasomal degradation is critical for dosage compensation (McShane et al, Cell, 2016). However, additional mechanisms are likely to contribute to dosage compensation as well, as shown by recent work from Santaguida and Ben-David labs (Ippolito et al, Cancer Discovery 2024).

Based on our mutation analysis, there are a few mutations in "evolved" cells that may enhance or alter protein dynamics and turnover. For example, in HCT116 we found CTIF, component of the translation initiation complex that also contributes to nonsense mediated mRNA decay; the E3 ligase TRIM38 that also regulates innate immunity; RPN1 which is associated with rough ER and with the regulatory subunit of the 26S proteasome; or EIF3B, a subunit of translation initiation factor 3 (new figures EV3). While the question whether translation and protein turnover is altered is very interesting, we consider ribosome profiling beyond the scope of this manuscript, in particular in the light of our finding that dosage compensation does not seem to directly affect proliferation.

The preprint from Fojjier lab (Rendo et al, 2023) is highly interesting, but it focuses on dosage compensation on RNA level. In our model systems we observed that dosage compensation is much stronger on the level of proteins than transcripts (Stingele et al, 2012, Selbach et al, 2016, Chunduri et al, 2021). Instead, we checked the relative protein abundance of COSMIC Cancer Gene Consensus tumor suppressor genes and oncogenes before and after evolution and observed isolated cases of significant changes in dosage compensation, but no general trend for multiple cell lines. The new data are now included in Appendix Figure S4 and discussed in the manuscript on page 7.

Or, I wonder, does the expression pattern from aneuploid chromosomes become more "tumor-like" over time? The authors could perform an "AADEPT"-like analysis, but focused specifically on which genes are over-expressed or compensated on gained chromosomes in tumors. Maybe in their engineered aneuploidy system, you get a raw over-expression of most genes, but over time you get evolution towards a balanced pattern that allows aneuploid tumors to grow robustly.

RE> While this is an interesting idea, it should be noted that there is an extensive dosage compensation of the gene expression of genes located on the gained chromosome in model trisomic cell lines already before evolution (see Stingle et al, 2012, McShane et al, 2016). Moreover, there is no evidence that increased dosage compensation improves proliferation of cancer cells.

However, we assessed the profiles of deregulated proteins encoded on the extra chromosomes before and after evolution and compared how they are expressed in corresponding aneuploid cancers of the CPTAC and found no general improvement or similarity. We showcase this in our results in Appendix Figure S4, and extensively discuss on page 7.

Referee #2 (Report for Author)

Summary: In this manuscript, the authors address the aneuploidy paradox (the fact that aneuploidy has a negative effect on cellular fitness but is a hallmark of cancer, a disease of uncontrolled cell growth) by aiming to identify expression changes that alleviate the negative consequences of aneuploidy on cell proliferation. The authors show that aneuploid cell lines created via microcell mediated chromosome transfer initially have a reduced proliferation rate compared to their parental cell lines, but allowing the cells to "evolve" over 50 passages resulted in a partial recovery of this fitness defect. Using proteogenomic analysis of the aneuploid cell lines before and after evolution identified expression changes in DNA replication and lysosomal genes, changes that the authors show are also observed in aneuploid tumors. The authors show that increased activity of the transcription factor FOXM1 is a common characteristic of the evolved aneuploid cells and highly aneuploid tumors and experimentally show that increasing FOXM1 expression can improve the proliferation of unevolved aneuploid cells. Finally, the authors show that the partial reversal of one of the studied trisomies (chr 5) is also associated with improved proliferation of cells in vitro and worse survival in cancer patients.

Overall impressions: Altogether, this manuscript makes the novel finding that the reversal of common aneuploidy-associated gene expression changes in DNA replication and lysosomal stress are associated with improved proliferation. This grants exciting insight into how aneuploidy can be tolerated and therefore so widely observed in tumors, potentially aiding the field in explaining the aneuploidy paradox in cancer. However, additional data is required to support the authors' conclusions.

RE> We thank the reviewer for this positive evaluation as well as for the suggestions how to improve the manuscript.

Major points:

1. The data shown in Figure 5D, E, F while very interesting would require additional data to confirm the conclusions. It is quite surprising the fact that deletion of the whole chr5 leads to almost zero colonies (as per figure and quantification). Since one of the clones that lost chr5 seems also to have

lost a telomeric portion of 8q where likely MYC is located, this could be a reason why this clone forms so few colonies. While this would not explain the behavior of the other clones, it suggests that the experiment would benefit from more clones (2-3 more) that lost chromosome 5 and ideally also 5p only. A more clear genomic profile including focal events should be added to the figure 5. Did the clones lose the same chromosome 5 (or 5p)? (paternal, maternal, the one added through MMCT) Deep whole genome sequencing should be able to address this point.

RE> We agree with the reviewer that the findings with the loss of chromosome 5 are very surprising. We appreciate the suggestion with the chr. 8q. Indeed, a reassessment of the sequencing data revealed that the third copy of MYC gene is lost in this cell line, however, immunoblotting to evaluate the expression of MYC in these clones did not show any significant abundance changes (Appendix Figure S9B). Moreover, we have analyzed additional clones from the REDACT experiment and found one more clone with disomy of chromosome 5 as verified with chromosome painting which shows the same phenotype as the two previously analyzed cell lines Hdi5_1 and Hdi5_2. We have not added this data to the manuscript, as they show exactly the same phenotype as the other two clones, but provide them for reviewer's perusal (Figure R2).

To understand the involved molecular mechanisms, we performed a proteomic analysis of these clones to further characterize the changes that lead to altered cell growth. This analysis revealed increased expression of factors associated with chromatid segregation and DNA replication in ReDACT Htr5p cell lines in comparison with the parental trisomy. Pathways associated with mitochondrial respiration and oxidative metabolism were downregulated. The changes related to replication and chromosome segregation are similar to those observed in the evolved cell lines and may contribute to the improved proliferation. However, we did not identify a specific factor on chromosome 5 that could be solely responsible for the observed phenotypes. No similar changes were observed in the ReDACT diploid clones Hdi5. These results are now a part of a new Figure EV5.

Based on deep WGS analysis we found that the exogenous arm of chromosome 5q was lost in the one case where we could analyze it. The analysis was possible only in one cell line, where the 30x WGS data were obtained. This might suggest that the exogenous chromosome 5 is particularly prone to damage or loss. We have decided not to include the data in the manuscript, since it is only one example and since, in our opinion, it does not change the conclusion that 5q gain has antiproliferative effects in HCT116 (and in many cancers).

- A. Chromosome paints with 32 quantified metaphases.
- B. Colony formation assay in 6 biological replicates with 3-4 technical replicates each.
- C. Representative western blot images for Chk1 phosphorylation and MC7 expression with respective quantification of three biological replicates with 1-4 technical replicates each.

2. Supp Figure 2E: Why doesn't p53 signaling increase between WT and polysomic cells? Is it because the aneuploidy was introduced via MMCT instead of missegregation, or are single gains not enough to activate p53? Others showed these cell lines do activate the pathway when aneuploid. See

<https://doi.org/10.1016/j.celrep.2021.108892>

RE> The p53 signaling is activated after acute missegregation, but not by constitutive aneuploidy. The mentioned paper, and several others, e.g., from Medema, Amon and Santaguida lab show that acute aneuploidy, 24 h – 48 h after chromosome missegregation induced by inhibition of spindle apparatus, leads to the activation of the p53 pathway. The p53 activation after acute missegregation has been previously linked to DNA damage and to oxidative stress arising after missegregation (e.g., Soto et al, Cell Reports 2017). In contrast, constitutive aneuploidy of a single chromosome (regardless of whether generated via MMCT or chromosome missegregation) does not activate this pathway.

3. In general, there needs to be more details/explanation of the connection between the markers of DNA replication stress investigated (DNA combing experiments, CHK1 signaling, γ H2AX) and the effect of FOXM1 overexpression/5q loss as they relate to proliferation rate (Supplementary Figure 7, Fig. 3, supp. Fig. 10). It is unclear to what degree/what kind of DNA damage or instability is different between WT, aneuploid, and evolved cells and how FOXM1 overexpression/5q loss might be contributing.

RE> We agree with the reviewer that understanding the link between DNA damage and FOXM1 overexpression/5q loss would help to recognize the contribution of genomic instability to the proliferation defect in these conditions. To this end, we analyzed DNA damage and genomic instability (γ H2AX and micronuclei) also in the cell lines overexpressing FOXM1, as well as in the engineered 5q loss clones. These experiments clearly showed that in all cases, the genomic instability and DNA damage is not reduced. However, we found that the FOXM1 levels reduced the number of cells in the G1 phase, likely facilitating the G1-S transfer, and increases the number of cells in G2 and M phase. These new results are now in the Figure 5E, F, and Appendix Figure S8. The loss of 5q combined with 5p gains was also not associated with any clear phenotype in DNA damage, but shows reduced CHK1 signaling; the disomic clones derived from the analysis revealed a strong reduction of replication proteins and increased CHK1 signaling and increased accumulation of DNA damage. These findings are now in Appendix Figures S9. Taken together, the data support the notion that either reduced DNA damage and genomic instability, or reduced signaling of the DNA damage is crucial for improved proliferation of aneuploid cells. We have now added additional text to Discussion.

a. When making the conclusion "We conclude that one of the strongest changes upon in vitro evolution of cells with additional chromosomes is an increased expression of DNA repair and replication factors and reduced genomic instability, which is reflected by reduced checkpoint signaling and decreased DNA damage," Why isn't γ -H2AX staining shown at this point? It is unclear how a change of +/-1% in anaphase bridges is supposed to contribute to the argument.

RE> We apologize for omitting the data for γ -H2AX in evolved and unevolved cell lines, which we now added to the manuscript (see Figures S2F, 5, S8, S9). The data show reduced γ -H2AX staining after evolution, but no clear phenotypes after FOXM1 overexpression or 5q loss, where rather reduced

checkpoint signaling has been observed. With also added text to Discussion of this new data, pointing out that also bypassing the DNA damage contributes to the improved proliferation.

b. What would be the significance of change in inter-origin distance?

RE> We believe that this reflects the MCM levels. It has been previously demonstrated that reduced MCM abundance results in increased inter-origin distance, and in turn in replication stress, as the areas to be replicated from one origin are larger (e.g., Ge et al, 2007, PNAS). This is indeed what we observe in polysomic cells. In contrast, upon evolution, the abundance of MCM increases and the distance is reduced, which corresponds to the reduced replication stress. We apologize for not making this clear. The explanation was now added to the revised version (page 11).

4. In general, there are a lot of assumptions that the observed changes in gene expression (especially in the tumor data) are related to the recovery of proliferation after evolution before showing causation. Did the authors consider or look at any other phenotypes that could differ before and after evolution and potentially contribute to malignancy in vivo (growth under different conditions, migration, etc)?

RE> We thank the reviewer for this interesting idea. We have not focused on this direction, because we are primarily interested in finding mechanisms underlying the defects in proliferation caused by aneuploidy, and how can cells adapt to this. We did not plan to address how this can contribute to malignancy in vivo, because in our opinion this would require an entirely different study design. We have now, out of curiosity, performed a scratch assay to test whether migration would be improved after in vitro evolution (Figure R2). This showed that the evolved aneuploid cells do not migrate better (somewhat reduced wound healing when thymidine, which inhibits proliferation, is included), but that they also in these conditions proliferate better (better wound healing when no thymidine is added in Htr5p50 compared to Htr5).

We did not add these results to the revised version, as we do not see any added value in these experiments. As explained above, we are convinced that experiments testing increased malignancy would have to be designed from the beginning differently and were not the focus of our project.

5. Fig. 4H and 5E: For the assessment of the effects of FOXM1 overexpression and 5q loss on proliferation of Htr5, why was a clonogenic assay (10 days) used instead of the MTT assay as used in figure 1 (4 days)? Was the effect on proliferation not noticeable after 4 days?

RE> The clonogenic assay works very robustly and reproducibly in our hands, which was the main reason to use it. Additionally, indeed, we expected that the effects would be weaker when evaluated only after 4 days. We now added proliferation assays using a cell based colorimetric assay for FOXM1 overexpression and knockdown cell lines and show the results in Appendix Figure 8C, D, G, H. This assay provided similar trends for FOXM1 overexpression, while no general trend of difference was noticeable after 4 days in FOXM1 depletion clones. We did not perform this assay for ReDACT clones, since the newly added proteome data suggest that the ATP metabolism is significantly altered and could thus confound the results if using ATP luminescence signal as a proxy for cell viability (Figure EV5B).

6. What is the proposed mechanism of FOXM1 restoring proliferation in evolved aneuploid cells? Is there any phenotype that could be compared before and after evolution (time in metaphase?) to further support it as the contributing factor to change in proliferation rate?

RE> This is indeed an important point, and we thank the reviewer for this suggestion. We have previously performed flow cytometry with these cells, and the DAPI-based cell cycle profile did show a small increase of G2/M fraction in Htr5 FOXM1, but not in HCT116 FOXM1. However, the difference in the time in mitosis would probably not result in any significant proliferation changes, since mitosis and specifically metaphase are very short compared to the entire cell cycle. Interestingly, however, our flow cytometry revealed a striking reduction of the G1 phase. Since trisomic cells have a significantly delayed G1-S transition (Williams et al, 2008; Stinglee et al, 2012), we believe that the reduced G1 phase can explain the improved proliferation. We added these new results into Figure 5F, G and EV4 H, I, and added a discussion of this aspect to the manuscript.

7. How frequent the loss of 5q was after evolution? Can the authors perform FISH or single cell sequencing on the evolved population? Can the competition assay be performed with the Htr5 and Htr5p cells? This would further support that trisomy of 5q gives cells a disadvantage.

RE> We thank the reviewer for this interesting idea. We used chromosome paints to analyze the evolved cell lines and to obtain a better overview of the frequency of the specific karyotypes. This data (now in expanded Figure S9A) show that while there is a certain population heterogeneity, majority of the cells have indeed lost the 5q arm after evolution. While the competition assay for Htr5 and Htr5p is a nice idea, we were not able to conduct this experiment, because the transduction efficiency was very low and variable in the Htr5p cells. We, however, believe that the proliferation assays support sufficiently the observation that 5p gain and 5q loss improves proliferation of trisomic cells.

8. The findings that the Hdi5 clones have reduced, rather than improved proliferation are very interesting and unexpected. It would be interesting to know if these cells that were reverted to the disomic state for chr 5 have expression patterns more closely resembling the trisomic cells or the parental HCT116 cell line. Can the authors propose an explanation for how the cells adapted to "depend" on chr 5 trisomy for proliferation, yet also evolve to minimize the stress induced by the gain over time?

RE> We share the reviewer's curiosity about the underlying mechanisms. To address the question, we performed proteomics analysis in these cells and compared it with the parental and evolved trisomies. Regarding the expression patterns of the Hdi5 clones, we do observe that they resemble Htr5 more than the parental HCT116 (Figure R4). However, this is also the case in Htr5p_2, and Htr5p_1 shows average differences in protein abundance relative to Htr5 and HCT116 that are both higher than in Hdi5_2. By this, we conclude that the resemblance of average expression patterns to HCT116 does not determine the differences in growth between Hdi5 and Htr5p clones. However, we do observe unique biological processes that are over-represented by proteins deregulated exclusively in Hdi5 clones (Figure EV5B, C). Among them are cytosolic ribosomal subunits and translation factors, which include PAIP1, a translation initiation factor encoded on chromosome 5p which is overexpressed in Htr5 (Figure EV5D). Similarly, TRIP13 and NIPBL, also encoded on chromosome 5p, show decreased expression in Hdi5 and are among the mitotic factors exclusively overabundant in the better growing Htr5p. We propose that perhaps it is these overabundant growth and cell cycle related proteins encoded on chromosome 5p that cells with constituent trisomy 5 evolve to rely on through altered protein homeostasis, leading to growth impairment and renewed proteostasis disruption once the extra chromosome arm is redacted. At the same time, trisomic cells could evolve to alleviate the stresses induced by the added dosage and protein abundance of the other genes encoded on the extra chromosome through increased FOXM1 activity or through removal of other extra DNA like chromosome arm 5q, whose encoded proteins are suggested to confer a net disadvantage to cellular fitness when overabundant. We show these results in Figure EV5 and describe them on page 15-16.

Minor points:

1. Fig. 4I,J: Replicate information is missing for the FOXM1 competition assays.

RE> Instead of triplicates we have performed a label swap experiment, which showed the same trend (see Sup. Fig. 9). In each of these experiments, we have evaluated the ratio of FoxM1 OE to untreated control cells at three different time points, which all show the same trend. For each datapoint a population of 50'000 cells was evaluated. However, if the editorial team insists, we are happy to add two additional replicates of the experiment shown in Fig. 4J, even though this seems to be beyond the state-of-the-art required for this type of experiments (see for example Girish et al., Science 2023).

2. Fig. 3H: Caption refers to anaphase bridges but the figure shows micronuclei percentages.

RE> We apologize for this error which will be corrected in the revised manuscript.

3. Consider citing Sheltzer 2017 in the background of trisomies causing proliferative defects that can be reversed through evolution/karyotype changes <http://dx.doi.org/10.1016/j.ccell.2016.12.004>

RE> We apologize for missing several important citations, which was now added.

Referee #3 (Report for Author)

Aneuploidy is a hallmark of cancer, a disease characterised by uncontrolled proliferation, yet it is very detrimental in untransformed cells. This suggests that aneuploid cancer cells can tolerate aneuploidy-associated cellular stresses. The study by Boekenkamp et al. identify adaptation strategies utilised by aneuploid cells to cope with the cellular stresses associated with the presence of an extra chromosome. They show that the enhanced proliferation of aneuploid cells was linked, among other factors, to diminished DNA damage and genomic instability, decreased replication stress, and modified expression of factors implicated in DNA replication and lysosomal degradation. These pathways and genes identified show a close correlation with the deregulated pathways observed in aneuploid cancers. The authors made several interesting observations, the data presented are convincing and the experiments well-done. The paper is well written, it reads well and the main conclusions are compelling and novel. Thus, the paper is suitable for publication in The EMBO Journal. This Reviewer has only minor points, listed below, that can be helpful in improving the work:

RE> We thank the reviewer for this very positive evaluation.

In the introduction, when referring to stresses present in cells with constitutive gain of an extra chromosome, it would be helpful to cite: <https://doi.org/10.1101/2023.01.27.525822>.
<https://doi.org/10.1101/2023.01.27.525826>

When discussing the selective advantage provided by aneuploidy, it would be worth citing: PMID: 34352223 34352222 34266887

RE> We apologize for omitting these important references. We now present findings from these papers as they are highly relevant to our results.

This study focuses on stable aneuploid clones. It would be worth trying to induce chromosome mis-segregation in HCT116 expressing or not FOXM1-dNdK and see whether this is enough to rescue also the proliferation defects observed upon acute induction of aneuploidy.

RE> We agree with the reviewer that this is a very interesting topic. However, the situation immediately after chromosome missegregation is much more complex because various chromosomes are missegregated, and gains, losses, and chromosome breakages occur (see Soto et al, 2017, Santaguida et al, 2017). Moreover, the p53 pathway is often activated immediately after chromosome missegregation. We are convinced that this experiment requires its own study and is beyond the scope of this manuscript. Additionally, findings implicating FOXM1 in cellular response to chromosome missegregation has been recently published by Sotillo group, and we have cited this paper.

From our manuscript: Moreover, FOXM1 has been found to support proliferation of chromosomally unstable and aging aneuploid cells in both mouse and human (Macedo et al., 2018, Pan et al., 2023).

(From the abstract from Pan et al, CDD 2023: Notably, we observed that FOXM1 is upregulated upon aneuploid induction in cells with dysfunctional SAC and error-prone mitosis, and these cells are sensitive to FOXM1 knockdown, indicating a novel vulnerability of aneuploid cells.)

Figure 4J lacks error bars.

RE>Instead of triplicates we have performed a label swap experiment, which showed the same trend (see Sup. Fig. 9). In each of these experiments, we have evaluated the ratio of FoxM1 OE to untreated control cells at three different time points, which all show the same trend. For each datapoint a population of 50,000 cells was evaluated. It should be noted that the competition experiments are often rather variable, as they are strongly affected by the efficiency of transfection and by the ratio of seeding at day0. Therefore, we have not calculated the average, but rather show individual experiments. Adding additional replicates of the experiment shown in Fig. 4J seems to be beyond the state-of-the-art required for this type of experiments (see for example Girish et al., Science 2023).

Dr. Zuzana Storchova
RPTU Kaiserslautern
Molecular Genetics
Paul Ehrlich Strasse 24
Kaiserslautern 67663
Germany

9th Dec 2024

Re: EMBOJ-2024-116981R
Proteogenomic analysis reveals adaptive strategies to alleviate the consequences of aneuploidy

Dear Zuzana,

Thank you for submitting your revised manuscript to The EMBO Journal, and my sincere apologies for the delay in getting back to you with a post-re-review decision. We received feedback from all three original referees, which unfortunately was not straightforward and necessitated careful further considerations and discussions within our editorial team. As you will see from the comments copied below, all referees consider the study significantly improved, but only referee 3 is already unconditionally in favor of acceptance. Referee 2 retains a few specific concerns, while referee 1 still feels that further extensions of the study would be needed to make it sufficiently insightful.

After detailed deliberations of the latest reports, also in light of the original reviews and your response letter, we decided that we would be open to pursuing the study further for The EMBO Journal, pending adequate answering of referee 2's remaining concerns. On the other hand, we concluded that referee 1 is to some extent requesting new experiments, while substantial revision work has already been added in response to their original comments. I am therefore returning the study to you once more for a final round of revision, in which I would ask you to respond to all remaining referee points, and to address the issues noted by referee 2. Ideally, please do get back to me briefly on how you could envision addressing these points.

When preparing a final resubmission, please also take care of a number of important editorial requirements:

- On the abstract page of the manuscript, please include 4-5 general keyword terms to enhance searchability.
- As we are switching from a free-text author contribution statement towards a more formal statement based on Contributor Role Taxonomy (CRediT) terms, please remove the present Author Contribution section and instead specify each author's contribution(s) directly in the Author Information page of our submission system during upload of the final manuscript. See <https://casrai.org/credit/> for more information.
- Please correct the reference list, making sure that for references with more than 10 authors on a paper, only the first 10 should be listed, followed by 'et al.' (please refer to our Guide to Authors for additional information on EMBO J reference format).
- Please double-check to make sure to all relevant funding information in the manuscript is congruent with the info entered into our submission system; they currently appear to be very different.
- Our routine revision image checks indicated that the Ponceau-stained loading control is identical between Appendix Figures S8D and S8E - please check & clarify this issue, if necessary with relevant raw data images. Furthermore, although Appendix Figure S6A is already mentioned as an "extended version", its figure legends (as well as the legends to Figure 3C) need to explicitly mention the respective figure in which duplicate blots are displayed a second time.
- Tables EV1-8 should be renamed into Dataset EV1-8 (in the files and throughout the text); their legends should be removed from the main text and instead included in each of the respective spreadsheet files, in a separate "legends" tab.
- For the Appendix material, Appendix figure legends should be removed from main manuscript file and instead included in the Appendix PDF, each legend below the respective figure. The Appendix further needs a title page with a Table of Contents and page numbers.
- In the Data Availability section, please remove referee access tokens at this stage, and make sure to include a direct URL to the respective database in which each deposited datasets can be accessed. On the other hand, previously generated datasets should not be included here, but instead be referenced as "data citations", as explained in our Guide to Authors: <https://www.embopress.org/page/journal/14602075/authorguide#referencesformat>
- Please provide suggestions for a short 'blurb' text prefacing and summing up the study in two sentences (max. 250 characters),

followed by 3-5 one-sentence 'bullet points' with brief factual statements of key results of the paper; they will form the basis of an editor-written 'Synopsis' accompanying the already provided synopsis image in the online version of the article.

- Please note that Source data files need to be saved in a scheme one figure/one folder, and then uploaded as .zip files. E.g. all the Source data files for figure 1 need to be saved in a single folder and this needs to be zipped and then uploaded as "SD figure 1.zip" file. For EV and/or appendix figures, on the other hand, ZIP together all source data in one combined archive.

- Finally, during routine pre-acceptance checks, our data editors have raised the following queries regarding figures, data, and legends; I would appreciate if you briefly answered to them in the cover letter of your final submission, and made the requested text modifications with changes/additions highlighted via the "Track changes" option, to facilitate our final checking.

1. Please note that the exact p values are not provided in the legends of figures 3E, 4C, E; 5C, J; 6E; EV4 A, B, E
2. Please indicate the statistical test used for data analysis in the legends of figures 2E, 4A, EV2 B; EV3 B; EV5 D.
3. Please note that the box plots need to be defined in terms of minima, maxima, bounds of box and whiskers, and percentile in the legends of figures 5C, J
4. Please note that the box plots need to be defined in terms of minima, maxima, centre, bounds of box and whiskers, and percentile in the legends of figures 6E; EV4 A, B.
5. Please note that information related to n is missing in the legends of figures 1B, EV1 B, C.
6. Although 'n' is provided, please describe the nature of entity for 'n' in the legends of figures 4C, D; 5C, J; EV4 A, B, D, E, F.
7. Please note that the error bars are not defined in the legends of figures 6F, EV3 F.
8. Please note that for heatmap present in figure EV2 C a numbered scale bar is not provided. This needs to be rectified.

Once we will have received your re-revised version, we should hopefully be able to swiftly proceed with acceptance and publication of the study. Please do not hesitate to contact me, should in you have any questions in this regard.

With kind regards,

Hartmut

9) To facilitate reproducibility and cross-laboratory adoption of methodologies, please structure the Materials & Methods section as outlined in our guide to authors, including a completed Reagents and Tools Table that can be downloaded from our author guidelines as well (<https://www.embopress.org/page/journal/14602075/authorguide#structuredmethods>).

10) Digital image enhancement is acceptable practice, as long as it accurately represents the original data and conforms to community standards. If a figure has been subjected to significant electronic manipulation, this must be clearly noted in the figure legend and/or the 'Materials and Methods' section. The editors reserve the right to request original versions of figures and the original images that were used to assemble the figure. Finally, we generally encourage uploading of numerical as well as gel/blot image source data; for details see: embopress.org/page/journal/14602075/authorguide#sourcedata

At EMBO Press, we ask authors to provide source data for the main manuscript figures. Our source data coordinator will contact you to discuss which figure panels we would need source data for and will also provide you with helpful tips on how to upload and organize the files.

In the interest of ensuring the conceptual advance provided by the work, we recommend submitting a revision within 3 months (9th Mar 2025). Please discuss the revision progress ahead of this time with the editor if you require more time to complete the revisions. Use the link below to submit your revision:

Link Not Available

Referee #1:

The new manuscript is significantly improved. However, I remain concerned that the central focus of the manuscript - the upregulation in cell cycle genes like FOXM1 - is a byproduct of the improved proliferation of aneuploid cells and not an actual cause of the improved proliferation. One experiment arguing against that interpretation is the competition experiment involving the over-expression of FOXM1 in the trisomic cells.

I am glad that the authors were able to sequence the evolved cell lines and identify mutations. It's interesting that the authors found a mutation in 53BP1 in the evolved RPE1 trisomy 21 cells and that suppressing 53BP1 enhanced proliferation in that cell line. They also found a mutation in ERCC6, another DNA repair-associated gene. Does suppressing this gene also increase fitness in the trisomy 21 cells?

More broadly - to increase the potential impact of this work for a high-profile venue like EMBO Journal, I think that it would be important to investigate whether their findings are specific to the individual cell line that they did the experiment in, specific to the gained chromosome, or general across aneuploid cells. I would like to see them 1) overexpress active FOXM1 and 2) knock down 53BP1, in other wild-type/aneuploid cell line pairs. Lots of Trisomy 21 cell lines are available - do they see the same phenotype when they knock down 53BP1, or is this specific for RPE1? Does overexpressing FOXM1 enhance the growth of multiple aneuploid cells? Perhaps there is a link to PMID: 30026603.

Referee #2:

The authors successfully addressed specific comments and critiques.

As we mentioned in our previous comments, for figure 5E (competition assay), it would be good to have replicates because it is a critical experiment and because the data for HCT116 cells over expressing FOXM1 are at odds with the literature, where

FOXM1 promotes proliferation in these cells (HCT116). See references below.

In the competition, HCT116 and Htr5 should be mixed together (transduced with empty vector); also HCT116 FOXM1 and Htr5 FOXM1 should be mixed together. Most importantly, the experiment should be repeated with several aneuploid clones for HCT116 and RPE.

I also agree with other reviewers that the Figure 5I-J should be done with proliferation assay not colonies as in the case of colonies one should also count the total number of them which seem different than their size.

References:

FoxM1 transactivates PTTG1 and promotes colorectal cancer cell migration and invasion Yun Zheng, Jinjun Guo, Jin Zhou, Jinjian Lu, Qi Chen, Cui Zhang, Chen Qing, H. Philip Koeffler & Yunguang Tong BMC Medical Genomics volume 8, Article number: 49 (2015)

FOXM1 promotes the growth and metastasis of colorectal cancer via activation of β -catenin signaling pathway Kankan Yang, Bing Jiang, Yecai Lu 1, Qingbing Shu, Pan Zhai, Qiaoming Zhi, Qixin Li PMID: 31118796 (2019)

Referee #3:

The authors have satisfactorily addressed my prior points.

Also, they have carefully considered comments and suggestions from the other Reviewers and have provided convincing responses. Thus, I strongly support publication in The EMBO Journal.

Response to the reviewers' comments

Referee #1:

The new manuscript is significantly improved. However, I remain concerned that the central focus of the manuscript - the upregulation in cell cycle genes like FOXM1 - is a byproduct of the improved proliferation of aneuploid cells and not an actual cause of the improved proliferation. One experiment arguing against that interpretation is the competition experiment involving the over-expression of FOXM1 in the trisomic cells.

>> We thank the reviewer for considering the revised manuscript version as significantly improved. As the reviewer mentions, the competition experiment involving the overexpression of FOXM1 in the trisomic cells provides a compelling argument for a more direct role of FOXM1 in adaptation to aneuploidy. Other experiments, which are beyond the scope of this study, will be required to further clarify the exact mechanisms underlying the FOXM1 function in aneuploidy.

I am glad that the authors were able to sequence the evolved cell lines and identify mutations. It's interesting that the authors found a mutation in 53BP1 in the evolved RPE1 trisomy 21 cells and that suppressing 53BP1 enhanced proliferation in that cell line. They also found a mutation in ERCC6, another DNA repair-associated gene. Does suppressing this gene also increase fitness in the trisomy 21 cells?

>> While this is certainly an interesting experiment, it is beyond the scope of this study and will be an objective of future studies.

More broadly - to increase the potential impact of this work for a high-profile venue like EMBO Journal, I think that it would be important to investigate whether their findings are specific to the individual cell line that they did the experiment in, specific to the gained chromosome, or general across aneuploid cells. I would like to see them 1) overexpress active FOXM1 and 2) knock down 53BP1, in other wild-type/aneuploid cell line pairs. Lots of Trisomy 21 cell lines are available - do they see the same phenotype when they knock down 53BP1, or is this specific for RPE1? Does overexpressing FOXM1 enhance the growth of multiple aneuploid cells? Perhaps there is a link to PMID: 30026603.

>> While this is certainly an interesting experiment, it is beyond the scope of this study and will be an objective of future studies.

Referee #2:

The authors successfully addressed specific comments and critiques.

1. As we mentioned in our previous comments, for figure 5E (competition assay), it would be good to have replicates because it is a critical experiment and because the data for HCT116 cells over expressing FOXM1 are at odds with the literature, where FOXM1 promotes proliferation in these cells (HCT116). See references below.

>> As already explained in the first revision, the accepted standard is two replicates with color swap (see Girish et al, 2023, Lukow et al, 2021). The reason provided by the reviewer for adding the replicates is that other literature suggests that FOXM1 overexpression promotes proliferation in HCT116. However, none of the two publications suggested by the referee supports this statement.

In the first publication (*FoxM1 transactivates PTTG1 and promotes colorectal cancer cell migration and invasion*, Yun Zheng, Jinjun Guo, Jin Zhou, Jinjian Lu, Qi Chen, Cui Zhang, Chen Qing, H. Philip Koeffler & Yunguang Tong, BMC Medical Genomics volume 8, Article number: 49, PMID: 26264222, 2015), no *in vitro* proliferation assays were performed. The authors show that FoxM1 depletion impaired migration and invasion of HCT116, while “. . . transfection of FoxM1 plasmids did not further enhance the migration and invasion of WT or PTTG1^{-/-} HCT116 cells”. Additionally, FOXM1 overexpression in HCT116 accelerated metastasis formation upon injection.

The second paper (*FOXM1 promotes the growth and metastasis of colorectal cancer via activation of β -catenin signaling pathway*, Kankan Yang, Bing Jiang, Yecai Lu, Qingbing Shu, Pan Zhai, Qiaoming Zhi, Qixin Li, PMID: 31118796, 2019) shows no data on FOXM1 overexpression in HCT116. In the relevant figure, Fig. 3, the paper shows that overexpression of FOXM1 increases proliferation in another cell line, DLD1. DLD1 and HCT116 are both mismatch repair deficient, colorectal cancer cell lines, but differ in many other aspects, such as p53 status, WNT dependency, etc. Thus, the findings from DLD1 cannot be transferred directly to HCT116 and *vice versa*. Therefore, none of the published findings are contradicting our observation that FOXM1 overexpression facilitates growth of cells with extra chromosomes. We have extensively scanned the literature, yet we did not find data suggesting that FOXM1 overexpression per se directly improves proliferation in HCT116. Thus, the referee's reasoning is not justified.

2. In the competition, HCT116 and Htr5 should be mixed together (transduced with empty vector); also HCT116 FOXM1 and Htr5 FOXM1 should be mixed together.

>> While this experiment is possible, with all due respect we do not see any benefit from performing competition assays with mixed HCT116 and Htr5 with and without FOXM1 overexpression. Nowhere in the manuscript we claim that Htr5 (evolved, or with FOXM1 overexpression) proliferates better than the parental cell line; rather, we observed that evolved polysomic cell lines do not improve the proliferation beyond the parental cell line. Importantly, we always compare polysomic cells or diploid cells within each other (evolved vs. non-evolved, with vs without FOXM1 overexpression, etc.). We consider this the only relevant comparison. Furthermore, this was not mentioned in the first round of reviewer comments, and the reviewer does not explain why s/he requests this experiment and which gap in knowledge would this experiment fill. As such, we consider that this suggested experiment will not add any new insight to the manuscript.

3. Most importantly, the experiment should be repeated with several aneuploid clones for HCT116 and RPE.

>> We consider this experiment beyond the scope of this study. We have analyzed several different aneuploidies in *in vitro* evolution and focused then on specific cell lines. In this study, we showcase the specificity of the link between FOXM1 expression and improved growth of aneuploid cancer cells through experiments conducted with trisomic Htr5 and parental HCT116 as a control. The universality of our findings is already underpinned by the prominent overexpression of FOXM1 and its targets, specifically in aneuploid primary tumors of both the TCGA and CPTAC databases (see AADEPT scores, manuscript Figure 4 C, E & F). Additionally, this request was not mentioned in the first round of the reviews.

4. I also agree with other reviewers that the Figure 5I-J should be done with proliferation assay not colonies as in the case of colonies one should also count the total number of them which seem different than their size.

>> We have performed metabolite based MTT cell viability assay (MTT) for the FOXM1 knockdown cell lines, which did not provide conclusive results. This can have several reasons. First, the assay

covers only a short period of time (usually 4 days). Second, the differences in proliferation that we observe using the MTT assay are characteristic of a whole population (batch result). These were the reasons for choosing the clonogenic assay, where proliferation is allowed for over 10 days. The number of colonies in clonogenic assays reflects colony forming ability and with that the survival of cells (how many cells can form a colony), not their proliferation. Colony size, which correlates with the number of cells formed during the 10 days of the growth, reflects the proliferation more directly. All these reasons were clarified in our previous response and were accepted by other reviewers.

Referee #3:

The authors have satisfactorily addressed my prior points.

Also, they have carefully considered comments and suggestions from the other Reviewers and have provided convincing responses. Thus, I strongly support publication in The EMBO Journal.

>> We thank the reviewer for this positive evaluation.

Dr. Zuzana Storchova
RPTU Kaiserslautern
Molecular Genetics
Paul Ehrlich Strasse 24
Kaiserslautern 67663
Germany

21st Jan 2025

Re: EMBOJ-2024-116981R1
Proteogenomic analysis reveals adaptive strategies for alleviating the consequences of aneuploidy in cancer

Dear Zuzana,

Thank you for submitting your final revised manuscript for our consideration. I am pleased to inform you that we have now accepted it for publication in The EMBO Journal.

With kind regards,

Hartmut
